# Traditional and Heavy Tailed Self Regularization in Neural Network Models

## Abstract

Random Matrix Theory (RMT) is applied to analyze the weight matrices of Deep Neural Networks (DNNs), including both production quality, pre-trained models such as AlexNet and Inception, and smaller models trained from scratch, such as LeNet5 and a miniature-AlexNet. Empirical and theoretical results clearly indicate that the empirical spectral density (ESD) of DNN layer matrices displays signatures of traditionally-regularized statistical models, even in the absence of exogenously specifying traditional forms of regularization, such as Dropout or Weight Norm constraints. Building on recent results in RMT, most notably its extension to Universality classes of Heavy-Tailed matrices, we develop a theory to identify *5+1 Phases of Training*, corresponding to increasing amounts of *Implicit Self-Regularization*. For smaller and/or older DNNs, this Implicit Self-Regularization is like traditional Tikhonov regularization, in that there is a "size scale" separating signal from noise. For state-of-the-art DNNs, however, we identify a novel form of *Heavy-Tailed Self-Regularization*, similar to the self-organization seen in the statistical physics of disordered systems. This implicit Self-Regularization can depend strongly on the many knobs of the training process. By exploiting the generalization gap phenomena, we demonstrate that we can cause a small model to exhibit all 5+1 phases of training simply by changing the batch size.

## 1 Introduction

The inability of optimization and learning theory to explain and predict the properties of NNs is not a new phenomenon. From the earliest days of DNNs, it was suspected that VC theory did not apply to these systems (1). It was originally assumed that local minima in the energy/loss surface were responsible for the inability of VC theory to describe NNs (1), and that the mechanism for this was that getting trapped in local minima during training limited the number of possible functions realizable by the network. However, it was very soon realized that the presence of local minima in the energy function was *not* a problem in practice (2; 3). Thus, another reason for the inapplicability of VC theory was needed. At the time, there did exist other theories of generalization based on statistical mechanics (4; 5; 6; 7), but for various technical and nontechnical reasons these fell out of favor in the ML/NN communities. Instead, VC theory and related techniques continued to remain popular, in spite of their obvious problems.

More recently, theoretical results of Choromanska et al. (8) (which are related to (4; 5; 6; 7)) suggested that the Energy/optimization Landscape of modern DNNs resembles the Energy Landscape of a zero-temperature *Gaussian* Spin Glass; and empirical results of Zhang et al. (9) have again pointed out that VC theory does not describe the properties of DNNs. Martin and Mahoney then suggested that the Spin Glass analogy may be useful to understand severe overtraining versus the inability to overtrain in modern DNNs (10).

We should note that it is not even clear how to define DNN regularization. The challenge in applying these well-known ideas to DNNs is that DNNs have *many* adjustable "knobs and switches," independent of the Energy Landscape itself, most of which can affect training accuracy, in addition to *many* model parameters. Indeed, nearly anything that improves generalization is called regularization (11). Evaluating and comparing these methods is challenging, in part since there are so many, and in part since they are often constrained by systems or other not-traditionally-ML considerations.

Motivated by this situation, we are interested here in two related questions.

- **Theoretical Question.** Why is regularization in deep learning seemingly quite different than regularization in other areas on ML; and what is the right theoretical framework with which to investigate regularization for DNNs?
- **Practical Question.** How can one control and adjust, in a theoretically-principled way, the many knobs and switches that exist in modern DNN systems, e.g., to train these models efficiently and effectively, to monitor their effects on the global Energy Landscape, etc.?

That is, we seek a *Practical Theory of Deep Learning*, one that is prescriptive and not just descriptive. This theory would provide useful tools for practitioners wanting to know *How* to characterize and control the Energy Landscape to engineer larger and betters DNNs; and it would also provide theoretical answers to broad open questions as *Why* Deep Learning even works.

**Main Empirical Results.** Our main empirical results consist in evaluating empirically the ESDs (and related RMT-based statistics) for weight matrices for a suite of DNN models, thereby probing the Energy Landscapes of these DNNs. For older and/or smaller models, these results are consistent with implicit *Self-Regularization* that is Tikhonov-like; and for modern state-of-the-art models, these results suggest novel forms of *Heavy-Tailed Self-Regularization*.

- **Self-Regularization in old/small models.** The ESDs of older/smaller DNN models (like LeNet5 and a toy MLP3 model) exhibit weak *Self-Regularization*, well-modeled by a perturbative variant of MP theory, the Spiked-Covariance model. Here, a small number of eigenvalues pull out from the random bulk, and thus the MP Soft Rank and Stable Rank both decrease. This weak form of *Self-Regularization* is like Tikhonov regularization, in that there is a "size scale" that cleanly separates "signal" from "noise," but it is different than explicit Tikhonov regularization in that it arises implicitly due to the DNN training process itself.
- **Heavy-Tailed Self-Regularization.** The ESDs of larger, modern DNN models (including AlexNet and Inception and nearly every other large-scale model we have examined) deviate strongly from the common Gaussian-based MP model. Instead, they appear to lie in one of the very different Universality classes of Heavy-Tailed random matrix models. We call this *Heavy-Tailed Self-Regularization*. The ESD appears Heavy-Tailed, but with finite support. In this case, there is *not* a "size scale" (even in the theory) that cleanly separates "signal" from "noise."

**Main Theoretical Results.** Our main theoretical results consist in an operational theory for DNN Self-Regularization. Our theory uses ideas from RMT—both vanilla MP-based RMT as well as extensions to other Universality classes based on Heavy-Tailed distributions—to provide a visual taxonomy for $5+1$ *Phases of Training*, corresponding to increasing amounts of Self-Regularization.

- **Modeling Noise and Signal.** We assume that a weight matrix $\mathbf{W}$ can be modeled as $\mathbf{W} \simeq \mathbf{W}^{rand} + \Delta^{sig}$, where $\mathbf{W}^{rand}$ is "noise" and where $\Delta^{sig}$ is "signal." For small to medium sized signal, $\mathbf{W}$ is well-approximated by an MP distribution—with elements drawn from the Gaussian Universality class—perhaps after removing a few eigenvectors. For large and strongly-correlated signal, $\mathbf{W}^{rand}$ gets progressively smaller, but we can model the non-random strongly-correlated signal $\Delta^{sig}$ by a Heavy-Tailed random matrix, i.e., a random matrix with elements drawn from a Heavy-Tailed (rather than Gaussian) Universality class.
- **5+1 Phases of Regularization.** Based on this, we construct a practical, visual taxonomy for 5+1 Phases of Training. Each phase is characterized by stronger, visually distinct signatures in the ESD of DNN weight matrices, and successive phases correspond to decreasing MP Soft Rank and increasing amounts of *Self-Regularization*. The 5+1 phases are: RANDOM-LIKE, BLEEDING-OUT, BULK+SPIKES, BULK-DECAY, HEAVY-TAILED, and RANK-COLLAPSE.

Based on these results, we speculate that all well optimized, large DNNs will display *Heavy-Tailed Self-Regularization* in their weight matrices.

**Evaluating the Theory.** We provide a detailed evaluation of our theory using a smaller MiniAlexNew model that we can train and retrain.

- **Effect of Explicit Regularization.** We analyze ESDs of MiniAlexNet by removing all explicit regularization (Dropout, Weight Norm constraints, Batch Normalization, etc.) and characterizing how the ESD of weight matrices behave during and at the end of Backprop training, as we systematically add back in different forms of explicit regularization.
- **Exhibiting the 5+1 Phases.** We demonstrate that we can exhibit all 5+1 phases by appropriate modification of the various knobs of the training process. In particular, by decreasing the batch size from 500 to 2, we can make the ESDs of the fully-connected layers of MiniAlexNet vary continuously from RANDOM-LIKE to HEAVY-TAILED, while increasing generalization accuracy along the way. These results illustrate the *Generalization Gap* pheneomena (12; 13; 14), and

they explain that pheneomena as being caused by the implicit Self-Regularization associated with models trained with smaller and smaller batch sizes.

## 2 BASIC RANDOM MATRIX THEORY (RMT)

In this section, we summarize results from RMT that we use. Several overviews of RMT are available ([15]; [16]; [17]; [18]; [19]; [20]; [21]; [22]). Here, we will describe a more general form of RMT.

### 2.1 MARCHENKO-PASTUR (MP) THEORY FOR RECTANGULAR MATRICES

MP theory considers the density of singular values $\rho(\nu_i)$ of random rectangular matrices $\mathbf{W}$. This is equivalent to considering the density of eigenvalues $\rho(\lambda_i)$, i.e., the ESD, of matrices of the form $\mathbf{X} = \mathbf{W}^T\mathbf{W}$. MP theory then makes strong statements about such quantities as the shape of the distribution in the infinite limit, it's bounds, expected finite-size effects, such as fluctuations near the edge, and rates of convergence.

To apply RMT, we need only specify the number of rows and columns of $\mathbf{W}$ and assume that the elements $W_{i,j}$ are drawn from a distribution that is a member of a certain *Universality class* (there are different results for different Universality classes). RMT then describes properties of the ESD, even at finite size; and one can compare perdictions of RMT with empirical results. Most well-known is the Universality class of Gaussian distributions. This leads to the basic or vanilla MP theory, which we describe in this section. More esoteric—but ultimately more useful for us—are Universality classes of Heavy-Tailed distributions. In Section 2.2, we describe this important variant.

**Gaussian Universality class.** We start by modeling $\mathbf{W}$ as an $N \times M$ random matrix, with elements from a Gaussian distribution, such that: $W_{ij} \sim N(0, \sigma_{mp}^2)$. Then, MP theory states that the ESD of the correlation matrix, $\mathbf{X} = \mathbf{W}^T\mathbf{W}$, has the limiting density given by the MP distribution $\rho(\lambda)$:

$$\rho_N(\lambda) \xrightarrow[Q \text{ fixed}]{N \to \infty} \begin{cases} \dfrac{Q}{2\pi\sigma_{mp}^2} \dfrac{\sqrt{(\lambda^+ - \lambda)(\lambda - \lambda^-)}}{\lambda} & \text{if } \lambda \in [\lambda^-, \lambda^+] \\ 0 & \text{otherwise.} \end{cases}$$

Here, $\sigma_{mp}^2$ is the element-wise variance of the original matrix, $Q = N/M \geq 1$ is the aspect ratio of the matrix, and the minimum and maximum eigenvalues, $\lambda^\pm$, are given by

$$\lambda^\pm = \sigma_{mp}^2 \left(1 \pm \frac{1}{\sqrt{Q}}\right)^2. \tag{1}$$

**Finite-size Fluctuations at the MP Edge.** In the infinite limit, all fluctuations in $\rho_N(\lambda)$ concentrate very sharply at the MP edge, $\lambda^\pm$, and the distribution of the maximum eigenvalues $\rho_\infty(\lambda_{max})$ is governed by the TW Law. Even for a single finite-sized matrix, however, MP theory states the upper edge of $\rho(\lambda)$ is very sharp; and even when the MP Law is violated, the TW Law, with finite-size corrections, works very well at describing the edge statistics. When these laws are violated, this is very strong evidence for the onset of more regular non-random structure in the DNN weight matrices, which we will interpret as evidence of *Self-Regularization*.

### 2.2 HEAVY-TAILED EXTENSIONS OF MP THEORY

MP-based RMT is applicable to a wide range of matrices; but it is *not* in general applicable when matrix elements are strongly-correlated. Strong correlations appear to be the case for many well-trained, production-quality DNNs. In statistical physics, it is common to *model* strongly-correlated systems by Heavy-Tailed distributions ([32]). The reason is that these models exhibit, more or less, the same large-scale statistical behavior as natural phenomena in which strong correlations exist ([32]; [19]). Moreover, recent results from MP/RMT have shown that new Universality classes exist for matrices with elements drawn from certain Heavy-Tailed distributions ([19]).

We use these Heavy-Tailed extensions of basic MP/RMT to build an operational and phenomenological theory of Regularization in Deep Learning; and we use these extensions to justify our analysis of both *Self-Regularization* and *Heavy-Tailed Self-Regularization*. Briefly, our theory for simple *Self-Regularization* is insipred by the Spiked-Covariance model of Johnstone ([33]) and it's interpretation as a form of *Self-Organization* by Sornette ([34]); and our theory for more sophisticated *Heavy-Tailed Self-Regularization* is inspired by the application of MP/RMT tools in quantitative finance by Bouchaud, Potters, and coworkers ([35]; [36]; [37]; [23]; [25]; [19]; [22]), as well as the relation of Heavy-Tailed phenomena more generally to *Self-Organized Criticality* in Nature ([32]). Here, we

| | Generative Model w/ elements from Universality class | Finite-$N$ Global shape $\rho_N(\lambda)$ | Limiting Global shape $\rho(\lambda)$, $N \to \infty$ | Bulk edge Local stats $\lambda \approx \lambda^+$ | (far) Tail Local stats $\lambda \approx \lambda_{max}$ |
|---|---|---|---|---|---|
| Basic MP | Gaussian | MP, i.e., Eqn. (1) | MP | TW | No tail. |
| Spiked-Covariance | Gaussian, + low-rank perturbations | MP + Gaussian spikes | MP | TW | Gaussian |
| Heavy tail, $4 < \mu$ | (Weakly) Heavy-Tailed | MP + PL tail | MP | Heavy-Tailed* | Heavy-Tailed* |
| Heavy tail, $2 < \mu < 4$ | (Moderately) Heavy-Tailed (or "fat tailed") | PL** $\sim \lambda^{-(a\mu+b)}$ | PL $\sim \lambda^{-(\frac{1}{2}\mu+1)}$ | No edge. | Frechet |
| Heavy tail, $0 < \mu < 2$ | (Very) Heavy-Tailed | PL** $\sim \lambda^{-(\frac{1}{2}\mu+1)}$ | PL $\sim \lambda^{-(\frac{1}{2}\mu+1)}$ | No edge. | Frechet |

Table 1: Basic MP theory, and the spiked and Heavy-Tailed extensions we use, including known, empirically-observed, and conjectured relations between them. Boxes marked "*" are best described as following "TW with large finite size corrections" that are likely Heavy-Tailed (23), leading to bulk edge statistics and far tail statistics that are indistinguishable. Boxes marked "**" are phenomenological fits, describing large ($2 < \mu < 4$) or small ($0 < \mu < 2$) finite-size corrections on $N \to \infty$ behavior. See (24; 23; 25; 26; 27; 28; 29; 30; 19; 31) for additional details.

highlight basic results for this generalized MP theory; see (24; 23; 25; 26; 27; 28; 29; 30; 19; 31) in the physics and mathematics literature for additional details.

**Universality classes for modeling strongly correlated matrices.** Consider modeling $\mathbf{W}$ as an $N \times M$ random matrix, with elements drawn from a Heavy-Tailed—e.g., a Pareto or Power Law (PL)—distribution:

$$W_{ij} \sim P(x) \sim \frac{1}{x^{1+\mu}}, \quad \mu > 0. \tag{2}$$

In these cases, if $\mathbf{W}$ is element-wise Heavy-Tailed, then the ESD $\rho_N(\lambda)$ likewise exhibits Heavy-Tailed properties, either globally for the entire ESD and/or locally at the bulk edge.

Table 1 summarizes these recent results, comparing basic MP theory, the Spiked-Covariance model, and Heavy-Tailed extensions of MP theory, including associated Universality classes. To apply the MP theory, at finite sizes, to matrices with elements drawn from a Heavy-Tailed distribution of the form given in Eqn. (2), we have one of the following three Universality classes.

- **(Weakly) Heavy-Tailed**, $4 < \mu$: Here, the ESD $\rho_N(\lambda)$ exhibits "vanilla" MP behavior in the infinite limit, and the expected mean value of the bulk edge is $\lambda^+ \sim M^{-2/3}$. Unlike standard MP theory, which exhibits TW statistics at the bulk edge, here the edge exhibits PL / Heavy-Tailed fluctuations at finite $N$. These finite-size effects appear in the edge / tail of the ESD, and they make it hard or impossible to distinguish the edge versus the tail at finite $N$.
- **(Moderately) Heavy-Tailed**, $2 < \mu < 4$: Here, the ESD $\rho_N(\lambda)$ is Heavy-Tailed / PL in the infinite limit, approaching $\rho(\lambda) \sim \lambda^{-1-\mu/2}$. In this regime, there is no bulk edge. At finite size, the global ESD can be modeled by $\rho_N(\lambda) \sim \lambda^{-(a\mu+b)}$, for all $\lambda > \lambda_{min}$, but the slope $a$ and intercept $b$ must be fit, as they display large finite-size effects. The maximum eigenvalues follow Frechet (not TW) statistics, with $\lambda_{max} \sim M^{4/\mu-1}(1/Q)^{1-2/\mu}$, and they have large finite-size effects. Thus, at any finite $N$, $\rho_N(\lambda)$ is Heavy-Tailed, but the tail decays moderately quickly.
- **(Very) Heavy-Tailed**, $0 < \mu < 2$: Here, the ESD $\rho_N(\lambda)$ is Heavy-Tailed / PL for all finite $N$, and as $N \to \infty$ it converges more quickly to a PL distribution with tails $\rho(\lambda) \sim \lambda^{-1-\mu/2}$. In this regime, there is no bulk edge, and the maximum eigenvalues follow Frechet (not TW) statistics. Finite-size effects exist, but they are are much smaller here than in the $2 < \mu < 4$ regime of $\mu$.

**Fitting PL distributions to ESD plots.** Once we have identified PL distributions visually, we can fit the ESD to a PL in order to obtain the exponent $\alpha$. We use the Clauset-Shalizi-Newman (CSN) approach (38), as implemented in the python PowerLaw package (39),[1]. Fitting a PL has many subtleties, most beyond the scope of this paper (38; 40; 41; 42; 43; 44; 39; 45; 46).

---

[1]See https://github.com/jeffalstott/powerlaw.

**Identifying the Universality class.** Given $\alpha$, we identify the corresponding $\mu$ and thus which of the three Heavy-Tailed Universality classes ($0 < \mu < 2$ or $2 < \mu < 4$ or $4 < \mu$, as described in Table 1) is appropriate to describe the system. The following are particularly important points. First, observing a Heavy-Tailed ESD may indicate the presence of a scale-free DNN. This suggests that the underlying DNN is strongly-correlated, and that we need more than just a few separated spikes, plus some random-like bulk structure, to model the DNN and to understand DNN regularization. Second, this does not necessarily imply that the matrix elements of $\mathbf{W}_l$ form a Heavy-Tailed distribution. Rather, the Heavy-Tailed distribution arises since we posit it as a model of the strongly correlated, highly non-random matrix $\mathbf{W}_l$. Third, we conjecture that this is more general, and that very well-trained DNNs will exhibit Heavy-Tailed behavior in their ESD for many the weight matrices.

## 3 EMPIRICAL RESULTS: ESDS FOR EXISTING, PRETRAINED DNNS

In this section, we describe our main empirical results for existing, pretrained DNNs. Early on, we observed that small DNNs and large DNNs have very different ESDs. For smaller models, ESDs tend to fit the MP theory well, with well-understood deviations, e.g., low-rank perturbations. For larger models, the ESDs $\rho_N(\lambda)$ almost never fit the theoretical $\rho_{mp}(\lambda)$, and they frequently have a completely different form. We use RMT to compare and contrast the ESDs of a smaller, older NN and many larger, modern DNNs. For the small model, we retrain a modern variant of one of the very early and well-known Convolutional Nets—LeNet5. For the larger, modern models, we examine selected layers from AlexNet, InceptionV3, and many other models (as distributed with pyTorch).

**Example: LeNet5 (1998).** LeNet5 is the prototype early model for DNNs (2). Since LeNet5 is older, we actually recoded and retrained it. We used Keras 2.0, using 20 epochs of the AdaDelta optimizer, on the MNIST data set. This model has $100.00\%$ training accuracy, and $99.25\%$ test accuracy on the default MNIST split. We analyze the ESD of the FC1 Layer. The FC1 matrix $\mathbf{W}_{FC1}$ is a $2450 \times 500$ matrix, with $Q = 4.9$, and thus it yields 500 eigenvalues.

Figures 1(a) and 1(b) present the ESD for FC1 of LeNet5, with Figure 1(a) showing the full ESD and Figure 1(b) zoomed-in along the X-axis. We show (red curve) our fit to the MP distribution $\rho_{emp}(\lambda)$. Several things are striking. First, the *bulk* of the density $\rho_{emp}(\lambda)$ has a large, MP-like shape for eigenvalues $\lambda < \lambda^+ \approx 3.5$, and the MP distribution fits this part of the ESD *very* well, including the fact that the ESD just below the best fit $\lambda^+$ is concave. Second, *some eigenvalue mass is bleeding out* from the MP bulk for $\lambda \in [3.5, 5]$, although it is quite small. Third, beyond the MP bulk and this bleeding out region, are several *clear outliers, or spikes*, ranging from $\approx 5$ to $\lambda_{max} \lesssim 25$. Overall, the shape of $\rho_{emp}(\lambda)$, the quality of the global bulk fit, and the statistics and crisp shape of the local bulk edge all agree well with MP theory augmented with a low-rank perturbation.

**Example: AlexNet (2012).** AlexNet was the first modern DNN (47). AlexNet resembles a scaled-up version of the LeNet5 architecture; it consists of 5 layers, 2 convolutional, followed by 3 FC layers (the last being a softmax classifier). We refer to the last 2 layers before the final softmax as layers FC1 and FC2, respectively. FC2 has a $4096 \times 1000$ matrix, with $Q = 4.096$.

Consider AlexNet FC2 (full in Figures 1(c), and zoomed-in in 1(d)). This ESD differs even more profoundly from standard MP theory. Here, we could find no good MP fit. The best MP fit (in red) does not fit the Bulk part of $\rho_{emp}(\lambda)$ well. The fit suggests there should be significantly more bulk eigenvalue mass (i.e., larger empirical variance) than actually observed. In addition, the bulk edge is indeterminate by inspection. It is only defined by the crude fit we present, and any edge statistics obviously do not exhibit TW behavior. In contrast with MP curves, which are convex near the bulk edge, the entire ESD is concave (nearly) everywhere. Here, a PL fit gives good fit $\alpha \approx 2.25$, indicating a $\mu \lesssim 3$. For this layer (and others), the shape of $\rho_{emp}(\lambda)$, the quality of the global bulk fit, and the statistics and shape of the local bulk edge are poorly-described by standard MP theory.

**Empirical results for other pre-trained DNNs.** We have also examined the properties of a wide range of other pre-trained models, and we have observed similar Heavy-Tailed properties to AlexNet in all of the larger, state-of-the-art DNNs, including VGG16, VGG19, ResNet50, InceptionV3, etc. Space constraints prevent a full presentation of these results, but several observations can be made. First, all of our fits, except for certain layers in InceptionV3, appear to be in the range $1.5 < \alpha \lesssim 3.5$ (where the CSN method is known to perform well). Second, we also check to see whether PL is the best fit by comparing the distribution to a Truncated Power Law (TPL), as well as an exponential, stretch-exponential, and log normal distributions. In all cases, we find either a PL or TPL fits best (with a p-value $\leq 0.05$), with TPL being more common for smaller values of $\alpha$. Third, even when

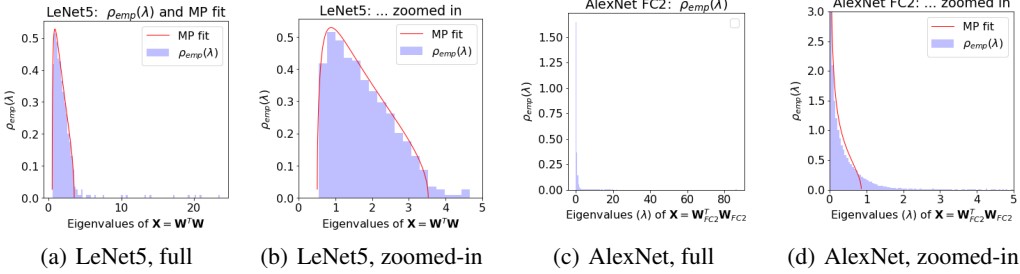

Figure 1: Full and zoomed-in ESD for LeNet5 (Layer FC1) and AlexNet (Layer FC2). Overlaid (in red) are fits of the MP distribution (which fit the bulk very well for LeNet5 but *not* well for AlexNet).

taking into account the large finite-size effects in the range $2 < \alpha < 4$, nearly all of the ESDs appear to fall into the $2 < \mu < 4$ Universality class.

**Towards a Theory of Self-Regularization.** For older and/or smaller models, like LeNet5, the *bulk* of their ESDs ($\rho_N(\lambda)$; $\lambda \ll \lambda^+$) can be well-fit to theoretical MP density $\rho_{mp}(\lambda)$, potentially with distinct, outlying *spikes* ($\lambda > \lambda^+$). This is consistent with the Spiked-Covariance model of Johnstone (33), a simple perturbative extension of the standard MP theory. This is also reminiscent of traditional Tikhonov regularization, in that there is a "size scale" ($\lambda^+$) separating signal (spikes) from noise (bulk). This demonstrates that the DNN training process itself engineers a form of implicit *Self-Regularization* into the trained model.

For large, deep, state-of-the-art DNNs, our observations suggest that there are profound deviations from traditional RMT. These networks are reminiscent of strongly-correlated disordered-systems that exhibit Heavy-Tailed behavior. What is this regularization, and how is it related to our observations of implicit Tikhonov-like regularization on LeNet5?

To answer this, recall that similar behavior arises in strongly-correlated physical systems, where it is known that strongly-correlated systems can be *modeled* by random matrices—with entries drawn from non-Gaussian Universality classes (32), e.g., PL or other Heavy-Tailed distributions. Thus, when we observe that $\rho_N(\lambda)$ has Heavy-Tailed properties, we can hypothesize that $\mathbf{W}$ is strongly-correlated,[2] and we can model it with a Heavy-Tailed distribution. Then, upon closer inspection, we find that the ESDs of large, modern DNNs behave as expected—when using the lens of Heavy-Tailed variants of RMT. Importantly, unlike the Spiked-Covariance case, which has a scale cut-off ($\lambda^+$), in these very strongly Heavy-Tailed cases, correlations appear on every size scale, and we can not find a clean separation between the MP bulk and the spikes. These observations demonstrate that modern, state-of-the-art DNNs exhibit a new form of *Heavy-Tailed Self-Regularization*.

## 4  5+1 PHASES OF REGULARIZED TRAINING

In this section, we develop an operational/phenomenological theory for DNN Self-Regularization.

**MP Soft Rank.** We first define the *MP Soft Rank* ($\mathcal{R}_{mp}$), that is designed to capture the "size scale" of the noise part of $\mathbf{W}_l$, relative to the largest eigenvalue of $\mathbf{W}_l^T \mathbf{W}_l$. Assume that MP theory fits *at least a bulk* of $\rho_N(\lambda)$. Then, we can identify a bulk edge $\lambda^+$ and a bulk variance $\sigma_{bulk}^2$, and define the *MP Soft Rank* as the ratio of $\lambda^+$ and $\lambda_{max}$: $\mathcal{R}_{mp}(\mathbf{W}) := \lambda^+/\lambda_{max}$. Clearly, $\mathcal{R}_{mp} \in [0, 1]$; $\mathcal{R}_{mp} = 1$ for a purely random matrix; and for a matrix with an ESD with outlying spikes, $\lambda_{max} > \lambda^+$, and $\mathcal{R}_{mp} < 1$. If there is no good MP fit because the entire ESD is well-approximated by a Heavy-Tailed distribution, then we can define $\lambda^+ = 0$, in which case $\mathcal{R}_{mp} = 0$.

**Visual Taxonomy.** We characterize *implicit Self-Regularization*, both for DNNs during SGD training as well as for *pre-trained* DNNs, as a visual taxonomy of *5+1 Phases of Training* (RANDOM-LIKE, BLEEDING-OUT, BULK+SPIKES, BULK-DECAY, HEAVY-TAILED, and RANK-COLLAPSE). See Table 2 for a summary. The 5+1 phases can be ordered, with each successive phase corresponding to a smaller Stable Rank / MP Soft Rank and to progressively more Self-Regularization

---

[2]For DNNs, these correlations arise in the weight matrices during Backprop training (at least when training on data of reasonable-quality). That is, the weight matrices "learn" the correlations in the data.

| | Operational Definition | Informal Description via Eqn. (3) | Edge/tail Fluctuation Comments | Illustration and Description |
|---|---|---|---|---|
| RANDOM-LIKE | ESD well-fit by MP with appropriate $\lambda^+$ | $\mathbf{W}^{rand}$ random; $\|\Delta^{sig}\|$ zero or small | $\lambda_{max} \approx \lambda^+$ is sharp, with TW statistics | Fig. 2(a) |
| BLEEDING-OUT | ESD RANDOM-LIKE, excluding eigenmass just above $\lambda^+$ | $\mathbf{W}$ has eigenmass at bulk edge as spikes "pull out"; $\|\Delta^{sig}\|$ medium | BPP transition, $\lambda_{max}$ and $\lambda^+$ separate | Fig. 2(b) |
| BULK+SPIKES | ESD RANDOM-LIKE plus $\geq 1$ spikes well above $\lambda^+$ | $\mathbf{W}^{rand}$ well-separated from low-rank $\Delta^{sig}$; $\|\Delta^{sig}\|$ larger | $\lambda^+$ is TW, $\lambda_{max}$ is Gaussian | Fig. 2(c) |
| BULK-DECAY | ESD less RANDOM-LIKE; Heavy-Tailed eigenmass above $\lambda^+$; some spikes | Complex $\Delta^{sig}$ with correlations that don't fully enter spike | Edge above $\lambda^+$ is not concave | Fig. 2(d) |
| HEAVY-TAILED | ESD better-described by Heavy-Tailed RMT than Gaussian RMT | $\mathbf{W}^{rand}$ is small; $\Delta^{sig}$ is large and strongly-correlated | No good $\lambda^+$; $\lambda_{max} \gg \lambda^+$ | Fig. 2(e) |
| RANK-COLLAPSE | ESD has large-mass spike at $\lambda = 0$ | $\mathbf{W}$ very rank-deficient; over-regularization | — | Fig. 2(f) |

Table 2: The 5+1 phases of learning we identified in DNN training. We observed BULK+SPIKES and HEAVY-TAILED in existing trained models (LeNet5 and AlexNet/InceptionV3, respectively; see Section 3); and we exhibited all 5+1 phases in a simple model (MiniAlexNet; see Section 6).

than previous phases. Figure 2 depicts typical ESDs for each phase, with the MP fits (in red). Earlier phases of training correspond to the final state of older and/or smaller models like LeNet5 and MLP3. Later phases correspond to the final state of more modern models like AlexNet, Inception, etc. While we can describe this in terms of SGD training, this taxonomy allows us to compare different architectures and/or amounts of regularization in a trained—or even pre-trained—DNN.

Each phase is visually distinct, and each has a natural interpretation in terms of RMT. One consideration is the *global properties of the ESD*: how well all or part of the ESD is fit by an MP distriution, for some value of $\lambda^+$, or how well all or part of the ESD is fit by a Heavy-Tailed or PL distribution, for some value of a PL parameter. A second consideration is *local properties of the ESD*: the form of fluctuations, in particular around the edge $\lambda^+$ or around the largest eigenvalue $\lambda_{max}$. For example, the shape of the ESD near to and immediately above $\lambda^+$ is very different in Figure 2(a) and Figure 2(c) (where there is a crisp edge) versus Figure 2(b) (where the ESD is concave) versus Figure 2(d) (where the ESD is convex).

**Theory of Each Phase.** RMT provides more than simple visual insights, and we can use RMT to differentiate between the *5+1 Phases of Training* using simple models that qualitatively describe the shape of each ESD. We model the weight matrices $\mathbf{W}$ as "noise plus signal," where the "noise" is modeled by a random matrix $\mathbf{W}^{rand}$, with entries drawn from the Gaussian Universality class (well-described by traditional MP theory) and the "signal" is a (small or large) correction $\Delta^{sig}$:

$$\mathbf{W} \simeq \mathbf{W}^{rand} + \Delta^{sig}. \tag{3}$$

Table 2 summarizes the theoretical model for each phase. Each model uses RMT to describe the global shape of $\rho_N(\lambda)$, the local shape of the fluctuations at the bulk edge, and the statistics and information in the outlying spikes, including possible Heavy-Tailed behaviors.

In the first phase (RANDOM-LIKE), the ESD is well-described by traditional MP theory, in which a random matrix has entries drawn from the Gaussian Universality class. In the next phases (BLEEDING-OUT, BULK+SPIKES), and/or for small networks such as LetNet5, $\Delta$ is a relatively-small perturbative correction to $\mathbf{W}^{rand}$, and vanilla MP theory (as reviewed in Section 2.1) can be applied, as least to the bulk of the ESD. In these phases, we will *model* the $\mathbf{W}^{rand}$ matrix by a vanilla $\mathbf{W}_{mp}$ matrix (for appropriate parameters), and the MP Soft Rank is relatively large ($\mathcal{R}_{mp}(\mathbf{W}) \gg 0$). In the BULK+SPIKES phase, the model resembles a Spiked-Covariance model, and the Self-Regularization resembles Tikhonov regularization.

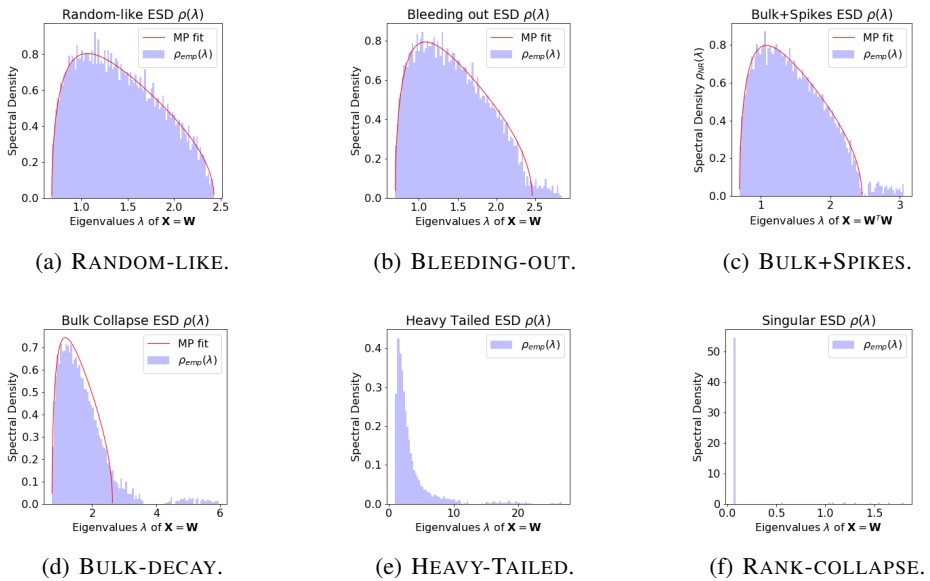

Figure 2: Taxonomy of trained models. Starting off with an initial random or RANDOM-LIKE model (2(a)), training can lead to a BULK+SPIKES model (2(c)), with data-dependent spikes on top of a random-like bulk. Depending on the network size and architecture, properties of training data, etc., additional training can lead to a HEAVY-TAILED model (2(e)), a high-quality model with long-range correlations. An intermediate BLEEDING-OUT model (2(b)), where spikes start to pull out from the bulk, and an intermediate BULK-DECAY model (2(d)), where correlations start to degrade the separation between the bulk and spikes, leading to a decay of the bulk, are also possible. In extreme cases, a severely over-regularized model (2(f)) is possible.

In later phases (BULK-DECAY, HEAVY-TAILED), and/or for modern DNNs such as AlexNet and InceptionV3, $\Delta$ becomes more complex and increasingly dominates over $\mathbf{W}^{rand}$. For these more strongly-correlated phases, $\mathbf{W}^{rand}$ is relatively much weaker, and the MP Soft Rank decreases. Vanilla MP theory is not appropriate, and instead the Self-Regularization becomes Heavy-Tailed. We will treat the noise term $\mathbf{W}^{rand}$ as small, and we will *model* the properties of $\Delta$ with Heavy-Tailed extensions of vanilla MP theory (as reviewed in Section 2.2) to Heavy-Tailed non-Gaussian universality classes that are more appropriate to model strongly-correlated systems. In these phases, the strongly-correlated model is still regularized, but in a very non-traditional way. The final phase, the RANK-COLLAPSE phase, is a degenerate case that is a prediction of the theory.

## 5 EMPIRICAL RESULTS: DETAILED ANALYSIS ON SMALLER MODELS

To validate and illustrate our theory, we analyzed MiniAlexNet,[3] a simpler version of AlexNet, similar to the smaller models used in (9), scaled down to prevent overtraining, and trained on CIFAR10. Space constraints prevent a full presentation of these results, but we mention a few key results here. The basic architecture consists of two 2D Convolutional layers, each with Max Pooling and Batch Normalization, giving 6 initial layers; it then has two Fully Connected (FC), or Dense, layers with ReLU activations; and it then has a final FC layer added, with 10 nodes and softmax activation. $\mathbf{W}_{FC1}$ is a $4096 \times 384$ matrix ($Q \approx 10.67$); $\mathbf{W}_{FC2}$ is a $384 \times 192$ matrix ($Q = 2$); and $\mathbf{W}_{FC3}$ is a $192 \times 10$ matrix. All models are trained using Keras 2.x, with TensorFlow as a backend. We use SGD with momentum, with a learning rate of $0.01$, a momentum parameter of $0.9$, and a baseline batch size of $32$; and we train up to $100$ epochs. We save the weight matrices at the end of every epoch, and we analyze the empirical properties of the $\mathbf{W}_{FC1}$ and $\mathbf{W}_{FC2}$ matrices.

For each layer, the matrix Entropy ($\mathcal{S}(\mathbf{W})$) gradually lowers; and the Stable Rank ($\mathcal{R}_s(\mathbf{W})$) shrinks. These decreases parallel the increase in training/test accuracies, and both metrics level off as the

---

[3] https://github.com/deepmind/sonnet/blob/master/sonnet/python/modules/nets/alexnet.py

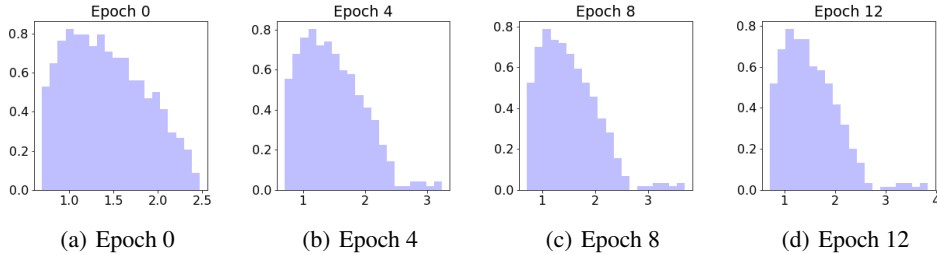

Figure 3: Baseline ESD for Layer FC1 of MiniAlexNet, during training.

training/test accuracies do. These changes are seen in the ESD, e.g., see Figure 3. For layer FC1, the initial weight matrix $\mathbf{W}^0$ looks very much like an MP distribution (with $Q \approx 10.67$), consistent with a RANDOM-LIKE phase. Within a very few epochs, however, eigenvalue mass shifts to larger values, and the ESD looks like the BULK+SPIKES phase. Once the Spike(s) appear(s), substantial changes are hard to see visually, but minor changes do continue in the ESD. Most notably, $\lambda^{max}$ increases from roughly $3.0$ to roughly $4.0$ during training, indicating further Self-Regularization, even within the BULK+SPIKES phase. Here, spike eigenvectors tend to be more localized than bulk eigenvectors. If explicit regularization (e.g., $L_2$ norm weight regularization or Dropout) is added, then we observe a greater decrease in the complexity metrics (Entropies and Stable Ranks), consistent with expectations, and this is casued by the eigenvalues in the spike being pulled to much larger values in the ESD. We also observe that eigenvector localization tends to be more prominent, presumably since explicit regularization can make spikes more well-separated from the bulk.

## 6 EXPLAINING THE GENERALIZATION GAP BY EXHIBITING THE PHASES

In this section, we demonstrate that we can exhibit all five of the main phases of learning by changing a single knob of the learning process. We consider the batch size since it is not traditionally considered a regularization parameter and due to its its implications for the generalization gap.

The *Generalization Gap* refers to the peculiar phenomena that DNNs generalize significantly less well when trained with larger mini-batches (on the order of $10^3 - 10^4$) (48; 12; 13; 14). Practically, this is of interest since smaller batch sizes makes training large DNNs on modern GPUs much less efficient. Theoretically, this is of interest since it contradicts simplistic stochastic optimization theory for convex problems. Thus, there is interest in the question: what is the mechanism responsible for the drop in generalization in models trained with SGD methods in the large-batch regime?

To address this question, we consider here using different batch sizes in the DNN training algorithm. We trained the MiniAlexNet model, just as in Section 5, except with batch sizes ranging from moderately large to very small ($b \in \{500, 250, 100, 50, 32, 16, 8, 4, 2\}$).

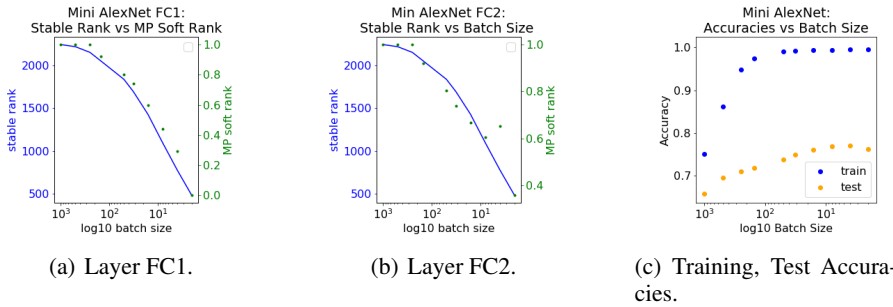

(a) Layer FC1.          (b) Layer FC2.          (c) Training, Test Accuracies.

Figure 4: Varying Batch Size. Stable Rank and MP Softrank for FC1 (4(a)) and FC2 (4(b)); and Training and Test Accuracies (4(c)) versus Batch Size for MiniAlexNet.

**Stable Rank, MP Soft Rank, and Training/Test Performance.** Figure 4 shows the Stable Rank and MP Softrank for FC1 (4(a)) and FC2 (4(b)) as well as the Training and Test Accuracies (4(c))

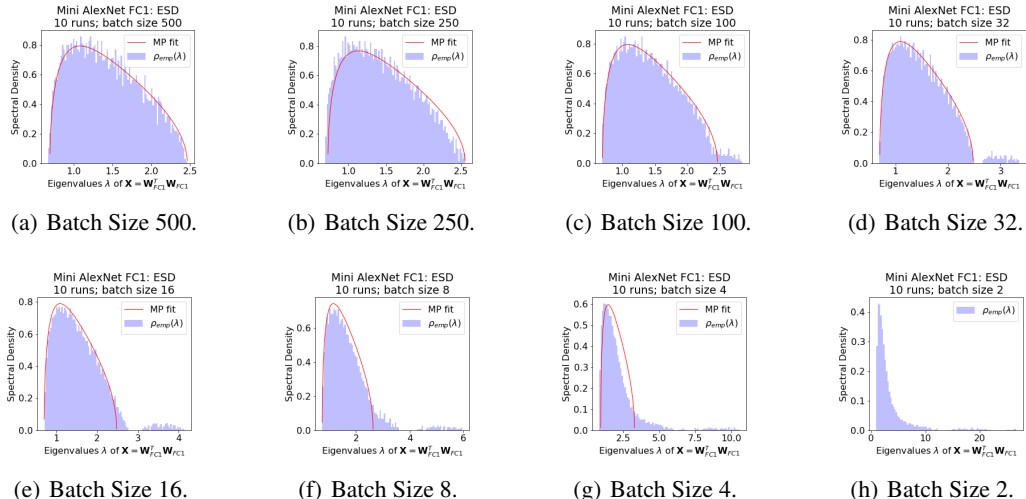

Figure 5: Varying Batch Size. ESD for Layer FC1 of MiniAlexNet, with MP fit (in red), for an ensemble of 10 runs, for Batch Size ranging from 500 down to 2. Smaller batch size leads to more implicitly self-regularized models. We exhibit all 5 of the main phases of training by varying only the batch size.

as a function of Batch Size. The MP Soft Rank ($\mathcal{R}_{mp}$) and the Stable Rank ($\mathcal{R}_s$) both track each other, and both systematically *decrease* with decreasing batch size, as the test accuracy *increases*. In addition, both the training and test accuracy decrease for larger values of $b$: training accuracy is roughly flat until batch size $b \approx 100$, and then it begins to decrease; and test accuracy actually increases for extremely small $b$, and then it gradually decreases as $b$ increases.

**ESDs: Comparisons with RMT.** Figure 5 shows the final ensemble ESD for each value of $b$ for Layer FC1. We see systematic changes in the ESD as batch size $b$ decreases. At batch size $b = 250$ (and larger), the ESD resembles a pure MP distribution with no outliers/spikes; it is RANDOM-LIKE. As $b$ decreases, there starts to appear an outlier region. For $b = 100$, the outlier region resembles BLEEDING-OUT. For $b = 32$, these eigenvectors become well-separated from the bulk, and the ESD resembles BULK+SPIKES. As batch size continues to decrease, the spikes grow larger and spread out more (observe the scale of the X-axis), and the ESD exhibits BULK-DECAY. Finally, at $b = 2$, extra mass from the main part of the ESD plot almost touches the spike, and the curvature of the ESD changes, consistent with HEAVY-TAILED. In addition, as $b$ decreases, some of the extreme eigenvectors associated with eigenvalues that are not in the bulk tend to be more localized.

**Implications for the generalization gap.** Our results here (both that training/test accuracies decrease for larger batch sizes and that smaller batch sizes lead to more well-regularized models) demonstrate that the generalization gap phenomenon arises since, for smaller values of the batch size $b$, the DNN training process itself implicitly leads to stronger Self-Regularization. (This Self-Regularization can be either the more traditional Tikhonov-like regularization or the Heavy-Tailed Self-Regularization corresponding to strongly-correlated models.) That is, training with smaller batch sizes implicitly leads to more well-regularized models, and it is this regularization that leads to improved results. The obvious mechanism is that, by training with smaller batches, the DNN training process is able to "squeeze out" more and more finer-scale correlations from the data, leading to more strongly-correlated models. Large batches, involving averages over many more data points, simply fail to see this very fine-scale structure, and thus they are less able to construct strongly-correlated models characteristic of the HEAVY-TAILED phase.

## 7  DISCUSSION AND CONCLUSION

Clearly, our theory opens the door to address numerous very practical questions. One of the most obvious is whether our RMT-based theory is applicable to other types of layers such as convolutional layers. Initial results suggest yes, but the situation is more complex than the relatively simple picture we have described here. These and related directions are promising avenues to explore.

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

# Implicit Self-Regularization in Deep Neural Networks: Evidence from Random Matrix Theory and Implications for Learning

Authors anonymized for ICLR Supplementary Material

## Abstract

Random Matrix Theory (RMT) is applied to analyze the weight matrices of Deep Neural Networks (DNNs), including both production quality, pre-trained models such as AlexNet and Inception, and smaller models trained from scratch, such as LeNet5 and a miniature-AlexNet. Empirical and theoretical results clearly indicate that the DNN training process itself implicitly implements a form of *Self-Regularization*, implicitly sculpting a more regularized energy or penalty landscape. In particular, the empirical spectral density (ESD) of DNN layer matrices displays signatures of traditionally-regularized statistical models, even in the absence of exogenously specifying traditional forms of explicit regularization, such as Dropout or Weight Norm constraints. Building on relatively recent results in RMT, most notably its extension to Universality classes of Heavy-Tailed matrices, and applying them to these empirical results, we develop a theory to identify *5+1 Phases of Training*, corresponding to increasing amounts of *Implicit Self-Regularization*. These phases can be observed during the training process as well as in the final learned DNNs. For smaller and/or older DNNs, this Implicit Self-Regularization is like traditional Tikhonov regularization, in that there is a "size scale" separating signal from noise. For state-of-the-art DNNs, however, we identify a novel form of *Heavy-Tailed Self-Regularization*, similar to the self-organization seen in the statistical physics of disordered systems (such as classical models of actual neural activity). This results from correlations arising at all size scales, which for DNNs arises implicitly due to the training process itself. This implicit Self-Regularization can depend strongly on the many knobs of the training process. In particular, by exploiting the generalization gap phenomena, we demonstrate that we can cause a small model to exhibit all 5+1 phases of training simply by changing the batch size. This demonstrates that—all else being equal—DNN optimization with larger batch sizes leads to less-well implicitly-regularized models, and it provides an explanation for the generalization gap phenomena. Our results suggest that large, well-trained DNN architectures should exhibit Heavy-Tailed Self-Regularization, and we discuss the theoretical and practical implications of this.

# Contents

# 1 Introduction

Very large very deep neural networks (DNNs) have received attention as a general purpose tool for solving problems in machine learning (ML) and artificial intelligence (AI), and they perform remarkably well on a wide range of traditionally hard if not impossible problems, such as speech recognition, computer vision, and natural language processing. The conventional wisdom seems to be "the bigger the better," "the deeper the better," and "the more hyper-parameters the better." Unfortunately, this usual *modus operandi* leads to large, complicated models that are extremely hard to train, that are extremely sensitive to the parameters settings, and that are extremely difficult to understand, reason about, and interpret. Relatedly, these models seem to violate what one would expect from the large body of theoretical work that is currently popular in ML, optimization, statistics, and related areas. This leads to theoretical results that fail to provide guidance to practice as well as to confusing and conflicting interpretations of empirical results. For example, current optimization theory fails to explain phenomena like the so-called *Generalization Gap*—the curious observation that DNNs generalize better when trained with smaller batches sizes—and it often does not provide even qualitative guidance as to how stochastic algorithms perform on non-convex landscapes of interest; and current statistical learning theory, e.g., VC-based methods, fails to provide even qualitative guidance as to the behavior of this class of learning methods that seems to have next to unlimited capacity and yet generalize without overtraining.

## 1.1 A historical perspective

The inability of optimization and learning theory to explain and predict the properties of NNs is not a new phenomenon. From the earliest days of DNNs, it was suspected that VC theory did not apply to these systems. For example, in 1994, Vapnik, Levin, and LeCun [144] said:

> [T]he [VC] theory is derived for methods that minimize the empirical risk. However, existing learning algorithms for multilayer nets cannot be viewed as minimizing the empirical risk over [the] entire set of functions implementable by the network.

It was originally assumed that local minima in the energy/loss surface were responsible for the inability of VC theory to describe NNs [144], and that the mechanism for this was that getting trapped in local minima during training limited the number of possible functions realizable by the network. However, it was very soon realized that the presence of local minima in the energy function was *not* a problem in practice [79, 39]. (More recently, this fact seems to have been rediscovered [108, 37, 56, 135].) Thus, another reason for the inapplicability of VC theory was needed. At the time, there did exist other theories of generalization based on statistical mechanics [127, 147, 60, 43], but for various technical and nontechnical reasons these fell out of favor in the ML/NN communities. Instead, VC theory and related techniques continued to remain popular, in spite of their obvious problems.

More recently, theoretical results of Choromanska et al. [30] (which are related to [127, 147, 60, 43]) suggested that the Energy/optimization Landscape of modern DNNs resembles the Energy Landscape of a zero-temperature *Gaussian* Spin Glass; and empirical results of Zhang et al. [156] have again pointed out that VC theory does not describe the properties of DNNs. Motivated by these results, Martin and Mahoney then suggested that the Spin Glass analogy may be useful to understand severe overtraining versus the inability to overtrain in modern DNNs [93].

Many puzzling questions about regularization and optimization in DNNs abound. In fact, it is not even clear how to define DNN regularization. In traditional ML, regularization can be either explicit or implicit. Let's say that we are optimizing some loss function $L(\cdot)$, specified by some

parameter vector or weight matrix $W$. When regularization is explicit, it involves making the loss function $L$ "nicer" or "smoother" or "more well-defined" by adding an explicit capacity control term directly to the loss, i.e., by considering a modified objective of the form $L(W) + \alpha \|W\|$. In this case, we tune the regularization parameter $\alpha$ by cross validation. When regularization is implicit, we instead have some adjustable operational procedure like early stopping of an iterative algorithm or truncating small entries of a solution vector. In many cases, we can still relate this back to the more familiar form of optimizing an effective function of the form $L(W) + \alpha \|W\|$. For a precise statement in simple settings, see [89, 115, 53]; and for a discussion of implicit regularization in a broader context, see [88] and references therein.

With DNNs, the situation is far less clear. The challenge in applying these well-known ideas to DNNs is that DNNs have *many* adjustable "knobs and switches," independent of the Energy Landscape itself, most of which can affect training accuracy, in addition to *many* model parameters. Indeed, nearly anything that improves generalization is called regularization, and a recent review presents a taxonomy over 50 different regularization techniques for Deep Learning [76]. The most common include ML-like Weight Norm regularization, so-called "tricks of the trade" like early stopping and decreasing the batch size, and DNN-specific methods like Batch Normalization and Dropout. Evaluating and comparing these methods is challenging, in part since there are so many, and in part since they are often constrained by systems or other not-traditionally-ML considerations. Moreover, Deep Learning avoids cross validation (since there are simply too many parameters), and instead it simply drives training error to zero (followed by subsequent fiddling of knobs and switches). Of course, it is still the case that test information can leak into the training process (indeed, perhaps even more severely for DNNs than traditional ML methods). Among other things, this argues for unsupervised metrics to evaluate model quality.

Motivated by this situation, we are interested here in two related questions.

- **Theoretical Question.** Why is regularization in deep learning seemingly quite different than regularization in other areas on ML; and what is the right theoretical framework with which to investigate regularization for DNNs?

- **Practical Question.** How can one control and adjust, in a theoretically-principled way, the many knobs and switches that exist in modern DNN systems, e.g., to train these models efficiently and effectively, to monitor their effects on the global Energy Landscape, etc.?

That is, we seek a *Practical Theory of Deep Learning*, one that is prescriptive and not just descriptive. This theory would provide useful tools for practitioners wanting to know *How* to characterize and control the Energy Landscape to engineer larger and betters DNNs; and it would also provide theoretical answers to broad open questions as *Why* Deep Learning even works. For example, it would provide metrics to characterize qualitatively-different classes of learning behaviors, as predicted in recent work [93]. Importantly, VC theory and related methods do *not* provide a theory of this form.

## 1.2   Overview of our approach

Let us write the Energy Landscape (or optimization function) for a typical DNN with $L$ layers, with activation functions $h_l(\cdot)$, and with weight matrices and biases $\mathbf{W}_l$ and $\mathbf{b}_l$, as follows:

$$E_{DNN} = h_L(\mathbf{W}_L \times h_{L-1}(\mathbf{W}_{L-1} \times h_{L-2}(\cdots) + \mathbf{b}_{L-1}) + \mathbf{b}_L). \tag{1}$$

For simplicity, we do not indicate the structural details of the layers (e.g., Dense or not, Convolutions or not, Residual/Skip Connections, etc.). We imagine training this model on some labeled

data $\{d_i, y_i\} \in \mathcal{D}$, using Backprop, by minimizing the loss $\mathcal{L}$ (i.e., the cross-entropy), between $E_{DNN}$ and the labels $y_i$, as follows:

$$\min_{W_l, b_l} \mathcal{L} \left( \sum_i E_{DNN}(d_i) - y_i \right). \tag{2}$$

We can initialize the DNN using random initial weight matrices $\mathbf{W}_l^0$, or we can use other methods such as transfer learning (which we will not consider here). There are various knobs and switches to tune such as the choice of solver, batch size, learning rate, etc.

Most importantly, to avoid overtraining, we must usually regularize our DNN. Perhaps the most familiar approach from ML for implementing this regularization explicitly constrains the norm of the weight matrices, e.g., modifying Objective (2) to give:

$$\min_{W_l, b_l} \mathcal{L} \left( \sum_i E_{DNN}(d_i) - y_i \right) + \alpha \sum_l \|\mathbf{W}_l\|, \tag{3}$$

where $\| \cdot \|$ is some matrix norm, and where $\alpha$ is an explicit regularization control parameter.

The point of Objective (3) is that *explicit* regularization shrinks the norm(s) of the $\mathbf{W}_l$ matrices. We may expect similar results to hold for *implicit* regularization. We will use advanced methods from Random Matrix Theory (RMT), developed in the theory of self organizing systems, to characterize DNN layer weight matrices, $\mathbf{W}_l$,[1] during and after the training process.

Here is an important (but often under-appreciated) point. We call $E_{DNN}$ the *Energy Landscape*. By this, we mean that part of the optimization problem parameterized by the heretofore unknown elements of the weight matrices and bias vectors, for a fixed $\alpha$ (in (3)), and as defined by the data $\{d_i, y_i\} \in \mathcal{D}$. Because we run Backprop training, we pass the data through the Energy function $E_{DNN}$ multiple times. Each time, we adjust the values of the weight matrices and bias vectors. In this sense, we may think of the total Energy Landscape (i.e., the optimization function that is nominally being optimized) as changing at each epoch.

## 1.3   Summary of our results

We analyze the distribution of eigenvalues, i.e., the Empirical Spectral Density (ESD), $\rho_N(\lambda)$, of the correlation matrix $\mathbf{X} = \mathbf{W}^T \mathbf{W}$ associated with the layer weight matrix $\mathbf{W}$. We do this for a wide range of large, pre-trained, readily-available state-of-the-art models, including the original LetNet5 convolutional net (which, due to its age, we retrain) and pre-trained models available in Keras and PyTorch such as AlexNet and Inception. In some cases, the ESDs are very well-described by Marchenko-Pastur (MP) RMT. In other cases, the ESDs are well-described by MP RMT, with the exception of one or more large eigenvalues that can be modeled by a Spiked-Covariance model [92, 68]. In still other cases—including nearly every current state-of-the-art model we have examined—the EDSs are poorly-described by traditional RMT, and instead they are more consistent with Heavy-Tailed behavior seen in the statistical physics of disordered systems [134, 24]. Based on our observations, we develop a develop a practical theory of *Implicit Self-Regularization* in DNNs. This theory takes the form of an operational theory characterizing 5+1 phases of DNN training. To test and validate our theory, we consider two smaller models, a

---

[1]We consider weight matrices $\mathbf{W}_l$ computed for individual dense layers $l$ and other layers that can be easily-represented as matrices, i.e., 2-index tensors. Nothing in our RMT-based theory, however, requires matrices to be constructed from dense layers—they could easily be applied to matrices constructed from convolutional (or other) layers. For example, we could can stack/reshape weight matrices in various ways, e.g., to study multi-dimensional tensors like Convolutional Layers or how different layers interact. We have unpublished results that indicate this is a promising direction, but we don't consider these variants here.

3-layer MLP (MLP3) and a miniature version of AlexNet (MiniAlexNet), trained on CIFAR10, that we can train ourselves repeatedly, adjusting various knobs and switches along the way.

**Main Empirical Results.** Our main empirical results consist in evaluating empirically the ESDs (and related RMT-based statistics) for weight matrices for a suite of DNN models, thereby probing the Energy Landscapes of these DNNs. For older and/or smaller models, these results are consistent with implicit *Self-Regularization* that is Tikhonov-like; and for modern state-of-the-art models, these results suggest novel forms of *Heavy-Tailed Self-Regularization.*

- **Capacity Control Metrics.** We study simple capacity control metrics, the Matrix Entropy, the linear algebraic or Hard Rank, and the Stable Rank. We also use MP RMT to define a new metric, the MP Soft Rank. These metrics track the amount of *Self-Regularization* that arises in a weight matrix $\mathbf{W}$, either during training or in a pre-trained DNN.

- **Self-Regularization in old/small models.** The ESDs of older/smaller DNN models (like LeNet5 and a toy MLP3 model) exhibit weak *Self-Regularization*, well-modeled by a perturbative variant of MP theory, the Spiked-Covariance model. Here, a small number of eigenvalues pull out from the random bulk, and thus the MP Soft Rank and Stable Rank both decrease. This weak form of *Self-Regularization* is like Tikhonov regularization, in that there is a "size scale" that cleanly separates "signal" from "noise," but it is different than explicit Tikhonov regularization in that it arises implicitly due to the DNN training process itself.

- **Heavy-Tailed Self-Regularization.** The ESDs of larger, modern DNN models (including AlexNet and Inception and nearly every other large-scale model we have examined) deviate strongly from the common Gaussian-based MP model. Instead, they appear to lie in one of the very different Universality classes of Heavy-Tailed random matrix models. We call this *Heavy-Tailed Self-Regularization*. Here, the MP Soft Rank vanishes, and the Stable Rank decreases, but the full Hard Rank is still retained. The ESD appears fully (or partially) Heavy-Tailed, but with finite support. In this case, there is *not* a "size scale" (even in the theory) that cleanly separates "signal" from "noise."

**Main Theoretical Results.** Our main theoretical results consist in an operational theory for DNN Self-Regularization. Our theory uses ideas from RMT—both vanilla MP-based RMT as well as extensions to other Universality classes based on Heavy-Tailed distributions—to provide a visual taxonomy for $5 + 1$ *Phases of Training*, corresponding to increasing amounts of Self-Regularization.

- **Modeling Noise and Signal.** We assume that a weight matrix $\mathbf{W}$ can be modeled as $\mathbf{W} \simeq \mathbf{W}^{rand} + \Delta^{sig}$, where $\mathbf{W}^{rand}$ is "noise" and where $\Delta^{sig}$ is "signal." For small to medium sized signal, $\mathbf{W}$ is well-approximated by an MP distribution—with elements drawn from the Gaussian Universality class—perhaps after removing a few eigenvectors. For large and strongly-correlated signal, $\mathbf{W}^{rand}$ gets progressively smaller, but we can model the non-random strongly-correlated signal $\Delta^{sig}$ by a Heavy-Tailed random matrix, i.e., a random matrix with elements drawn from a Heavy-Tailed (rather than Gaussian) Universality class.

- **5+1 Phases of Regularization.** Based on this approach to modeling noise and signal, we construct a practical, visual taxonomy for 5+1 Phases of Training. Each phase is characterized by stronger, visually distinct signatures in the ESD of DNN weight matrices, and successive phases correspond to decreasing MP Soft Rank and increasing amounts of

*Self-Regularization.* The 5+1 phases are: RANDOM-LIKE, BLEEDING-OUT, BULK+SPIKES, BULK-DECAY, HEAVY-TAILED, and RANK-COLLAPSE.

- **Rank-collapse.** One of the predictions of our RMT-based theory is the existence of a pathological phase of training, the RANK-COLLAPSE or "+1" Phase, corresponding to a state of *over-regularization.* Here, one or a few very large eigenvalues dominate the ESD, and the rest of the weight matrix loses nearly all Hard Rank.

Based on these results, we speculate that all well optimized, large DNNs will display *Heavy-Tailed Self-Regularization* in their weight matrices.

**Evaluating the Theory.** We provide a detailed evaluation of our theory using a smaller MiniAlexNew model that we can train and retrain.

- **Effect of Explicit Regularization.** We analyze ESDs of MiniAlexNet by removing all explicit regularization (Dropout, Weight Norm constraints, Batch Normalization, etc.) and characterizing how the ESD of weight matrices behave during and at the end of Backprop training, as we systematically add back in different forms of explicit regularization.

- **Implementation Details.** Since the details of the methods that underlies our theory (e.g., fitting Heavy-Tailed distributions, finite-size effects, etc.) are likely not familiar to ML and NN researchers, and since the details matter, we describe in detail these issues.

- **Exhibiting the 5+1 Phases.** We demonstrate that we can exhibit all 5+1 phases by appropriate modification of the various knobs of the training process. In particular, by decreasing the batch size from 500 to 2, we can make the ESDs of the fully-connected layers of MiniAlexNet vary continuously from RANDOM-LIKE to HEAVY-TAILED, while increasing generalization accuracy along the way. These results illustrate the *Generalization Gap* phenomena [64, 72, 57], and they explain that phenomena as being caused by the implicit Self-Regularization associated with models trained with smaller and smaller batch sizes. By adding extreme Weight Norm regularization, we can also induce the RANK-COLLAPSE phase.

**Main Methodological Contribution.** Our main methodological contribution consists in using empirical observations as well as recent developments in RMT to motivate a practical predictive DNN theory, rather than developing a descriptive DNN theory based on general theoretical considerations. Essentially, we treat the training of different DNNs as if we are running novel laboratory experiments, and we follow the traditional scientific method:

*Make Observations → Form Hypotheses → Build a Theory → Test the theory, literally.*

In particular, this means that we can observe and analyze many large, production-quality, pre-trained models directly, without needing to retrain them, and we can also observe and analyze smaller models during the training process. In adopting this approach, we are interested in both "scientific questions" (e.g., "Why is regularization in deep learning seemingly quite different . . . ?") as well as "engineering questions" (e.g., "How can one control and adjust . . . ?).

To accomplish this, recall that, given an architecture, the Energy Landscape is completely defined by the DNN weight matrices. Since its domain is exponentially large, the Energy Landscape is challenging to study directly. We can, however, analyze the weight matrices, as well as their correlations. (This is analogous to analyzing the expected moments of a complicated

distribution.) In principle, this permits us to analyze both local and global properties of the Energy Landscape, as well as something about the class of functions (e.g., VC class, Universality class, etc.) being learned by the DNN. Since the weight matrices of many DNNs exhibit strong correlations and can be modeled by random matrices with elements drawn from the Universality class of Heavy-Tailed distributions, this severely restricts the class of functions learned. It also connects back to the Energy Landscape since it is known that the Energy Landscape of Heavy-Tailed random matrices is very different than that of Gaussian-like random matrices.

## 1.4 Outline of the paper

In Section 2, we provide a warm-up, including simple capacity metrics and their transitions during Backprop. Then, in Sections 3 and 4, we review background on RMT necessary to understand our experimental methods, and we present our initial experimental results. Based on this, in Section 5, we present our main theory of 5+1 Phases of Training. Then, in Sections 6 and 7, we evaluate our main theory, illustrating the effect of explicit regularization, and demonstrating implications for the generalization gap phenomenon. Finally, in Section 8, we provide a discussion of our results in a broader context. The accompanying code is available at ((link anonymized for ICLR Supplementary Material)). For reference, we provide in Table 1 and Table 2 a summary of acronyms and notation used in the following.

| Acronym | Description |
|---------|-------------|
| DNN | Deep Neural Network |
| ML | Machine Learning |
| SGD | Stochastic Gradient Descent |
| RMT | Random Matrix Theory |
| MP | Marchenko Pastur |
| ESD | Empirical Spectral Density |
| PL | Power Law |
| HT | Heavy-Tailed |
| TW | Tracy Widom (Law) |
| SVD | Singular Value Decomposition |
| FC | Fully Connected (Layer) |
| VC | Vapnik Chrevonikis (Theory) |
| SMTOG | Statistical Mechanics Theory of Generalization |

Table 1: Definitions of acronyms used in the text.

| Notation | Description |
|----------|-------------|
| $\mathbf{W}$ | DNN layer weight matrix of size $N \times M$, with $N \geq M$ |
| $\mathbf{W}_l$ | DNN layer weight matrix for $l^{th}$ layer |
| $\mathbf{W}_l^e$ | DNN layer weight matrix for $l^{th}$ layer at $e^{th}$ epoch |
| $\mathbf{W}^{rand}$ | random rectangular matrix, elements from truncated Normal distribution |
| $\mathbf{W}(\mu)$ | random rectangular matrix, elements from Pareto distribution |
| $\mathbf{X} = (1/N)\mathbf{W}^T\mathbf{W}$ | normalized correlation matrix for layer weight matrix $\mathbf{W}$ |
| $Q = N/M > 0$ | apsect ratio of $\mathbf{W}$ |
| $\nu$ | singular value of $\mathbf{W}$ |
| $\lambda$ | eigenvalue of $\mathbf{X}$ |
| $\lambda_{max}$ | maximum eigenvalue in an ESD |

| | |
|---|---|
| $\lambda^+$ | eigenvalue at edge of MP Bulk |
| $\lambda_k$ | eigenvalue lying outside MP Bulk, $\lambda^+ < \lambda_k \le \lambda_{max}$ |
| $\rho_{emp}(\lambda)$ | actual ESD, from some $\mathbf{W}$ matrix |
| $\rho(\lambda)$ | theoretical ESD, infinite limit |
| $\rho_N(\lambda)$ | theoretical ESD, finite $N$ size |
| $\rho(\nu)$ | theoretical empirical density of singular values, infinite limit |
| $\sigma^2_{mp}$ | elementwise variance of $\mathbf{W}$, used to define MP distribution |
| $\sigma^2_{shuf}$ | elementwise variance of $\mathbf{W}$, as measured after random shuffling |
| $\sigma^2_{bulk}$ | elementwise variance of $\mathbf{W}$, after removing/ignoring all spikes $\lambda_k > \lambda^+$ |
| $\sigma^2_{emp}$ | elementwise variance of $\mathbf{W}$, determined empirically |
| $\mathcal{R}(\mathbf{W})$ | Hard Rank, number of non-zero singular values, Eqn. (5) |
| $\mathcal{S}(\mathbf{W})$ | Matrix Entropy, as defined on $\mathbf{W}$, Eqn. (6) |
| $\mathcal{R}_s(\mathbf{W})$ | Stable Rank, measures decay of singular values, Eqn. (7) |
| $\mathcal{R}_{mp}(\mathbf{W})$ | MP Soft Rank, applied after and depends on MP fit, Eqn. (11) |
| $\mathcal{S}(\mathbf{v})$ | Vector Entropy, as defined on vector $\mathbf{v}$ |
| $\mathcal{L}(\mathbf{v})$ | Localization Ratio, as defined on vector $\mathbf{v}$ |
| $\mathcal{P}(\mathbf{v})$ | Participation Ratio, as defined on vector $\mathbf{v}$ |
| $p(x) \sim x^{-1-\mu}$ | Pareto distribution, parameterized by $\mu$ |
| $p(x) \sim x^{-\alpha}$ | Pareto distribution, parameterized by $\alpha$ |
| $\rho(\lambda) \sim \lambda^{-(\mu/2+1)}$ | theoretical relation, for ESD of $\mathbf{W}(\mu)$, between $\alpha$ and $\mu$ (for $0 < \mu < 4$) |
| $\rho_N(\lambda) \sim \lambda^{-(a\mu+b)}$ | empiricial relation, for ESD of $\mathbf{W}(\mu)$, between $\alpha$ and $\mu$ (for $2 < \mu < 4$) |
| $\Delta\lambda = \|\lambda - \lambda^+\|$ | empirical uncertainty, due to finite-size effects, in theoretical MP bulk edge |
| $\Delta$ | model of perturbations and/or strong correlations in $\mathbf{W}$ |

Table 2: Definitions of notation used in the text.

## 2  Simple Capacity Metrics and Transitions during Backprop

In this section, we describe simple spectral metrics to characterize DNN weight these matrices as well as initial empirical observations on the capacity properties of training DNNs.

### 2.1  Simple capacity control metrics

A DNN is defined by its detailed architecture and the values of the weights and biases at each layer. We seek a simple capacity control metric for a learned DNN model that: is easy to compute both during training and for already-trained models; can describe changes in the gross behavior of weight matrices during the Backprop training process; and can identify the onset of subtle structural changes in the weight matrices.

One possibility is to use the Euclidean distance between the initial weight matrix, $\mathbf{W}_l^0$, and the weight matrix at epoch $e$ of training, $\mathbf{W}_l^e$, i.e., $\Delta(\mathbf{W}_l^e) = \|\mathbf{W}_l^0 - \mathbf{W}_l^e\|_2$. This distance, however, is *not* scale invariant. In particular, during training, and *with regularization turned off*, the weight matrices may shift in scale, gaining or losing Frobenius mass or variance,[2] and this distance metric is sensitive to that change. Indeed, the whole point of a BatchNorm layer is to *try* to prevent this. To start, then, we will consider two scale-invariant measures of capacity control:

---

[2] We use these terms interchangably, i.e., even if the matrices are not mean-centered.

the Matrix Entropy ($\mathcal{S}$), and the Stable Rank ($\mathcal{R}_s$). For an arbitrary matrix $\mathbf{W}$, both of these metrics are defined in terms of its spectrum.

Consider $N \times M$ (real valued) layer weight matrices $\mathbf{W}_l$, where $N \geq M$. Let the Singular Value Decomposition of $\mathbf{W}$ be

$$\mathbf{W} = \mathbf{U}\boldsymbol{\Sigma}\mathbf{V}^T,$$

where $\nu_i = \boldsymbol{\Sigma}_{ii}$ is the $i^{th}$ singular value[3] of $\mathbf{W}$, and let $p_i = \nu_i^2 / \sum_i \nu_i^2$. We also define the associated $M \times M$ (uncentered) correlation matrix

$$\mathbf{X} = \mathbf{X}_l = \frac{1}{N}\mathbf{W}_l^T\mathbf{W}_l, \tag{4}$$

where we sometimes drop the ($l$) subscript for $\mathbf{X}$, and where $\mathbf{X}$ is normalized by $1/N$. We compute the eigenvalues of $\mathbf{X}$,

$$\mathbf{X}\mathbf{v}_i = \lambda_i\mathbf{v}_i,$$

where $\{\lambda_i, i = 1, \dots, M\}$ are the squares of the singular values: $\lambda_i = \nu_i^2$. Given the singular values of $\mathbf{W}$ and/or eigenvalues of $\mathbf{X}$, there are several well-known matrix complexity metrics.

- The *Hard Rank* (or linear algebraic rank),

$$Hard\ Rank: \quad \mathcal{R}(\mathbf{W}) = \sum_i \delta(\nu_i), \tag{5}$$

  is the number of singular values greater than zero, $\nu_i > 0$, to within a numerical cutoff.

- The *Matrix Entropy*,

$$Matrix\ Entropy: \quad \mathcal{S}(\mathbf{W}) = \frac{-1}{\log(R(\mathbf{W}))} \sum_i p_i \log\ p_i, \tag{6}$$

  is also known as the Generalized von-Neumann Matrix Entropy.[4]

- The *Stable Rank*,

$$Stable\ Rank: \quad \mathcal{R}_s(\mathbf{W}) = \frac{\|\mathbf{W}\|_F^2}{\|\mathbf{W}\|_2^2} = \frac{\sum_i \nu_i^2}{\nu_{max}^2} = \frac{\sum_i \lambda_i}{\lambda_{max}}, \tag{7}$$

  the ratio of the Frobenius norm to Spectral norm, is a robust variant of the *Hard Rank*.

We also refer to the Matrix Entropy $\mathcal{S}(\mathbf{X})$ and Stable Rank $\mathcal{R}_s(\mathbf{X})$ of $\mathbf{X}$. By this, we mean the metrics computed with the associated eigenvalues. Note $\mathcal{S}(\mathbf{X}) = \mathcal{S}(\mathbf{W})$ and $\mathcal{R}_s(\mathbf{X}) = \mathcal{R}_s(\mathbf{W})$.

It is known that a random matrix has maximum Entropy, and that lower values for the Entropy correspond to more structure/regularity. If $\mathbf{W}$ is a random matrix, then $\mathcal{S}(\mathbf{W}) = 1$. For example, we initialize our weight matrices with a truncated random matrix $\mathbf{W}^0$, then $S(\mathbf{W}^0) \lesssim 1$. When $\mathbf{W}$ has significant and observable non-random structure, we expect $\mathcal{S}(\mathbf{W}) < 1$. We will see, however, that in practice these differences are quite small, and we would prefer a more discriminative metric. In nearly every case, for *well-trained* DNNs, all the weight matrices retain full Hard Rank $\mathcal{R}$; but the weight matrices do "shrink," in a sense captured by the Stable Rank. Both $\mathcal{S}$ and $\mathcal{R}_s$ measure matrix capacity, and, up to a scale factor, we will see that they exhibit qualitatively similar behavior.

---

[3]We use $\nu_i$ rather than $\sigma_i$ to denote singular values since $\sigma$ denote variance elsewhere.
[4]This resembles the entanglement Entropy, suggested for the analysis of convolutional nets [81].

## 2.2 Empirical results: Capacity transitions while training

We start by illustrating the behavior of two simple complexity metrics during Backprop training on MLP3, a simple 3-layer Multi-Layer Perceptron (MLP), described in Table 4. MLP3 consists of 3 fully connected (FC) / dense layers with 512 nodes and ReLU activation, with a final FC layer with 10 nodes and softmax activation. This gives 4 layer weight matrices of shape $(N \times M)$ and with $Q = N/M$:

$$
\begin{aligned}
\mathbf{W}_1 &= (\cdot \times 512) \\
\mathbf{W}_2 &= (512 \times 512) \quad \text{(Layer FC1)} \quad (Q = 1) \\
\mathbf{W}_3 &= (512 \times 512) \quad \text{(Layer FC2)} \quad (Q = 1) \\
\mathbf{W}_4 &= (512 \times 10).
\end{aligned}
$$

For the training, each $\mathbf{W}_l$ matrix is initialized with a Glorot normalization [54]. The model is trained on CIFAR10, up to 100 epochs, with SGD (learning rate=0.01, momentum=0.9) and with a stopping criteria of 0.0001 on the MSE loss.[5]

Figure 1 presents the layer entropy (in Figure 1(a)) and the stable rank (in Figure 1(b)), plotted as a function of training epoch, for FC1 and FC2. Both metrics decrease during training (note the scales of the Y axes): the stable rank decreases by approximately a factor of two, and the matrix entropy decreases by a small amount, from roughly 0.92 to just below 0.91 (this is for FC2, and there is an even more modest change for FC1). They both track nearly the same changes; and the stable rank is more informative for our purposes; but we will see that the changes to the matrix entropy, while subtle, are significant.

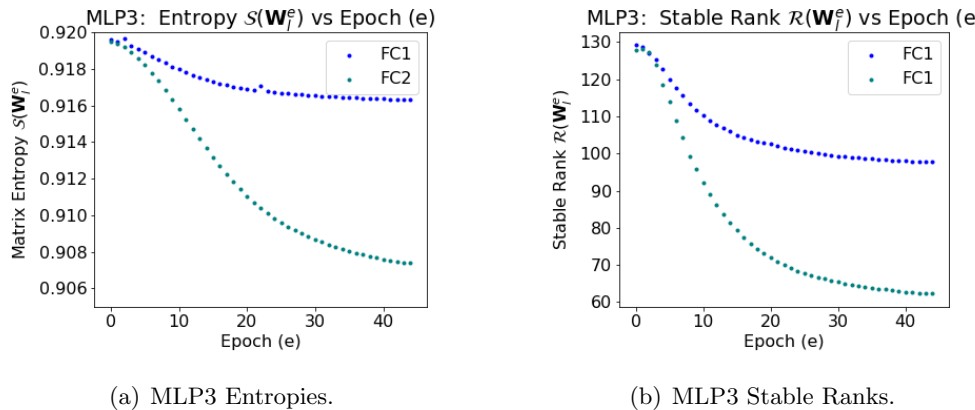

(a) MLP3 Entropies.          (b) MLP3 Stable Ranks.

Figure 1: The behavior of two complexity measures, the Matrix Entropy $\mathcal{S}(\mathbf{W})$ and the Stable Rank $\mathcal{R}_s(\mathbf{W})$, for Layers FC1 and FC2, during Backprop training, for MLP3. Both measures display a transition during Backprop training.

Figure 2 presents scree plots for the initial $\mathbf{W}_l^0$ and final $\mathbf{W}_l$ weight matrices for the FC1 and FC2 layers of our MLP3. A scree plot plots the decreasing variability in the matrix as a function of the increasing index of the corresponding eigenvector [59]. Thus, such scree plots present similar information to the stable rank—e.g., observe the Y-axis of Figure 2(b), which shows that there is a slight increase in the largest eigenvalue for FC1 (again, note the scales of the Y axes) and a larger increase in the largest eigenvalue for FC2, which is consistent with the changes in the stable rank in Figure 1(b))—but they too give a coarse picture of the matrix. In

---

[5]Code will be available at ((link anonymized for ICLR Supplementary Material)).

particular, they lack the detailed insight into subtle changes in the entropy and rank associated with the Self-Regularization process, e.g., changes that reside in just a few singular values and vectors, that we will need in our analysis.

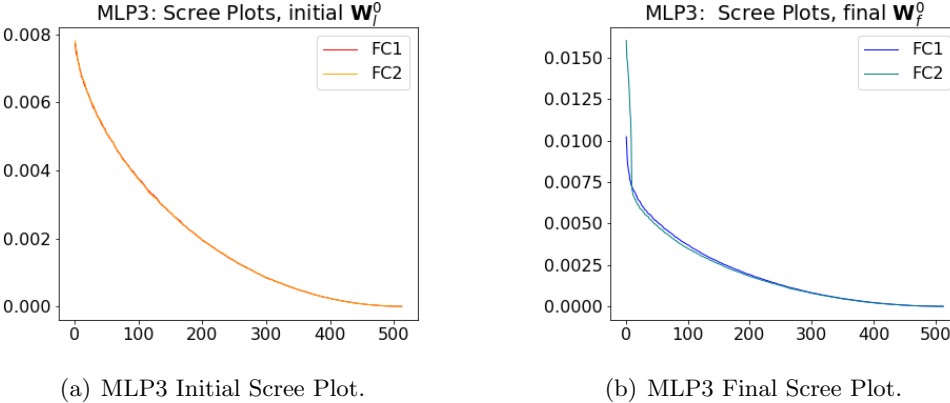

(a) MLP3 Initial Scree Plot.

(b) MLP3 Final Scree Plot.

Figure 2: Scree plots for initial and final configurations for Layers FC1 and FC2, during Backprop training, for MLP3.

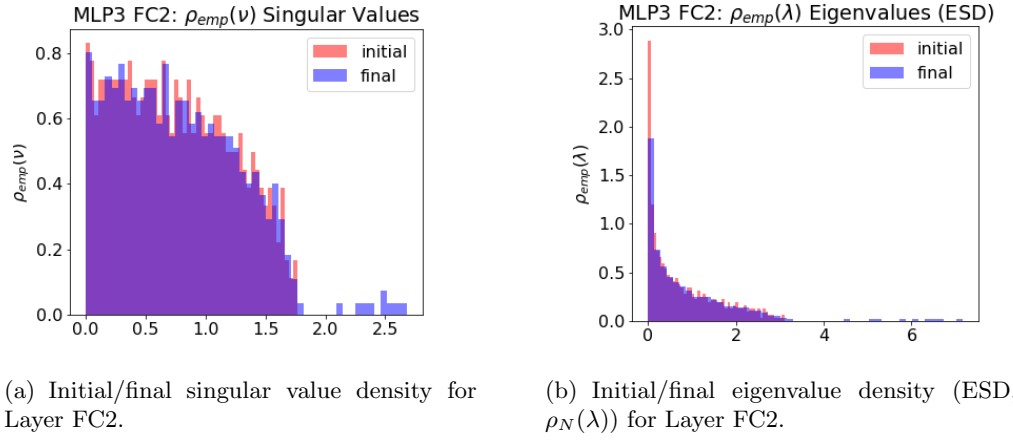

(a) Initial/final singular value density for Layer FC2.

(b) Initial/final eigenvalue density (ESD, $\rho_N(\lambda)$) for Layer FC2.

Figure 3: Histograms of the Singular Values $\nu_i$ and associated Eigenvalues $\lambda_i = \nu_i^2$, comparing initial $\mathbf{W}_l^0$ and final $\mathbf{W}_l$ weight matrices (which are $N \times M$, with $N = M$) for Layer FC2 of a MLP3 trained on CIFAR10.

**Limitations of these metrics.** We can gain more detailed insight into changes in $\mathbf{W}_l$ during training by creating histograms of the singular values and/or eigenvalues ($\lambda_i = \nu_i^2$). Figure 3(a) displays the density of singular values of $\mathbf{W}_{FC2}^0$ and $\mathbf{W}_{FC2}$ for the FC2 layer of the MLP3 model. Figure 3(b) displays the associated eigenvalue densities, $\rho_N(\lambda)$, which we call the *Empirical Spectral Density (ESD)* (defined in detail below) plots. Observe that the initial density of singular values (shown in red/purple), resembles a quarter circle,[6] and the final density of singular values (blue) consists of a *bulk* quarter circle, of about the same width, with several *spikes* of singular value density beyond the bulk's edge. Observe that the similar heights and widths and shapes of

---

[6]The initial weight matrix $\mathbf{W}_{FC2}^0$ is just a random (Glorot Normal) matrix.

the bulks imply the variance, or Frobenius norm, does not change much: $\|\mathbf{W}_{FC2}\|_F \approx \|\mathbf{W}_{FC2}^0\|_F$. Observe also that the initial ESD, $\rho_N(\lambda)$ (red/purple), is crisply bounded between $\lambda^- = 0$ and $\lambda^+ \sim 3.2$ (and similarly for the density of singular values at the square root of this value), whereas the final ESD (blue) has less density at $\lambda^- = 0$ and several *spikes* $\lambda \gg \lambda^+$. The largest eigenvalue is $\lambda_{max} \sim 7.2$, i.e., $\|\mathbf{W}_{FC2}\|_2^2 \approx 2 \times \|\mathbf{W}_{FC2}^0\|_2^2$. We see now why the stable rank for FC2 decreases by $\sim 2X$; the Frobenius norm does not change much, but the squared Spectral norm is $\sim 2X$ larger.

The fine-scale structure that is largely hidden from Figures 1 and 2 but that is easily-revealed by singular/eigen value density plots of Figure 3 suggests that a RMT analysis might be fruitful.

# 3 Basic Random Matrix Theory (RMT)

In this section, we summarize results from RMT that we use. RMT provides a kind-of Central Limit Theorem for matrices, with unique results for both square and rectangular matrices. Perhaps the most well-known results from RMT are the *Wigner Semicircle Law*, which describes the eigenvalues of random square *symmetric* matrices, and the *Tracy Widom (TW) Law*, which states how the maximum eigenvalue of a (more general) random matrix is distributed. Two issues arise with applying these well-known versions of RMT to DNNs. First, very rarely do we encounter *symmetric* weight matrices. Second, in training DNNs, we only have one instantiation of each weight matrix, and so it is not generally possible to apply the TW Law.[7] Several overviews of RMT are available [143, 41, 71, 142, 22, 42, 110, 24]. Here, we will describe a more general form of RMT, the *Marchenko-Pastur* (MP) theory, applicable to rectangular matrices, including (but not limited to) DNN weight matrices $\mathbf{W}$.

## 3.1 Marchenko-Pastur (MP) theory for rectangular matrices

MP theory considers the density of singular values $\rho(\nu_i)$ of random rectangular matrices $\mathbf{W}$. This is equivalent to considering the density of eigenvalues $\rho(\lambda_i)$, i.e., the ESD, of matrices of the form $\mathbf{X} = \mathbf{W}^T\mathbf{W}$. MP theory then makes strong statements about such quantities as the shape of the distribution in the infinite limit, it's bounds, expected finite-size effects, such as fluctuations near the edge, and rates of convergence. When applied to DNN weight matrices, MP theory assumes that $\mathbf{W}$, while trained on very specific datasets, exhibits statistical properties that do not depend on the specific details of the elements $W_{i,j}$, and holds even at finite size. This *Universality* concept is "borrowed" from Statistical Physics, where it is used to model, among other things, strongly-correlated systems and so-called critical phenomena in nature [134].

To apply RMT, we need only specify the number of rows and columns of $\mathbf{W}$ and assume that the elements $W_{i,j}$ are drawn from a specific distribution that is a member of a certain *Universality class* (there are different results for different Universality classes). RMT then describes properties of the ESD, even at finite size; and one can compare perdictions of RMT with empirical results. Most well-known and well-studied is the Universality class of Gaussian distributions. This leads to the basic or vanilla MP theory, which we describe in this section. More esoteric—but ultimately more useful for us—are Universality classes of Heavy-Tailed distributions. In Section 3.2, we describe this important variant.

---

[7]This is for "production" use, where one performs only a single training run; below, we will generate such distributions of weight matrices for our smaller models to denoise and illustrate better their properties.

**Gaussian Universality class.** We start by modeling $\mathbf{W}$ as an $N \times M$ random matrix, with elements drawn from a Gaussian distribution, such that:

$$W_{ij} \sim N(0, \sigma_{mp}^2).$$

Then, MP theory states that the ESD of the correlation matrix, $\mathbf{X} = \mathbf{W}^T\mathbf{W}$, has the limiting density given by the MP distribution $\rho(\lambda)$:

$$
\rho_N(\lambda) \quad := \quad \frac{1}{N}\sum_{i=1}^{M}\delta\left(\lambda - \lambda_i\right)
$$

$$
\xrightarrow[Q \text{ fixed}]{N\to\infty} \quad
\begin{cases}
\dfrac{Q}{2\pi\sigma_{mp}^2}\dfrac{\sqrt{(\lambda^+ - \lambda)(\lambda - \lambda^-)}}{\lambda} & \text{if } \lambda \in [\lambda^-, \lambda^+]\\
0 & \text{otherwise.}
\end{cases}
\tag{8}
$$

Here, $\sigma_{mp}^2$ is the element-wise variance of the original matrix, $Q = N/M \geq 1$ is the aspect ratio of the matrix, and the minimum and maximum eigenvalues, $\lambda^{\pm}$, are given by

$$\lambda^{\pm} = \sigma_{mp}^2\left(1 \pm \frac{1}{\sqrt{Q}}\right)^2. \tag{9}$$

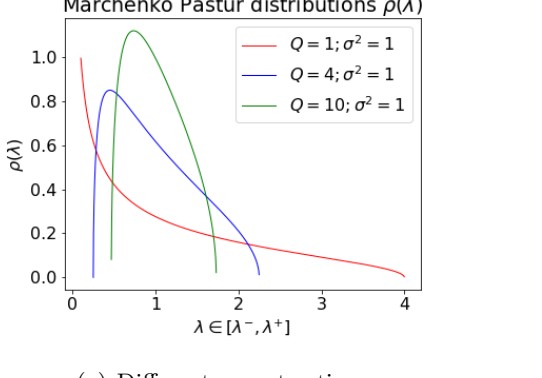 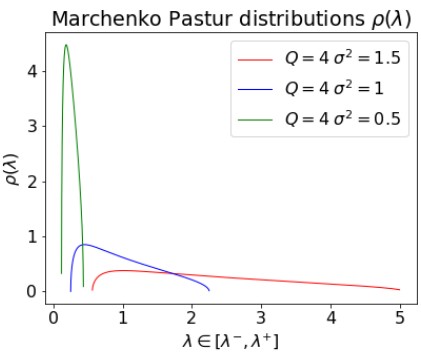

(a) Different aspect ratios        (b) Different variance parameters

Figure 4: Marchenko-Pastur (MP) distributions, see Eqns. (8) and (9), as the aspect ratio $Q$ and variance parameter $\sigma$ are modified.

**The MP distribution for different aspect ratios $Q$ and variance parameters $\sigma_{mp}$.** The shape of the MP distribution only depends on two parameters, the variance $\sigma_{mp}^2$ and the aspect ratio $Q$. See Figure 4 for an illustration. In particular, see Figure 4(a) for a plot of the MP distribution of Eqns. (8) and (9), for several values of $Q$; and see Figure 4(b) for a plot of the MP distribution for several values of $\sigma_{mp}$.

As a point of reference, when $Q = 4$ and $\sigma_{mp} = 1$ (blue in both subfigures), the mass of $\rho_N$ skews slightly to the left, and is bounded in $[0.3 - 2.3]$. For fixed $\sigma_{mp}$, as $Q$ increases, the support (i.e., $[\lambda^-, \lambda^+]$) narrows, and $\rho_N$ becomes less skewed. As $Q \to 1$, the support widens and $\rho_N$ skews more leftward. Also, $\rho_N$ is concave for larger $Q$, and it is partially convex for smaller $Q = 1$.

Although MP distribution depends on $Q$ and $\sigma_{mp}^2$, in practice $Q$ is fixed, and thus we are interested how $\sigma_{mp}^2$ varies—distributionally for random matrices, and empirically for weight matrices. Due to Eqn. (9), if $\sigma_{mp}^2$ is fixed, then $\lambda^+$ (i.e., the largest eigenvalue of the bulk, as well as $\lambda^-$) is determined, and vice versa.[8]

---

[8]In practice, relating $\lambda^+$ and $\sigma_{mp}^2$ raises some subtle technical issues, and we discuss these in Section 6.3.

**The Quarter Circle Law for $Q = 1$.** A special case of Eqn. (8) arises when $Q = 1$, i.e., when $\mathbf{W}$ is a square non-symmetric matrix. In this case, the eigenvalue density $\rho(\lambda)$ is very peaked with a bounded tail, and it is sometimes more convenient to consider the density of singular values of $\mathbf{W}_l$, $\rho(\nu)$, which takes the form of a *Quarter-Circle*:

$$\rho(\nu) = \frac{1}{\pi\sigma_{mp}^2}\sqrt{4 - \nu^2}.$$

We will not pursue this further, but we saw this earlier, in Figure 3(b), with our toy MLP3 model.

**Finite-size Fluctuations at the MP Edge.** In the infinite limit, all fluctuations in $\rho_N(\lambda)$ concentrate very sharply at the MP edge, $\lambda^\pm$, and the distribution of the maximum eigenvalues $\rho_\infty(\lambda_{max})$ is governed by the TW Law. Even for a single finite-sized matrix, however, MP theory states the upper edge of $\rho(\lambda)$ is very sharp; and even when the MP Law is violated, the TW Law, with finite-size corrections, works very well at describing the edge statistics. When these laws are violated, this is very strong evidence for the onset of more regular non-random structure in the DNN weight matrices, which we will interpret as evidence of *Self-Regularization*.

In more detail, in many cases, one or more of the empirical eigenvalues will extend beyond the sharp edge predicted by the MP fit, i.e., such that $\lambda_{max} > \lambda^+$ (where $\lambda_{max}$ is the largest eigenvalue of $\mathbf{X}$). It will be important to distinguish the case that $\lambda_{max} > \lambda^+$ simply due the finite size of $\mathbf{W}$ from the case that $\lambda_{max}$ is "truly" outside the MP bulk. According to MP theory [22], for finite $(N, M)$, and with $\frac{1}{N}$ normalization, the fluctuations at the bulk edge scale as $\mathcal{O}(M^{-\frac{2}{3}})$:

$$\Delta\lambda_M := \|\lambda_{max} - \lambda^+\|^2 = \frac{1}{\sqrt{Q}}(\lambda^+)^{2/3}M^{-2/3},$$

where $\lambda^+$ is given by Eqn (9). Since $Q = N/M$, we can also express this in terms of $N^{-2/3}$, but with different prefactors [68]. Most importantly, within MP theory (and even more generally), the $\lambda_{max}$ fluctuations, centered and rescaled, will follow TW statistics.

In the DNNs we consider, $M \gtrsim 400$, and so the maximum deviation is only $\Delta\lambda_M \lesssim 0.02$. In many cases, it will be obvious whether a given $\lambda_{max}$ is an outlier. When it is not, one could generate an ensemble of $N_R$ runs and study the information content of the eigenvalues (shown below) and/or apply TW theory (not discussed here).

**Fitting MP Distributions.** Several technical challenges with fitting MP distributions, i.e., selecting the bulk edge $\lambda^+$, are discussed in Section 6.3.

## 3.2 Heavy-Tailed extensions of MP theory

MP-based RMT is applicable to a wide range of matrices (even those with large low-rank perturbations $\Delta^{large}$ to i.i.d. normal behavior); but it is *not* in general applicable when matrix elements are strongly-correlated. Strong correlations appear to be the case for many well-trained, production-quality DNNs. In statistical physics, it is common to *model* strongly-correlated systems by Heavy-Tailed distributions [134]. The reason is that these models exhibit, more or less, the same large-scale statistical behavior as natural phenomena in which strong correlations exist [134, 22]. Moreover, recent results from MP/RMT have shown that new Universality classes exist for matrices with elements drawn from certain Heavy-Tailed distributions [22].

We use these Heavy-Tailed extensions of basic MP/RMT to build an operational and phenomenological theory of Regularization in Deep Learning; and we use these extensions to justify

| | Generative Model w/ elements from Universality class | Finite-$N$ Global shape $\rho_N(\lambda)$ | Limiting Global shape $\rho(\lambda)$, $N \to \infty$ | Bulk edge Local stats $\lambda \approx \lambda^+$ | (far) Tail Local stats $\lambda \approx \lambda_{max}$ |
|---|---|---|---|---|---|
| Basic MP | Gaussian | MP, i.e., Eqn. (8) | MP | TW | No tail. |
| Spiked-Covariance | Gaussian, + low-rank perturbations | MP + Gaussian spikes | MP | TW | Gaussian |
| Heavy tail, $4 < \mu$ | (Weakly) Heavy-Tailed | MP + PL tail | MP | Heavy-Tailed* | Heavy-Tailed* |
| Heavy tail, $2 < \mu < 4$ | (Moderately) Heavy-Tailed (or "fat tailed") | PL** $\sim \lambda^{-(a\mu+b)}$ | PL $\sim \lambda^{-(\frac{1}{2}\mu+1)}$ | No edge. | Frechet |
| Heavy tail, $0 < \mu < 2$ | (Very) Heavy-Tailed | PL** $\sim \lambda^{-(\frac{1}{2}\mu+1)}$ | PL $\sim \lambda^{-(\frac{1}{2}\mu+1)}$ | No edge. | Frechet |

Table 3: Basic MP theory, and the spiked and Heavy-Tailed extensions we use, including known, empirically-observed, and conjectured relations between them. Boxes marked "*" are best described as following "TW with large finite size corrections" that are likely Heavy-Tailed [20], leading to bulk edge statistics and far tail statistics that are indistinguishable. Boxes marked "**" are phenomenological fits, describing large ($2 < \mu < 4$) or small ($0 < \mu < 2$) finite-size corrections on $N \to \infty$ behavior. See [38, 20, 19, 111, 7, 40, 8, 26, 22, 21] for additional details.

our analysis of both *Self-Regularization* and *Heavy-Tailed Self-Regularization*.[9] Briefly, our theory for simple *Self-Regularization* is inspired by the Spiked-Covariance model of Johnstone [68] and it's interpretation as a form of *Self-Organization* by Sornette [92]; and our theory for more sophisticated *Heavy-Tailed Self-Regularization* is inspired by the application of MP/RMT tools in quantitative finance by Bouchuad, Potters, and coworkers [49, 77, 78, 20, 19, 22, 24], as well as the relation of Heavy-Tailed phenomena more generally to *Self-Organized Criticality* in Nature [134]. Here, we highlight basic results for this generalized MP theory; see [38, 20, 19, 111, 7, 40, 8, 26, 22, 21] in the physics and mathematics literature for additional details.

**Universality classes for modeling strongly correlated matrices.** Consider modeling $\mathbf{W}$ as an $N \times M$ random matrix, with elements drawn from a Heavy-Tailed—e.g., a Pareto or Power Law (PL)—distribution:

$$W_{ij} \sim P(x) \sim \frac{1}{x^{1+\mu}}, \ \ \mu > 0. \tag{10}$$

In these cases, if $\mathbf{W}$ is element-wise Heavy-Tailed,[10] then the ESD $\rho_N(\lambda)$ likewise exhibits Heavy-Tailed properties, either globally for the entire ESD and/or locally at the bulk edge.

Table 3 summarizes these (relatively) recent results, comparing basic MP theory, the Spiked-Covariance model,[11] and Heavy-Tailed extensions of MP theory, including associated Universality classes. To apply the MP theory, at finite sizes, to matrices with elements drawn from a Heavy-Tailed distribution of the form given in Eqn. (10), then, depending on the value of $\mu$, we have

---

[9] The *Universality of RMT* is a concept broad enough to apply to classes of problems that appear well beyond its apparent range of validity. It is in this sense that we apply RMT to understand DNN Regularization.

[10] Heavy-Tailed phenomena have many subtle properties [121]; we consider here only the most simple cases.

[11] We discuss Heavy-Tailed extensions to MP theory in this section. Extensions to large low-rank perturbations are more straightforward and are described in Section 5.3.

one of the following three[12] Universality classes:

- **(Weakly) Heavy-Tailed**, $4 < \mu$: Here, the ESD $\rho_N(\lambda)$ exhibits "vanilla" MP behavior in the infinite limit, and the expected mean value of the bulk edge is $\lambda^+ \sim M^{-2/3}$. Unlike standard MP theory, which exhibits TW statistics at the bulk edge, here the edge exhibits PL / Heavy-Tailed fluctuations at finite $N$. These finite-size effects appear in the edge / tail of the ESD, and they make it hard or impossible to distinguish the edge versus the tail at finite $N$.

- **(Moderately) Heavy-Tailed**, $2 < \mu < 4$: Here, the ESD $\rho_N(\lambda)$ is Heavy-Tailed / PL in the infinite limit, approaching the form $\rho(\lambda) \sim \lambda^{-1-\mu/2}$. In this regime of $\mu$, there is no bulk edge. At finite size, the global ESD can be modeled by the form $\rho_N(\lambda) \sim \lambda^{-(a\mu+b)}$, for all $\lambda > \lambda_{min}$, but the slope $a$ and intercept $b$ must be fit, as they display very large finite-size effects. The maximum eigenvalues follow Frechet (not TW) statistics, with $\lambda_{max} \sim M^{4/\mu-1}(1/Q)^{1-2/\mu}$, and they have large finite-size effects. Even if the ESD tends to zero, the raw number of eigenvalues can still grow—just not as quickly as $N$ (i.e., we may expect some $\lambda_{max} > \lambda^+$, in the infinite limit, but the eigenvalue *density* $\rho(\lambda) \to 0$). Thus, at any finite $N$, $\rho_N(\lambda)$ is Heavy-Tailed, but the tail decays moderately quickly.

- **(Very) Heavy-Tailed**, $0 < \mu < 2$: Here, the ESD $\rho_N(\lambda)$ is Heavy-Tailed / PL for all finite $N$, and as $N \to \infty$ it converges more quickly to a PL distribution with tails $\rho(\lambda) \sim \lambda^{-1-\mu/2}$. In this regime, there is no bulk edge, and the maximum eigenvalues follow Frechet (not TW) statistics. Finite-size effects exist here, but they are are much smaller here than in the $2 < \mu < 4$ regime of $\mu$.

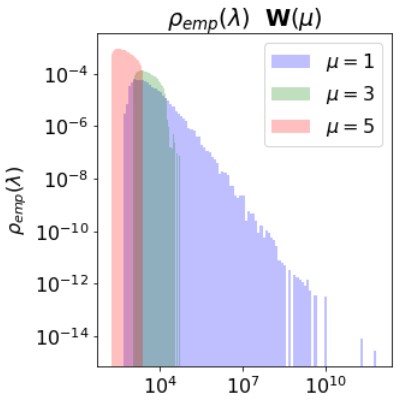 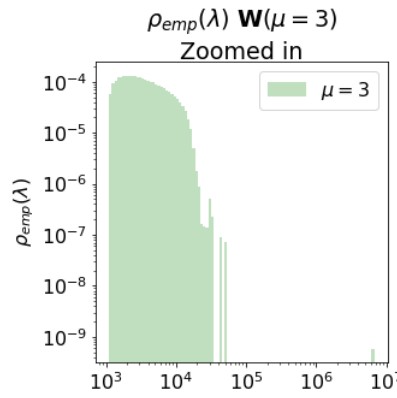

(a) Three log-log histograms.  (b) Zoomed-in histogram for $\mu = 3$.

Figure 5: The log-log histogram plots of the ESD for three Heavy-Tailed random matrices **M** with same aspect ratio $Q = 3$, with $\mu = 1.0, 3.0, 5.0$, corresponding to the three Heavy-Tailed Universality classes ($0 < \mu < 2$ vs $2 < \mu < 4$ and $4 < \mu$) described in Table 3.

**Visualizing Heavy-Tailed distributions.** It is often fruitful to perform visual exploration and classification of ESDs by plotting them on linear-linear coordinates, log-linear coordinates (linear horizontal/X axis and logarithmic vertical/Y axis), and/or log-log coordinates (logarithmic

---

[12]Both $\mu = 2$ and $\mu = 4$ are "corner cases" that we don't expect to be able to resolve at finite $N$.

horizontal/X axis and logarithmic vertical/Y axis). It is known that data from a PL distribution will appear as a convex curve in a linear-linear plot and a log-linear plot and as a straight line in a log-log plot; and that data from a Gaussian distribution will appear as a bell-shaped curve in a linear-linear plot, as an inverted parabola in a log-linear plot, and as a strongly concave curve in a log-log plot. Examining data from an unknown ESD on different axes *suggests* a classification for them. (See Figures 5 and 16.) More quantitative analysis *may* lead to more definite conclusions, but that too comes with technical challenges.

To illustrate this, we provide a visual and operational approach to understand the limiting forms for different $\mu$. See Figure 5. Figure 5(a) displays the log-log histograms for the ESD $\rho_N(\lambda)$ for three Heavy-Tailed random matrices $\mathbf{M}_N(\mu)$, with $\mu = 1.0, 3.0, 5.0$ For $\mu = 1.0$ (blue), the log-log histogram is linear over 5 log scales, from $10^3 - 10^8$. If $N$ increases (not shown), $\lambda_{max}$ will grow, but this plot will remain linear, and the tail will not decay. In the infinite limit, the ESD will still be Heavy-Tailed. Contrast this with the ESD drawn from the same distribution, except with $\mu = 3.0$ (green). Here, due to larger finite-size effects, most of the mass is confined to one or two log scales, and it starts to vanish when $\lambda > 10^3$. This effect is amplified for $\mu = 5.0$ (red), which shows almost no mass for eigenvalues beyond the MP bulk (i.e. $\lambda > \lambda^+$). Zooming in, in Figure 5(b), we see that the log-log plot is linear—*in the central region only*—and the tail vanishes very quickly. If $N$ increases (not shown), the ESD will remain Heavy-Tailed, but the mass will grow much slower than when $\mu < 2$. This illustrates that, while ESDs can be Heavy-Tailed at finite size, the tails decay at different rates for different Heavy-Tailed Universality classes ($0 < \mu < 2$ or $2 < \mu < 4$ or $4 < \mu$).

**Fitting PL distributions to ESD plots.** Once we have identified PL distributions visually (using a log-log histogram of the ESD, and looking for visual characteristics of Figure 5), we can fit the ESD to a PL in order to obtain the exponent $\alpha$. For this, we use the Clauset-Shalizi-Newman (CSN) approach [33], as implemented in the python PowerLaw package [2],[13] which computes an $\alpha$ such that

$$\rho_{emp}(\lambda) \sim \lambda^{-\alpha}.$$

Generally speaking, fitting a PL has many subtleties, most beyond the scope of this paper [33, 55, 91, 100, 16, 73, 35, 2, 145, 58]. For example, care must be taken to ensure the distribution is actually linear (in some regime) on a log-log scale before applying the PL estimator, lest it give spurious results; and the PL estimator only works reasonably well for exponents in the range $1.5 < \alpha \lesssim 3.5$.

To illustrate this, consider Figure 6. In particular, Figure 6(a) shows that the CSN estimator performs well for the regime $0 < \mu < 2$, while for $2 < \mu < 4$ there are substantial deviations due to finite-size effects, and for $4 < \mu$ no reliable results are obtained; and Figures 6(b) and 6(c) show that the finite-size effects can be quite complex (for fixed $M$, increasing $Q$ leads to larger finite-size effects, while for fixed $N$, decreasing $Q$ leads to larger finite-size effects).

**Identifying the Universality class.** Given $\alpha$, we identify the corresponding $\mu$ (as illustrated in Figure 6) and thus which of the three Heavy-Tailed Universality classes ($0 < \mu < 2$ or $2 < \mu < 4$ or $4 < \mu$, as described in Table 5) is appropriate to describe the system. For our theory, the following are particularly important points. First, observing a Heavy-Tailed ESD may indicate the presence of a scale-free DNN. This suggests that the underlying DNN is strongly-correlated, and that we need more than just a few separated spikes, plus some random-like bulk structure, to model the DNN and to understand DNN regularization. Second, this does not necessarily imply

---

[13]See https://github.com/jeffalstott/powerlaw.

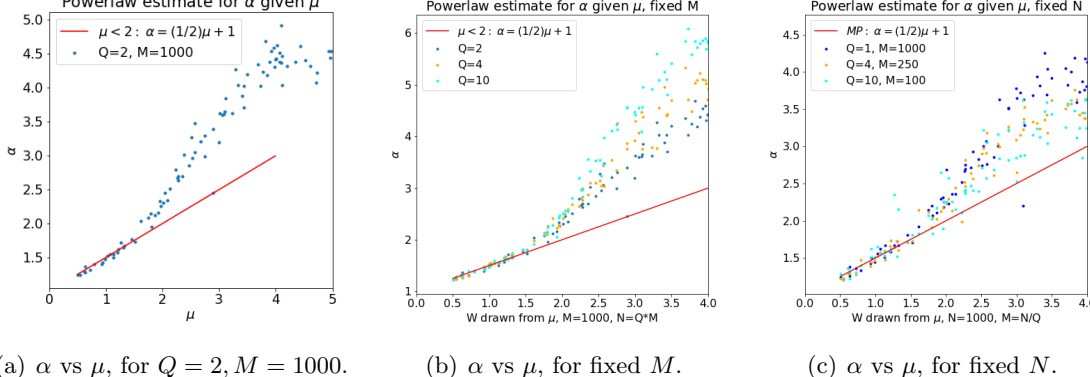

(a) $\alpha$ vs $\mu$, for $Q = 2, M = 1000$.  (b) $\alpha$ vs $\mu$, for fixed $M$.  (c) $\alpha$ vs $\mu$, for fixed $N$.

Figure 6: Dependence of $\alpha$ (the fitted PL parameter) on $\mu$ (the hypothesized limiting PL parameter). In (6(a)), the PL exponent $\alpha$ is fit, using the CSN estimator, for the ESD $\rho_{emp}(\lambda)$ for a random, rectangular Heavy-Tailed matrix $\mathbf{W}(\mu)$ ($Q = 2, M = 1000$), with elements drawn from a Pareto distribution $p(x) \sim x^{-1-\mu}$. For $0 < \mu < 2$, finite-size effects are modest, and the ESD follows the theoretical prediction $\rho_{emp}(\lambda) \sim \lambda^{-1-\mu/2}$. For $2 < \mu < 4$, the ESD still shows roughly linear behavior, but with significant finite-size effects, giving the more general phenomenological relation $\rho_{emp}(\lambda) \sim \lambda^{-a\mu+b}$. For $4 < \mu$, the CSN method is known to fail to perform well. In (6(b)) and (6(c)), plots are shown for varying $Q$, with $M$ and $N$ fixed, respectively.

that the matrix elements of $\mathbf{W}_l$ form a Heavy-Tailed distribution. Rather, the Heavy-Tailed distribution arises since we posit it as a model of the strongly correlated, highly non-random matrix $\mathbf{W}_l$. Third, we conjecture that this is more general, and that very well-trained DNNs will exhibit Heavy-Tailed behavior in their ESD for many the weight matrices (as we have observed so far with many pre-trained models).

## 3.3 Eigenvector localization

When entries of a random matrix are drawn from distributions in the Gaussian Universality class, and under typical assumptions, eigenvectors tend to be delocalized, i.e., the mass of the eigenvector tends to be spread out on most or all the components of that vector. For other models, eigenvectors can be localized. For example, spike eigenvectors in Spiked-Covariance models as well as extremal eigenvectors in Heavy-Tailed random matrix models tend to be more localized [110, 24]. Eigenvector delocalization, in traditional RMT, is modeled using the *Thomas Porter Distribution* [118]. Since a typical *bulk* eigenvector $\mathbf{v}$ should have maximum entropy, therefore it's components $v_i$ should be Gaussian distributed, according to:

$$\textit{Thomas Porter Distribution}: \quad P_{tp}(v_i) = \frac{1}{2\pi} \exp\left(-\frac{v_i^2}{2}\right).$$

Here, we normalize $\mathbf{v}$ such that the empirical variance of the elements is unity, $\sigma_{v_i}^2 = 1$. Based on this, we can define several related eigenvector localization metrics.

- The *Generalized Vector Entropy*, $S(\mathbf{v}) := \sum_i P(v_i) \ln P(v_i)$, is computed using a histogram estimator.

- The *Localization Ratio*, $\mathcal{L}(\mathbf{v}) := \dfrac{\|\mathbf{v}\|_1}{\|\mathbf{v}\|_\infty}$, measures the sum of the absolute values of the elements of $\mathbf{v}$, relative to the largest absolute value of an element of $\mathbf{v}$.

- The *Participation Ratio*, $\mathcal{P}(\mathbf{v}) := \dfrac{\|\mathbf{v}\|_2}{\|\mathbf{v}\|_4}$, is a robust variant of the Localization Ratio.

For all three metrics, the lower the value, the more localized the eigenvector $\mathbf{v}$ tends to be. We use deviations from delocalization as a diagnostic that the corresponding eigenvector is more structured/regularized.

# 4 Empirical Results: ESDs for Existing, Pretrained DNNs

In this section, we describe our main empirical results for existing, pretrained DNNs.[14] Early on, we observed that small DNNs and large DNNs have very different ESDs. For smaller models, ESDs tend to fit the MP theory well, with well-understood deviations, e.g., low-rank perturbations. For larger models, the ESDs $\rho_N(\lambda)$ almost never fit the theoretical $\rho_{mp}(\lambda)$, and they frequently have a completely different functional form. We use RMT to compare and contrast the ESDs of a smaller, older NN and many larger, modern DNNs. For the small model, we retrain a modern variant of one of the very early and well-known Convolutional Nets—LeNet5. We use Keras (2), and we train LeNet5 on MNIST. For the larger, modern models, we examine selected layers from AlexNet, InceptionV3, and many other models (as distributed with pyTorch). Table 4 provides a summary of models we analyzed in detail.

## 4.1 Example: LeNet5 (1998)

LeNet5 predates both the current Deep Learning revolution and the so-called AI Winter, dating back to the late 1990s [79]. It is the prototype early model for DNNs; it is the most widely-known example of a Convolutional Neural Network (CNN); and it was used in production systems for recognizing hand written digits [79]. The basic design consists of 2 Convolutional (Conv2D) and MaxPooling layers, followed by 2 Dense, or Fully Connected (FC), layers, FC1 and FC2. This design inspired modern DNNs for image classification, e.g., AlexNet, VGG16 and VGG19. All of these latter models consist of a few Conv2D and MaxPooling layers, followed by a few FC layers. Since LeNet5 is older, we actually recoded and retrained it. We used Keras 2.0, using 20 epochs of the AdaDelta optimizer, on the MNIST data set. This model has 100.00% training accuracy, and 99.25% test accuracy on the default MNIST split. We analyze the ESD of the FC1 Layer (but not the FC2 Layer since it has only 10 eigenvalues). The FC1 matrix $\mathbf{W}_{FC1}$ is a $2450 \times 500$ matrix, with $Q = 4.9$, and thus it yields 500 eigenvalues.

**FC1: MP Bulk+Spikes, with edge Bleeding-out.** Figure 7 presents the ESD for FC1 of LeNet5, with Figure 7(a) showing the full ESD and Figure 7(b) showing the same ESD, zoomed-in along the X-axis to highlight smaller peaks outside the main bulk of our MP fit. In both cases, we show (red curve) our fit to the MP distribution $\rho_{emp}(\lambda)$. Several things are striking. First, the *bulk* of the density $\rho_{emp}(\lambda)$ has a large, MP-like shape for eigenvalues $\lambda < \lambda^+ \approx 3.5$, and the MP distribution fits this part of the ESD *very* well, including the fact that the ESD just below the best fit $\lambda^+$ is concave. Second, *some eigenvalue mass is bleeding out* from the MP bulk for $\lambda \in [3.5, 5]$, although it is quite small. Third, beyond the MP bulk and this bleeding out region, are several *clear outliers, or spikes*, ranging from $\approx 5$ to $\lambda_{max} \lesssim 25$.

**Summary.** The shape of $\rho_{emp}(\lambda)$, the quality of the global bulk fit, and the statistics and crisp shape of the local bulk edge all agree well with standard MP theory, or at least the variant of

---

[14]A practical theory for DNNs should be applicable to very large—production-quality, and even pre-trained—models, as well as to models during training.

| | Used for: | Section | Layer | Key observation in ESD(s) |
|---|---|---|---|---|
| MLP3 | Initial illustration of entropy and spectral properties | 2 | FC1 FC2 | MP BULK+SPIKES |
| LeNet5 | Old state-of-the-art model | 4.1 | FC1 | MP BULK+SPIKES, with edge BLEEDING-OUT |
| AlexNet | More recent pre-trained state-of-the-art model | 4.2 | FC1 | ESD BULK-DECAY into a HEAVY-TAILED |
| | Typical properties from Heavy-Tailed Universality classes | 5.5 | FC2 | $\mu \approx 2$ |
| | | | FC3 | $\mu \approx 2.5$ |
| InceptionV3 | More recent pre-trained state-of-the-art model with unusual properties | 4.3 | L226 | Bimodel ESD w/ HEAVY-TAILED envelope |
| | | 4.3 | L302 | Bimodal and "fat" or HEAVY-TAILED ESD |
| MiniAlexNet | Detailed analysis illustrating properties as training knobs change | 6 | FC1 FC2 | Exhibits all 5+1 Phases of Training by changing batch size |

Table 4: Description of main DNNs used in our analysis and the key observations about the ESDs of the specific layer weight matrices using RMT. Names in the "Key observation" column are defined in Section 5 and described in Table 7.

MP theory augmented with a low-rank perturbation. In this sense, this model can be viewed as a real-world example of the Spiked-Covariance model [68].

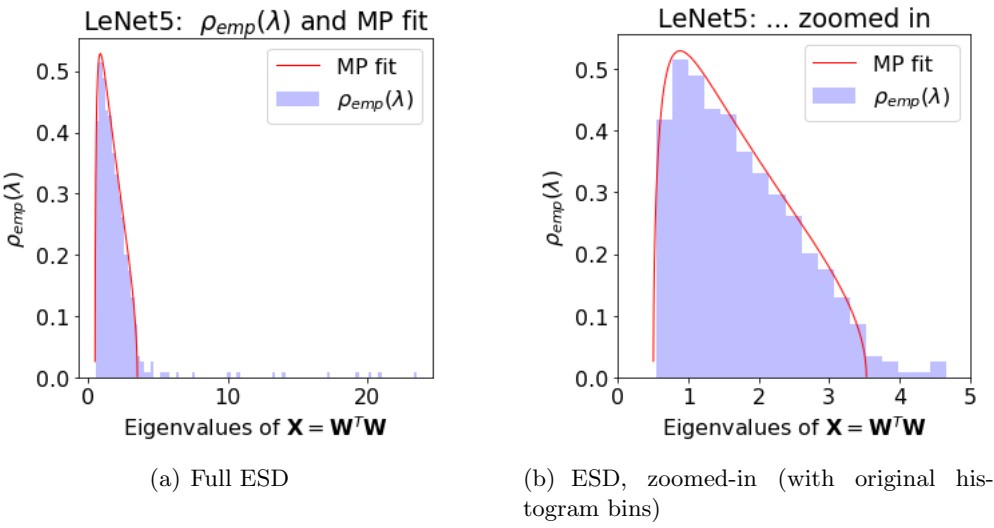

(a) Full ESD

(b) ESD, zoomed-in (with original histogram bins)

Figure 7: Full and zoomed-in ESD for LeNet5, Layer FC1. Overlaid (in red) are gross fits of the MP distribution (which fit the bulk of the ESD very well).

## 4.2 Example: AlexNet (2012)

AlexNet was the first modern DNN, and its spectacular performance opened the door for today's revolution in Deep Learning. Specifically, it was top-5 on the ImageNet ILSVRC2012 classification task [74], achieving an error of 16.4%, over 11% ahead of the first runner up. AlexNet resembles a scaled-up version of the LeNet5 architecture; it consists of 5 layers, 2 convolutional, followed by 3 FC layers (the last being a softmax classifier).[15] We will analyze the version of AlexNet currently distributed with pyTorch (version 0.4.1). In this version, FC1 has a $9216 \times 4096$ matrix, with $Q = 2.25$; FC2 has a $4096 \times 4096$ matrix, with $Q = 1.0$; and FC3 has a $4096 \times 1000$ matrix, with $Q = 4.096 \approx 4.1$. Notice that FC3 is the final layer and connects AlexNet to the labels.

Figures 8, 9, and 10 present the ESDs for weight matrices of AlexNet for Layers FC1, FC2, and FC3, with Figures 8(a), 9(a), and 10(a) showing the full ESD, and Figures 8(b), 9(b), and 10(b) showing the results "zoomed-in" along the X-axis. In each cases, we present best MP fits, as determined by holding $Q$ fixed, adjusting the $\sigma$ parameter, and selecting the best bulk fit by visual inspection. Fitting $\sigma$ fixes $\lambda^+$, and the $\lambda^+$ estimates differ for different layers because the matrices have different aspect ratios $Q$. In each case, the ESDs exhibit moderate to strong deviations from the best standard MP fit.

**FC1: Bulk-decay into Heavy-Tailed.** Consider first AlexNet FC1 (in Figures 8(a) and 8(b)). The eigenvalues range from near 0 up to ca. 30, just as with LeNet5. The full ESD, however, is shaped very differently than any theoretical $\rho_{mp}(\lambda)$, for any value of $\lambda$. The best MP fit (in red in Figure 8) does capture a good part of the eigenvalue mass, but there are important differences: the peak is not filled in, there is substantial eigenvalue mass bleeding out from the bulk, and the shape of the ESD is convex in the region near to and just above the best fit for $\lambda^+$ of the bulk edge. Contrast this with the excellent MP fit for the ESD for FC1 of LeNet5 (Figure 7(b)), where the red curve captures all of the bulk mass, and only a few outlying spikes appear. Moreover, and very importantly, in AlexNet FC1, the bulk edge is *not* crisp. In fact, it is not visible at all; and $\lambda^+$ is solely defined operationally by selecting the $\sigma$ parameter. As such, the edge fluctuations, $\Delta\lambda$, do not resemble a TW distribution, and the bulk itself appears to just *decay* into the heavy tail. Finally, a PL fit gives good fit $\alpha \approx 2.29$, suggesting (due to finite size effects) $\mu \lesssim 2.5$.

**FC2: (nearly very) Heavy-Tailed ESD.** Consider next AlexNet FC2 (in Figures 9(a) and 9(b)). This ESD differs even more profoundly from standard MP theory. Here, we could find no good MP fit, even by adjusting $\sigma$ and $Q$ simultaneously. The best MP fit (in red) does not fit the Bulk part of $\rho_{emp}(\lambda)$ at all. The fit suggests there should be significantly more bulk eigenvalue mass (i.e., larger empirical variance) than actually observed. In addition, as with FC1, the bulk edge is indeterminate by inspection. It is only defined by the crude fit we present, and any edge statistics obviously do not exhibit TW behavior. In contrast with MP curves, which are convex near the bulk edge, the entire ESD is concave (nearly) everywhere. Here, a PL fit gives good fit $\alpha \approx 2.25$, smaller than FC1 and FC3, indicating a $\mu \lesssim 3$.

**FC3: Heavy-Tailed ESD.** Consider finally AlexNet FC3 (in Figures 10(a) and 10(b)). Here, too, the ESDs deviate strongly from predictions of MP theory, both for the global bulk properties and for the local edge properties. A PL fit gives good fit $\alpha \approx 3.02$, which is larger than FC1 and FC2. This suggests a $\mu \lesssim 2.5$ (which is also shown with a log-log histogram plot in Figure 16 in Section 5 below).

---

[15]It was the first CNN Net to win this competition, and it was also the first DNN to use ReLUs in a wide scale competition. It also used a novel Local Contrast Normalization layer, which is no longer in widespread use, having been mostly replaced with Batch Normalization or similar methods.

**Summary.** For all three layers, the shape of $\rho_{emp}(\lambda)$, the quality of the global bulk fit, and the statistics and shape of the local bulk edge are poorly-described by standard MP theory. Even when we may think we have moderately a good MP fit because the bulk shape is qualitatively captured with MP theory (at least visual inspection), we may see a complete breakdown RMT at the bulk edge, where we expect crisp TW statistics (or at least a concave envelope of support). In other cases, the MP theory may even be a poor estimator for even the bulk.

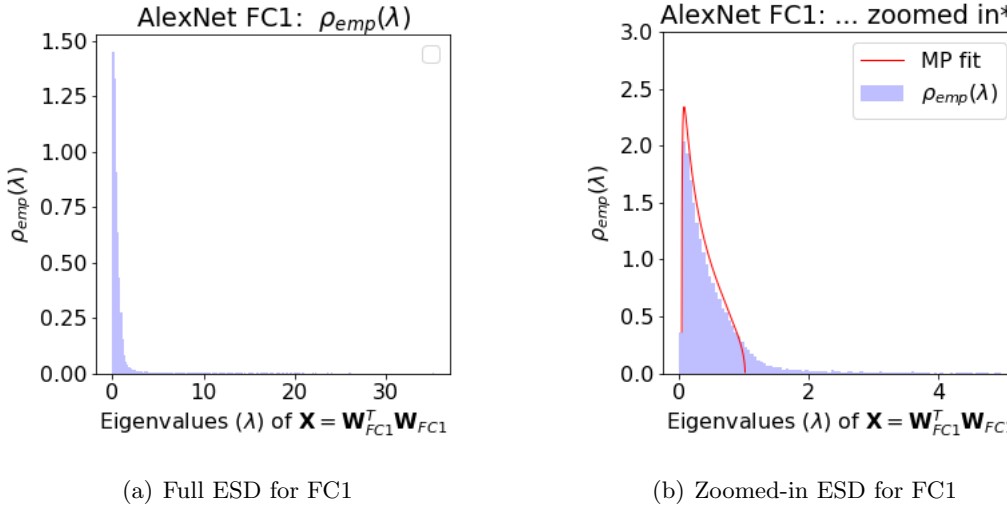

(a) Full ESD for FC1

(b) Zoomed-in ESD for FC1

Figure 8: ESD for Layer FC1 of AlexNet. Overlaid (in red) are gross fits of the MP distribution.

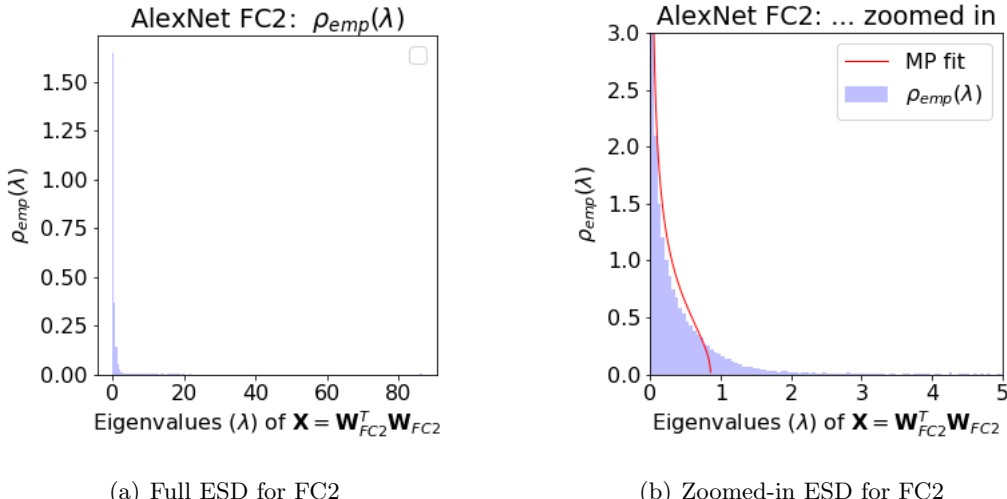

(a) Full ESD for FC2

(b) Zoomed-in ESD for FC2

Figure 9: ESD for Layer FC2 of AlexNet. Overlaid (in red) are gross fits of the MP distribution.

### 4.3 Example: InceptionV3 (2014)

In the few years after AlexNet, several new, deeper DNNs started to win the ILSVRC ImageNet completions, including ZFNet(2013) [155], VGG(2014) [130], GoogLeNet/Inception (2014) [139], and ResNet (2015) [61]. We have observed that nearly all of these DNNs have properties that

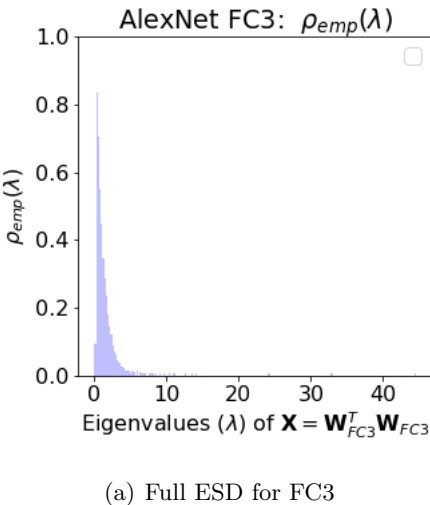
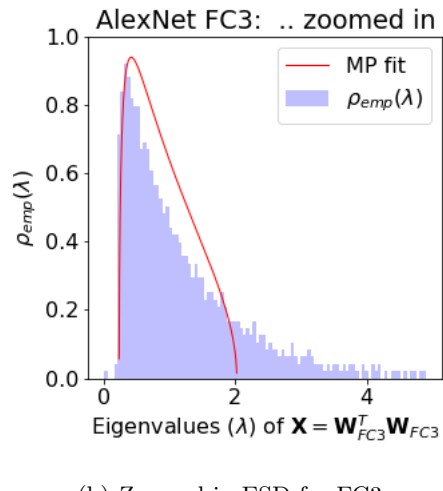

(a) Full ESD for FC3

(b) Zoomed-in ESD for FC3

Figure 10: ESD for Layer FC3 of AlexNet. Overlaid (in red) are gross fits of the MP distribution.

are similar to AlexNet. Rather than describe them all in detail, in Section 4.4, we perform power law fits on the Linear/FC layers in many of these models. Here, we want to look more deeply at the Inception model, since it displays some unique properties.[16]

In 2014, the VGG [130] and GoogLeNet [139] models were close competitors in the ILSVRC2014 challenges. For example, GoogLeNet won the classification challenge, but VGG performed better on the localization challenge. These models were quite deep, with GoogLeNet having 22 layers, and VGG having 19 layers. The VGG model is ~2X as deep as AlexNet, but it replaces each larger AlexNet filter with more, smaller filters. Presumably this deeper architecture, with more non-linearities, can capture the correlations in the network better. The VGG features of the second to last FC layer generalize well to other tasks. A downside of the VGG models is that they have a lot of parameters and that they use a lot of memory.

The GoogLeNet/Inception design resembles the VGG architecture, but it is even more computationally efficient, which (practically) means smaller matrices, fewer parameters (12X fewer than AlexNet), and a very different architecture, including no internal FC layers, except those connected to the labels. In particular, it was noted that most of the activations in these DNNs are redundant because they are so strongly correlated. So, a sparse architecture should perform just as well, but with much less computational cost—if implemented properly to take advantage of low level BLAS calculations on the GPU. So, an Inception module was designed. This module approximates a sparse Convolutional Net, but using many smaller, dense matrices, leading to many small filters of different sizes, concatenated together. The Inception modules are then stacked on top of each other to give the full DNN. GoogLeNet also replaces the later FC layers (i.e., in AlexNet-like architectures) with global average pooling, leaving only a single FC / Dense layer, which connects the DNN to the labels. Being so deep, it is necessary to include an Auxiliary block that also connects to the labels, similar to the final FC layer. From this, we can extract a single rectangular $768 \times 1000$ tensor. This gives 2 FC layers to analyze.

For our analysis of InceptionV3 [139], we select a layer (L226) from in the Auxiliary block, as well as the final (L302) FC layer. Figure 11 presents the ESDs for InceptionV3 for Layer L226 and Layer L302, two large, fully-connected weight matrices with aspect ratios $Q \approx 1.3$ and $Q = 2.048$, respectively. We also show typical MP fits for matrices with the same aspect ratios

[16]Indeed, these results suggest that Inception models do not truly account for all the correlations in the data.

$Q$. As with AlexNet, the ESDs for both the L226 and L302 layers display distinct and strong deviations from the MP theory.

**L226: Bimodal ESDs.** Consider first L226 of InceptionV3. Figure 11(a) displays the L226 ESD. (Recall this is not a true Dense layer, but it is part of the Inception Auxiliary module, and it looks very different from the other FC layers, both in AlexNet and below.) At first glance, we might hope to select the bulk edge at $\lambda^+ \approx 5$ and treat the remaining eigenvalue mass as an extended spike; but this visually gives a terrible MP fit (not shown). Selecting $\lambda^+ \approx 10$ produces an MP fit with a reasonable shape to the envelope of support of the bulk; but this fit strongly over-estimates the bulk variance / Frobenius mass (in particular near $\lambda \approx 5$), and it strongly under-estimates the spike near 0. We expect this fit would fail any reasonable statistical confidence test for an MP distribution. As in all cases, numerous *Spikes* extend all the way out to $\lambda_{max} \approx 30$, showing a longer, heavier tail than any MP fit. It is unclear whether or not the edge statistics are TW. There is no good MP fit for the ESD of L226, but it is unclear whether this distribution is "truly" Heavy-Tailed or simply appears Heavy-Tailed as a result of the bimodality. Visually, at least the envelope of the L226 ESD to resembles a *Heavy-Tailed* MP distribution. It is also possible that the DNN itself is also not fully optimized, and we hypothesize that further refinements could lead to a true Heavy-Tailed ESD.

**L302: Bimodal fat or Heavy-Tailed ESDs.** Consider next L302 of InceptionV3 (in Figure 11(b)). The ESD for L302 is slightly bimodal (on a log-log plot), but nowhere near as strongly as L226, and we can not visually select any bulk edge $\lambda^+$. The bulk barely fits any MP density; our best attempt is shown. Also, the global ESD the wrong shape; and the MP fit is concave near the edge, where the ESD is convex, illustrating that the edge decays into the tail. For any MP fit, significant eigenvalue mass extends out continuously, forming a long tail extending al the way to $\lambda_{max} \approx 23$. The ESD of L302 resembles that of the Heavy-Tailed FC2 layer of AlexNet, except for the small bimodal structure. These initial observations illustrate that we need a more rigorous approach to make strong statements about the specific kind of distribution (i.e., Pareto vs other Heavy-Tailed) and what Universality class it may lay in. We present an approach to resolve these technical details this in Section 5.5.

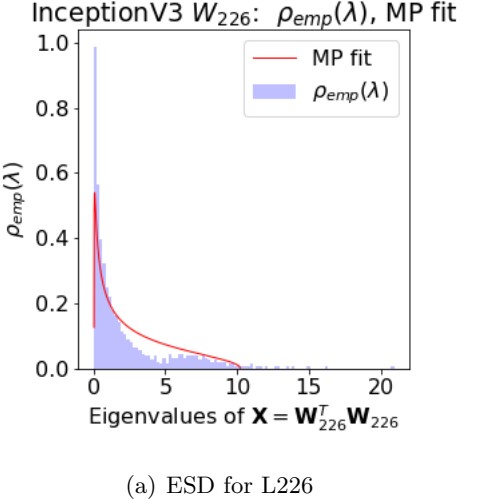
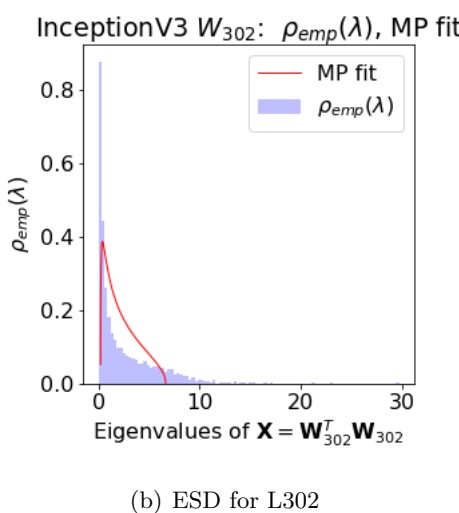

(a) ESD for L226

(b) ESD for L302

Figure 11: ESD for Layers L226 and L302 in InceptionV3, as distributed with pyTorch. Overlaid (in red) are gross fits of the MP distribution (neither of which which fit the ESD well).

## 4.4 Empirical results for other pre-trained DNNs

In addition to the models from Table 4 that we analyzed in detail, we have also examined the properties of a wide range of other pre-trained models, including models from both Computer Vision as well as Natural Language Processing (NLP). This includes models trained on ImageNet, distributed with the pyTorch package, including VGG16, VGG19, ResNet50, InceptionV3, etc. See Table 5. This also includes different NLP models, distributed in AllenNLP [51], including models for Machine Comprehension, Constituency Parsing, Semantic Role Labeling, Coreference Resolution, and Named Entity Recognition, giving a total of 84 linear layers. See Table 6. Rather remarkably, we have observed similar Heavy-Tailed properties, visually and in terms of Power Law fits, in all of these larger, state-of-the-art DNNs, leading to results that are nearly universal across these widely different architectures and domains. We have also seen Hard Rank deficiency in layers in several of these models. We provide a brief summary of those results here.

**Power Law Fits.** We have performed Power Law (PL) fits for the ESD of selected (linear) layers from all of these pre-trained ImageNet and NLP models.[17] Table 5 summarizes the detailed results for the ImageNet models. Several observations can be made. First, all of our fits, except for certain layers in InceptionV3, appear to be in the range $1.5 < \alpha \lesssim 3.5$ (where the CSN method is known to perform well). Second, we also check to see whether PL is the best fit by comparing the distribution to a Truncated Power Law (TPL), as well as an exponential, stretch-exponential, and log normal distributions. Column "Best Fit" reports the best distributional fit. In all cases, we find either a PL or TPL fits best (with a p-value $\leq 0.05$), with TPL being more common for smaller values of $\alpha$. Third, even when taking into account the large finite-size effects in the range $2 < \alpha < 4$, as illustrated in Figure 6, nearly all of the ESDs appear to fall into the $2 < \mu < 4$ Universality class. Figure 12 displays the distribution of PL exponents $\alpha$ for each set of models. Figure 12(a) shows the fit power law exponents $\alpha$ for all of the linear layers in pre-trained ImageNet models available in PyTorch (in Table 5), with $Q \gg 1$; and Figure 12(b) shows the same for the pre-trained models available in AllenNLP (in Table 6). Overall, there are 24 ImageNet layers with $Q \gg 1$, and 82 AllenNet FC layers. More than 80% of all the layers have $\alpha \in [2, 4]$, and nearly all of the rest have $\alpha < 6$. One of these, InceptionV3, was discussed above, precisely since it was unusual, leading to an anomalously large value of $\alpha$ due to the dip in its ESD.

**Rank Collapse.** RMT also predicts that for matrices with $Q > 1$, the minimum singular value will be greater than zero, i.e., $\nu_{min} > 0$. We test this by again looking at all of the FC layers in the pre-trained ImageNet and AllenNLP models. See Figure 13 for a summary of the results. While the ImageNet models mostly follow this rule, 6 of the 24 of FC layers have $\nu_{min} \sim 0$. In fact, for 4 layers, $\nu_{min} < 0.00001$, i.e., it is close to the numerical threshold for 0. In these few cases, the ESD still exhibits Heavy-Tailed properties, but the rank loss ranges from one eigenvalue equal to 0 up to 15% of the eigenvalue mass. For the NLP models, we see no rank collapse, i.e., all of the 82 AllenNLP layers have $\nu_{min} > 0$.

## 4.5 Towards a theory of Self-Regularization

In a few cases (e.g., LetNet5 in Section 4.1), MP theory appears to apply to the bulk of the ESD, with only a few outlying eigenvalues larger than the bulk edge. In other more realistic cases (e.g., AlexNet and InceptionV3 in Sections 4.2 and 4.3, respectively, and every other large-scale DNN

---

[17]We use the default method (MLE, for continuous distributions), and we set $xmax = \lambda_{max}$, i.e., to the maximum eigenvalue of the ESD.

| Model | Layer | Q | $(M \times N)$ | $\alpha$ | D | Best Fit |
|---|---|---|---|---|---|---|
| alexnet | 17/FC1 | 2.25 | $(4096 \times 9216)$ | 2.29 | 0.0527 | PL |
| | 20/FC2 | 1 | $(4096 \times 4096)$ | 2.25 | 0.0372 | PL |
| | 22/FC3 | 4.1 | $(1000 \times 4096)$ | 3.02 | 0.0186 | PL |
| densenet121 | 432 | 1.02 | $(1000 \times 1024)$ | 3.32 | 0.0383 | PL |
| densenet121 | 432 | 1.02 | $(1000 \times 1024)$ | 3.32 | 0.0383 | PL |
| densenet161 | 572 | 2.21 | $(1000 \times 2208)$ | 3.45 | 0.0322 | PL |
| densenet169 | 600 | 1.66 | $(1000 \times 1664)$ | 3.38 | 0.0396 | PL |
| densenet201 | 712 | 1.92 | $(1000 \times 1920)$ | 3.41 | 0.0332 | PL |
| inception v3 | L226 | 1.3 | $(768 \times 1000)$ | 5.26 | 0.0421 | PL |
| | L302 | 2.05 | $(1000 \times 2048)$ | 4.48 | 0.0275 | PL |
| resnet101 | 286 | 2.05 | $(1000 \times 2048)$ | 3.57 | 0.0278 | PL |
| resnet152 | 422 | 2.05 | $(1000 \times 2048)$ | 3.52 | 0.0298 | PL |
| resnet18 | 67 | 1.95 | $(512 \times 1000)$ | 3.34 | 0.0342 | PL |
| resnet34 | 115 | 1.95 | $(512 \times 1000)$ | 3.39 | 0.0257 | PL |
| resnet50 | 150 | 2.05 | $(1000 \times 2048)$ | 3.54 | 0.027 | PL |
| vgg11 | 24 | 6.12 | $(4096 \times 25088)$ | 2.32 | 0.0327 | PL |
| | 27 | 1 | $(4096 \times 4096)$ | 2.17 | 0.0309 | TPL |
| | 30 | 4.1 | $(1000 \times 4096)$ | 2.83 | 0.0398 | PL |
| vgg11 bn | 32 | 6.12 | $(4096 \times 25088)$ | 2.07 | 0.0311 | TPL |
| | 35 | 1 | $(4096 \times 4096)$ | 1.95 | 0.0336 | TPL |
| | 38 | 4.1 | $(1000 \times 4096)$ | 2.99 | 0.0339 | PL |
| vgg16 | 34 | 6.12 | $(4096 \times 25088)$ | 2.3 | 0.0277 | PL |
| | 37 | 1 | $(4096 \times 4096)$ | 2.18 | 0.0321 | TPL |
| | 40 | 4.1 | $(1000 \times 4096)$ | 2.09 | 0.0403 | TPL |
| vgg16 bn | 47 | 6.12 | $(4096 \times 25088)$ | 2.05 | 0.0285 | TPL |
| | 50 | 1 | $(4096 \times 4096)$ | 1.97 | 0.0363 | TPL |
| | 53 | 4.1 | $(1000 \times 4096)$ | 3.03 | 0.0358 | PL |
| vgg19 | 40 | 6.12 | $(4096 \times 25088)$ | 2.27 | 0.0247 | PL |
| | 43 | 1 | $(4096 \times 4096)$ | 2.19 | 0.0313 | PL |
| | 46 | 4.1 | $(1000 \times 4096)$ | 2.07 | 0.0368 | TPL |
| vgg19 bn | 56 | 6.12 | $(4096 \times 25088)$ | 2.04 | 0.0295 | TPL |
| | 59 | 1 | $(4096 \times 4096)$ | 1.98 | 0.0373 | TPL |
| | 62 | 4.1 | $(1000 \times 4096)$ | 3.03 | 0.035 | PL |

Table 5: Fit of PL exponents for the ESD of selected (2D Linear) layer weight matrices $\mathbf{W}_l$ in pre-trained models distributed with pyTorch. Layer is identified by the enumerated id of the pyTorch model; $Q = N/M \geq 1$ is the aspect ratio; $(M \times N)$ is the shape of $\mathbf{W}_l^T$; $\alpha$ is the PL exponent, fit using the numerical method described in the text; $D$ is the Komologrov-Smirnov distance, measuring the goodness-of-fit of the numerical fitting; and "Best Fit" indicates whether the fit is better described as a PL (Power Law) or TPL (Truncated Power Law) (no fits were found to be better described by Exponential or LogNormal).

| Model | Problem Domain | # Linear Layers |
|-------|----------------|-----------------|
| MC | Machine Comprehension | 16 |
| CP | Constituency Parsing | 17 |
| SRL | Semantic Role Labeling | 32 |
| COREF | Coreference Resolution | 4 |
| NER | Named Entity Recognition | 16 |

Table 6: Allen NLP Models and number of Linear Layers per model examined.

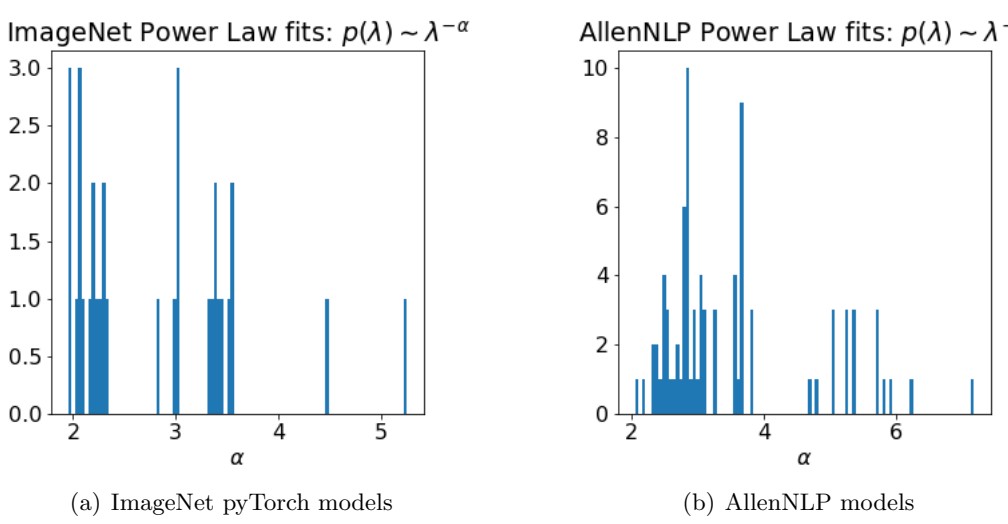

(a) ImageNet pyTorch models                    (b) AllenNLP models

Figure 12: Distribution of power law exponents $\alpha$ for linear layers in pre-trained models trained on ImageNet, available in pyTorch, and for those NLP models, available in AllenNLP.

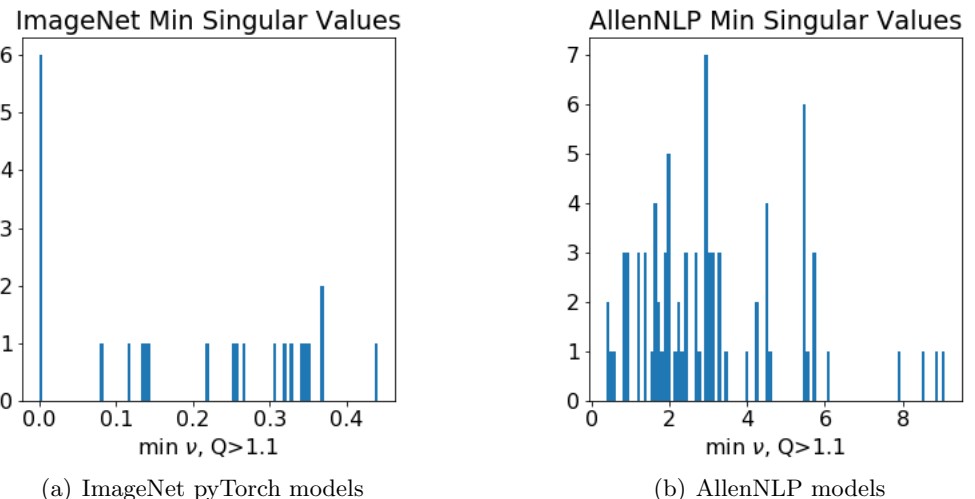

(a) ImageNet pyTorch models                    (b) AllenNLP models

Figure 13: Distribution of minimum singular values $\nu_{min}$ for linear layers in pre-trained models trained on ImageNet, available in pyTorch, and for those NLP models, available in AllenNLP.

we have examined, as summarized in Section 4.4), the ESDs do *not* resemble anything predicted

by standard RMT/MP theory. This should not be unexpected—a well-trained DNN should have highly non-random, strongly-correlated weight matrices $\mathbf{W}$, in which case MP theory would not seem to apply. Moreover, except for InceptionV3, which was chosen to illustrate several unusual properties, nearly every DNN displays Heavy-Tailed properties such as those seen in AlexNet.

These empirical results suggest the following: first, that we can construct an operational and phenomenological theory (both to obtain fundamental insights into DNN regularization and to help guide the training of very large DNNs); and second, that we can build this theory by applying the full machinery of modern RMT to characterize the state of the DNN weight matrices.

For older and/or smaller models, like LeNet5, the *bulk* of their ESDs ($\rho_N(\lambda)$; $\lambda \ll \lambda^+$) can be well-fit to theoretical MP density $\rho_{mp}(\lambda)$, potentially with several distinct, outlying *spikes* ($\lambda > \lambda^+$). This is consistent with the Spiked-Covariance model of Johnstone [68], a simple per-turbative extension of the standard MP theory.[18] This is also reminiscent of traditional Tikhonov regularization, in that there is a "size scale" ($\lambda^+$) separating signal (spikes) from noise (bulk). In this sense, the small NNs of yesteryear—and smallish models used in many research studies—may in fact behave more like traditional ML models. In the context of disordered systems theory, as developed by Sornette [92], this model is a form of *Self-Organizaton*. Putting this all together demonstrates that the DNN training process itself engineers a form of implicit *Self-Regularization* into the trained model.

For large, deep, state-of-the-art DNNs, our observations suggest that there are profound deviations from traditional RMT. These networks are reminiscent of strongly-correlated disordered-systems that exhibit Heavy-Tailed behavior. What is this regularization, and how is it related to our observations of implicit Tikhonov-like regularization on LeNet5?

To answer this, recall that similar behavior arises in strongly-correlated physical systems, where it is known that strongly-correlated systems can be *modeled* by random matrices—with entries drawn from non-Gaussian Universality classes [134], e.g., PL or other Heavy-Tailed distri-butions. Thus, when we observe that $\rho_N(\lambda)$ has Heavy-Tailed properties, we can hypothesize that $\mathbf{W}$ is strongly-correlated,[19] and we can model it with a Heavy-Tailed distribution. Then, upon closer inspection, we find that the ESDs of large, modern DNNs behave as expected—when using the lens of Heavy-Tailed variants of RMT. Importantly, unlike the Spiked-Covariance case, which has a scale cut-off ($\lambda^+$), in these very strongly Heavy-Tailed cases, correlations appear on every size scale, and we can not find a clean separation between the MP bulk and the spikes. These observations demonstrate that modern, state-of-the-art DNNs exhibit a new form of *Heavy-Tailed Self-Regularization*.

In the next few sections, we construct and test (on miniature AlexNet) our new theory.

## 5    5+1 Phases of Regularized Training

In this section, we develop an operational and phenomenological theory for DNN Self-Regularization that is designed to address questions such as the following. How does DNN Self-Regularization differ between older models like LetNet5 and newer models like AlexNet or Inception? What happens to the Self-Regularization when we adjust the numerous knobs and switches of the solver itself during SGD/Backprop training? How are knobs, e.g., early stopping, batch size, and learning rate, related to more familiar regularizers like Weight Norm constraints and Tikhonov regularization? Our theory builds on empirical results from Section 4; and our theory has conse-quences and makes predictions that we test in Section 6.

---

[18]It is also consistent with Heavy-Tailed extensions of MP theory, for larger values of the PL exponent.

[19]For DNNs, these correlations arise in the weight matrices during Backprop training (at least when training on data of reasonable-quality). That is, the weight matrices "learn" the correlations in the data.

**MP Soft Rank.** We first define a metric, the *MP Soft Rank* ($\mathcal{R}_{mp}$), that is designed to capture the "size scale" of the noise part of the layer weight matrix $\mathbf{W}_l$, relative to the largest eigenvalue of $\mathbf{W}_l^T \mathbf{W}_l$. Going beyond spectral methods, this metric exploits MP theory in an essential way.

Let's first assume that MP theory fits *at least a bulk* of $\rho_N(\lambda)$. Then, we can identify a bulk edge $\lambda^+$ and a bulk variance $\sigma_{bulk}^2$, and define the *MP Soft Rank* as the ratio of $\lambda^+$ and $\lambda_{max}$:

$$\text{MP Soft Rank}: \quad \mathcal{R}_{mp}(\mathbf{W}) := \frac{\lambda^+}{\lambda_{max}}. \tag{11}$$

Clearly, $\mathcal{R}_{mp} \in [0, 1]$; $\mathcal{R}_{mp} = 1$ for a purely random matrix (as in Section 5.1); and for a matrix with an ESD with outlying spikes (as in Section 5.3), $\lambda_{max} > \lambda^+$, and $\mathcal{R}_{mp} < 1$. If there is no good MP fit because the entire ESD is well-approximated by a Heavy-Tailed distribution (as described in Section 5.5, e.g., for a strongly correlated weight matrix), then we can define $\lambda^+ = 0$ and still use Eqn. (11), in which case $\mathcal{R}_{mp} = 0$.

The MP Soft Rank is interpreted differently than the Stable Rank ($\mathcal{R}_s$), which is proportional to the bulk MP variance $\sigma_{mp}^2$ divided by $\lambda_{max}$:

$$\mathcal{R}_s(\mathbf{W}) \propto \frac{\sigma_{mp}^2}{\lambda_{max}}. \tag{12}$$

As opposed to the Stable Rank, the MP Soft Rank is defined in terms of the MP distribution, and it depends on how the bulk of the ESD is fit. While the Stable Rank $\mathcal{R}_s(\mathbf{M})$ indicates how many eigencomponents are necessary for a relatively-good low-rank approximation of an arbitrary matrix, the MP Soft Rank $\mathcal{R}_{mp}(\mathbf{W})$ describes how well MP theory fits part of the matrix ESD $\rho_N(\lambda)$. Empirically, $\mathcal{R}_s$ and $\mathcal{R}_{mp}$ often correlate and track similar changes. Importantly, though, there may be no good low-rank approximation of the layer weight matrices $\mathbf{W}_l$ of a DNN—especially a well trained one.

**Visual Taxonomy.** We characterize *implicit Self-Regularization*, both for DNNs during SGD training as well as for *pre-trained* DNNs, as a visual taxonomy of *5+1 Phases of Training* (RANDOM-LIKE, BLEEDING-OUT, BULK+SPIKES, BULK-DECAY, HEAVY-TAILED, and RANK-COLLAPSE). See Table 7 for a summary. The 5+1 phases can be ordered, with each successive phase corresponding to a smaller Stable Rank / MP Soft Rank and to progressively more Self-Regularization than previous phases. Figure 14 depicts typical ESDs for each phase, with the MP fits (in red). Earlier phases of training correspond to the final state of older and/or smaller models like LeNet5 and MLP3. Later phases correspond to the final state of more modern models like AlexNet, Inception, etc. Thus, while we can describe this in terms of SGD training, this taxonomy does not just apply to the temporal ordering given by the training process. It also allows us to compare different architectures and/or amounts of regularization in a trained—or even pre-trained—DNN.

Each phase is visually distinct, and each has a natural interpretation in terms of RMT. One consideration is the *global properties of the ESD*: how well all or part of the ESD is fit by an MP distribution, for some value of $\lambda^+$, or how well all or part of the ESD is fit by a Heavy-Tailed or PL distribution, for some value of a PL parameter. A second consideration is *local properties of the ESD*: the form of fluctuations, in particular around the edge $\lambda^+$ or around the largest eigenvalue $\lambda_{max}$. For example, the shape of the ESD near to and immediately above $\lambda^+$ is very different in Figure 14(a) and Figure 14(c) (where there is a crisp edge) versus Figure 14(b) (where the ESD is concave) versus Figure 14(d) (where the ESD is convex). Gaussian-based RMT (when elements are drawn from the Gaussian Universality class) versus Heavy-Tailed RMT (when elements are drawn from a Heavy-Tailed Universality class) provides guidance, as we describe below.

| | Operational Definition | Informal Description via Eqn. (13) | Edge/tail Fluctuation Comments | Illustration and Description |
|---|---|---|---|---|
| RANDOM-LIKE | ESD well-fit by MP with appropriate $\lambda^+$ | $\mathbf{W}^{rand}$ random; $\|\Delta^{sig}\|$ zero or small | $\lambda_{max} \approx \lambda^+$ is sharp, with TW statistics | Fig. 14(a) and Sxn. 5.1 |
| BLEEDING-OUT | ESD RANDOM-LIKE, excluding eigenmass just above $\lambda^+$ | $\mathbf{W}$ has eigenmass at bulk edge as spikes "pull out"; $\|\Delta^{sig}\|$ medium | BPP transition, $\lambda_{max}$ and $\lambda^+$ separate | Fig. 14(b) and Sxn. 5.2 |
| BULK+SPIKES | ESD RANDOM-LIKE plus $\geq 1$ spikes well above $\lambda^+$ | $\mathbf{W}^{rand}$ well-separated from low-rank $\Delta^{sig}$; $\|\Delta^{sig}\|$ larger | $\lambda^+$ is TW, $\lambda_{max}$ is Gaussian | Fig. 14(c) and Sxn. 5.3 |
| BULK-DECAY | ESD less RANDOM-LIKE; Heavy-Tailed eigenmass above $\lambda^+$; some spikes | Complex $\Delta^{sig}$ with correlations that don't fully enter spike | Edge above $\lambda^+$ is not concave | Fig. 14(d) and Sxn. 5.4 |
| HEAVY-TAILED | ESD better-described by Heavy-Tailed RMT than Gaussian RMT | $\mathbf{W}^{rand}$ is small; $\Delta^{sig}$ is large and strongly-correlated | No good $\lambda^+$; $\lambda_{max} \gg \lambda^+$ | Fig. 14(e) and Sxn. 5.5 |
| RANK-COLLAPSE | ESD has large-mass spike at $\lambda = 0$ | $\mathbf{W}$ very rank-deficient; over-regularization | — | Fig. 14(f) and Sxn. 5.6 |

Table 7: The 5+1 phases of learning we identified in DNN training. We observed BULK+SPIKES and HEAVY-TAILED in existing trained models (LeNet5 and AlexNet/InceptionV3, respectively; see Section 4); and we exhibited all 5+1 phases in a simple model (MiniAlexNet; see Section 7).

As an illustration, Figure 15 depicts the 5+1 phases for a typical (hypothetical) run of Backprop training for a modern DNN. Figure 15(a) illustrates that we can track the decrease in MP Soft Rank, as $\mathbf{W}_l^e$ changes from an initial random (Gaussian-like) matrix to its final $\mathbf{W}_l = \mathbf{W}_l^f$ form; and Figure 15(b) illustrates that (at least for the early phases) we can fit its ESD (or the bulk of its ESD) using MP theory, with $\Delta$ corresponding to non-random signal eigendirections. Observe that there are eigendirections (below $\lambda^+$) that fit very well the MP bulk, there are eigendirections (well above $\lambda^+$) that correspond to a spike, and there are eigendirections (just slightly above $\lambda^+$) with (convex) curvature more like Figure 14(d) than (the concave curvature of) Figure 14(b). Thus, Figure 15(b) is BULK+SPIKES, with early indications of BULK-DECAY.

**Theory of Each Phase.** RMT provides more than simple visual insights, and we can use RMT to differentiate between the *5+1 Phases of Training* using simple models that qualitatively describe the shape of each ESD. In each phase, we model the weight matrices $\mathbf{W}$ as "noise plus signal," where the "noise" is modeled by a random matrix $\mathbf{W}^{rand}$, with entries drawn from the Gaussian Universality class (well-described by traditional MP theory) and the "signal" is a (small or very large) correction $\Delta^{sig}$:

$$\mathbf{W} \simeq \mathbf{W}^{rand} + \Delta^{sig}. \tag{13}$$

Table 7 summarizes the theoretical model for each phase. Each model uses RMT to describe the global shape of $\rho_N(\lambda)$, the local shape of the fluctuations at the bulk edge, and the statistics and information in the outlying spikes, including possible Heavy-Tailed behaviors.

In the first phase (RANDOM-LIKE), the ESD is well-described by traditional MP theory, in which a random matrix has entries drawn from the Gaussian Universality class. This does not

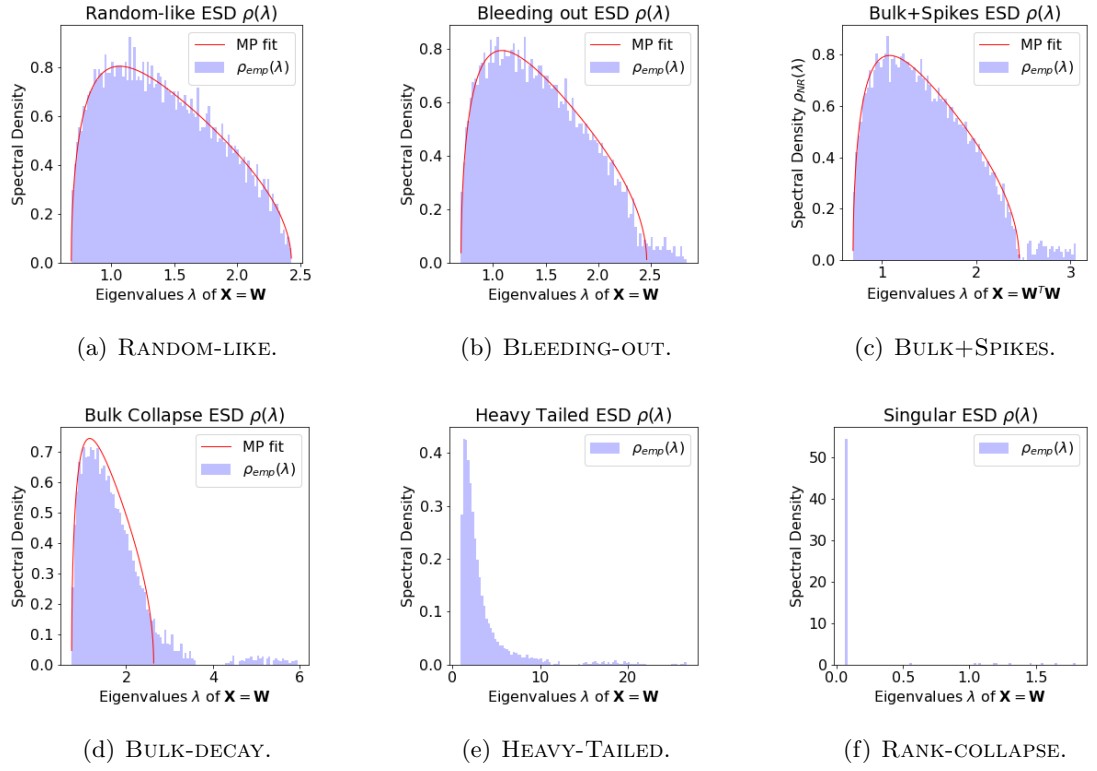

(a) RANDOM-LIKE.  (b) BLEEDING-OUT.  (c) BULK+SPIKES.

(d) BULK-DECAY.  (e) HEAVY-TAILED.  (f) RANK-COLLAPSE.

Figure 14: Taxonomy of trained models. Starting off with an initial random or RANDOM-LIKE model (14(a)), training can lead to a BULK+SPIKES model (14(c)), with data-dependent spikes on top of a random-like bulk. Depending on the network size and architecture, properties of training data, etc., additional training can lead to a HEAVY-TAILED model (14(e)), a high-quality model with long-range correlations. An intermediate BLEEDING-OUT model (14(b)), where spikes start to pull out from the bulk, and an intermediate BULK-DECAY model (14(d)), where correlations start to degrade the separation between the bulk and spikes, leading to a decay of the bulk, are also possible. In extreme cases, a severely over-regularized model (14(f)) is possible.

mean that the weight matrix $\mathbf{W}$ is random, but it does mean that the signal in $\mathbf{W}$ is too weak to be seen when viewed via the lens of the ESD. In the next phases (BLEEDING-OUT, BULK+SPIKES), and/or for small networks such as LetNet5, $\Delta$ is a relatively-small perturbative correction to $\mathbf{W}^{rand}$, and vanilla MP theory (as reviewed in Section 3.1) can be applied, as least to the bulk of the ESD. In these phases, we will *model* the $\mathbf{W}^{rand}$ matrix by a vanilla $\mathbf{W}_{mp}$ matrix (for appropriate parameters), and the MP Soft Rank is relatively large ($\mathcal{R}_{mp}(\mathbf{W}) \gg 0$). In the BULK+SPIKES phase, the model resembles a Spiked-Covariance model, and the Self-Regularization resembles Tikhonov regularization.

In later phases (BULK-DECAY, HEAVY-TAILED), and/or for modern DNNs such as AlexNet and InceptionV3, $\Delta$ becomes more complex and increasingly dominates over $\mathbf{W}^{rand}$. For these more strongly-correlated phases, $\mathbf{W}^{rand}$ is relatively much weaker, and the MP Soft Rank collapses ($\mathcal{R}_{mp}(\mathbf{W}) \to 0$). Consequently, vanilla MP theory is not appropriate, and instead the Self-Regularization becomes Heavy-Tailed. In these phases, we will treat the noise term $\mathbf{W}^{rand}$ as small, and we will *model* the properties of $\Delta$ with Heavy-Tailed extensions of vanilla MP theory (as reviewed in Section 3.2) to Heavy-Tailed non-Gaussian universality classes that are more

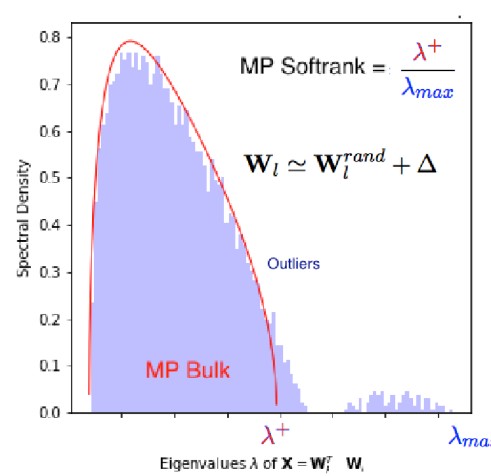

(a) Depiction of how decreasing MP Soft Rank corresponds to different phases of training.

(b) Depiction of each how each phase is related to the MP Soft Rank.

Figure 15: Pictorial illustration of the 5 Phases of Training and the MP Soft Rank

appropriate to model strongly-correlated systems. In these phases, the strongly-correlated model is still regularized, but in a very non-traditional way. The final phase, the RANK-COLLAPSE phase, is a degenerate case that is a prediction of the theory.

We now describe in more detail each phase in turn.

## 5.1 Random-like

In the first phase, the RANDOM-LIKE phase, shown in Figure 14(a), the DNN weight matrices **W** resemble a Gaussian random matrix. The ESDs are easily-fit to an MP distribution, with the same aspect ratio $Q$, by fitting the empirical variance $\sigma^2_{emp}$. Here, $\sigma^2_{emp}$ is the element-wise variance (which depends on the normalization of **W**).

Of course, an initial random weight matrix $\mathbf{W}^0_l$ will show a near perfect MP fit. Even in well trained DNNs, however, the empirical ESDs may be RANDOM-LIKE, even when the model has a non-zero, and even somewhat large, generalization accuracy.[20] That is, being fit well by an MP distribution does *not* imply that the weight matrix **W** is random. It simply implies that **W**, while having structure, can be modeled as the sum of a random "noise" matrix $\mathbf{W}^{rand}$, with the same $Q$ and $\sigma^2_{emp}$, and some small-sized matrix $\Delta^{small}$, as:

$$\mathbf{W} \simeq \mathbf{W}^{rand} + \Delta^{small},$$

where $\Delta^{small}$ represents "signal" learned during the training process. In this case, $\lambda_{max}$ is sharply bounded, to within $M^{-\frac{2}{3}}$, to the edge of the MP distribution.

## 5.2 Bleeding-out

In the second phase, the BLEEDING-OUT phase, shown in Figure 14(b), the bulk of the ESD still looks reasonably random, except for one or a small number $K \ll \min\{N, M\}$ of eigenvalues that extend *at or just beyond* the MP edge $\lambda^+$. That is, for the given value of $Q$, we can choose a $\sigma_{emp}$

---

[20]In particular, as described below, we observe this with toy models when trained with very large batch sizes.

(or $\lambda^+$) parameter so that: (1) most of the ESD is well-fit; and (2) the part of the ESD that is not well-fit consists of a "shelf" of mass, much more than expected by chance, just above $\lambda^+$:

$$[\lambda_1, \lambda_2 \cdots \lambda_K] \approx \lambda^+ + \mathcal{O}(M^{-\frac{2}{3}}) \gtrsim \lambda^+, \ \ \lambda_k \in \text{Bleeding-out}.$$

This corresponds to modeling $\mathbf{W}$ as the sum of a random "noise" matrix $\mathbf{W}^{rand}$ and some medium-sized matrix $\Delta^{medium}$, as:

$$\mathbf{W} \simeq \mathbf{W}^{rand} + \Delta^{medium},$$

where $\Delta^{medium}$ represents "signal" learned during the training process.

As the spikes just begin to pull out from the bulk, i.e., when $\|\lambda_{max} - \lambda^+\|$ is small, it may be difficult to determine unambiguously whether any particular eigenvalue is spike or bulk. The reason is that, since the matrix is of finite size, we expect the spike locations to be Gaussian-distributed, with fluctuations of order $N^{-\frac{1}{2}}$. One option is to try to estimate $\sigma_{bulk}$ precisely from a single run. Another option is to perform an ensemble of runs and plot $\rho_{N_R}(\lambda)$ for the ensemble. Then, if the model is in the BLEEDING-OUT phase, there will be a small bump of eigenvalue mass, shaped like a Gaussian,[21] which is very close to but bleeding-out from the bulk edge.

When modeling DNN training in terms of RMT and MP theory, the transition from RANDOM-LIKE to BLEEDING-OUT corresponds to the so-called *BPP phase transition* [9, 23, 45, 20]. This transition represents a "condensation" of the eigenvector corresponding to the largest eigenvalue $\lambda_{max}$ onto the eigenvalue of the rank-one (or, more generally, rank-$k$, if the perturbation is higher rank) perturbation $\Delta$ [23].

### 5.3   Bulk+Spikes

In the third phase, the BULK+SPIKES phase, shown in Figure 14(c), the bulk of the ESD still looks reasonably random, except for one or a small number $K \ll \min\{N, M\}$ of eigenvalues that extend *well beyond* the MP edge $\lambda^+$. That is, for the given value of $Q$, we can choose a $\sigma_{emp}$ (or $\lambda^+$) parameter so that: (1) most of the ESD is well-fit; and (2) the part of the ESD that is not well-fit consists of several ($K$) eigenvalues, or Spikes, that are much larger than $\lambda^+$:

$$[\lambda_1, \lambda_2 \cdots \lambda_K] \gg \lambda^+, \ \ \lambda_k \in \text{Spikes}.$$

This corresponds to modeling $\mathbf{W}$ as the sum of a random "noise" matrix $\mathbf{W}^{rand}$ and some moderately large-sized matrix $\Delta^{large}$, as:

$$\mathbf{W} \simeq \mathbf{W}^{rand} + \Delta^{large},$$

where $\Delta^{large}$ represents "signal" learned during the training process.

For a single run, it may be challenging to identify the spike locations unambiguously. If we perform an ensemble of runs, however, then the Spike density is clearly visible, distinct, and separated from bulk, although it is much smaller in total mass. We can try to estimate $\sigma_{bulk}$ precisely, but in many cases we can select the edge of the bulk, $\lambda^+$, by visual inspection. As in the BLEEDING-OUT phase, the empirical bulk variance $\sigma^2_{bulk}$ is smaller than both the full element-wise variance, $\sigma^2_{bulk} < \sigma^2_{full}$, and the shuffled variance (fit to the MP bulk), $\sigma^2_{bulk} < \sigma^2_{shuf}$, because we remove several large eigendirections from the bulk. (See Section 6.3 for more on this.)

When modeling DNN training in terms of RMT and MP theory, the BULK+SPIKES phase corresponds to vanilla MP theory plus a large low-rank perturbation, and it is what we observe in the LeNet5 model. In statistics, this corresponds to the Spiked Covariance model [68, 109, 69]. Relatedly, in the BULK+SPIKES phase, we see clear evidence of Tikhonov-like *Self-Regularization*.

---

[21]We are being pedantic here since, in later phases, bleeding-out mass will be Heavy-Tailed, not Gaussian.

**MP theory with large low-rank perturbations.** To understand, from the perspective of MP theory, the properties of the ESD as eigenvalues bleed out and start to form spikes, consider modeling $\mathbf{W}$ as $\mathbf{W} \simeq \mathbf{W}^{rand} + \Delta^{large}$. If $\Delta$ is a rank-1 perturbation[22], but now larger, then one can show that the maximum eigenvalue $\lambda_{max}$ that bleeds out will extend beyond theoretical MP bulk edge $\lambda^+$ and is given by

$$\lambda_{max} = \sigma^2 \left( \frac{1}{Q} + \frac{|\Delta|^2}{N} \right) \left( 1 + \frac{N}{|\Delta|^2} \right),$$

where

$$|\Delta| > (NM)^{\frac{1}{4}}.$$

Here, by $\sigma^2$, we mean the theoretical variance of the un-perturbed $\mathbf{W}^{rand}$.[23] Moreover, in an ensemble of runs, each of these Spikes will have Gaussian fluctuations on the order $N^{-1/2}$.

**Eigenvector localization.** Eigenvector localization on extreme eigenvalues can be a diagnostic for Spike eigenvectors (as well as for extreme eigenvectors in the HEAVY-TAILED phase). The interpretation is that when the perturbation $\Delta$ is large, "information" in $\mathbf{W}$ will concentrate on a small number of components of the eigenvectors associated with the outlier eigenvalues.

## 5.4 Bulk-decay

The fourth phase, the BULK-DECAY phase, is illustrated in Figure 14(d), and is characterized by the onset of Heavy-Tailed behavior, both in the very long tail, and at the Bulk edge.[24] The BULK-DECAY phase is intermediate between having a large, low-rank perturbation $\Delta$ to an MP Bulk (as in the BULK+SPIKES phase) and having strong correlations at all scales (as in the HEAVY-TAILED phase). Viewed naïvely, the ESDs in BULK-DECAY resemble a combination of the BLEEDING-OUT and BULK+SPIKES phases: there is a large amount of mass above $\lambda^+$ (from any reasonable MP fit); and there are a large number of eigenvectors much larger than this value of $\lambda^+$. However, quantitatively, the ESDs are quite different than either BLEEDING-OUT or BULK+SPIKES: there is much more mass bleeding-out; there is much greater deterioration of the Bulk; and the Spikes lie much farther out.

In BULK-DECAY, the Bulk region is both hard to identify and difficult to fit with MP theory. Indeed, the properties of the Bulk start to look less and less consistent the an MP distribution (with elements drawn from the Universality class of Gaussian matrices), for any parameter values. This implies that $\lambda_{max}$ can be quite large, in which case the MP Soft Rank is much smaller. The best MP fit neglects a large part of the eigenvalue mass, and so we usually have to select $\lambda^+$ numerically. Most importantly, the mass at the bulk edge now starts to exhibit Heavy-Tailed, not Gaussian, properties; and the overall shape of the ESD is itself taking on a HEAVY-TAILED form. Indeed, the ESDs are may be consistent with (weakly) Heavy-Tailed $(4 < \mu)$ Universality class, in which the local edge statistics exhibit Heavy-Tailed behavior due to finite-size effects.[25]

---

[22] More generally, $\Delta$ may be a rank-$k$ perturbation, for $k \ll M$, and similar results should hold.

[23] In typical theory, this is scaled to unity (i.e., $\sigma^2 = 1$). In typical practice, we do not *a priori* know $\sigma^2$, and it may be non-trivial to estimate because the scale $\|\mathbf{W}\|$ may shift during Backprop training. As a rule-of-thumb, one can select the bulk edge $\lambda^+$ well to provide a good fit for the bulk variance $\sigma^2_{bulk}$.

[24] We observe BULK-DECAY in InceptionV3 (Figure 11(b)). This may indicate that this model, while extremely good, might actually lend itself to more fine tuning and might not be fully optimized.

[25] Making this connection more precise—e.g., measuring $\alpha$ in this regime, relating $\alpha$ to $\mu$ in this regime, having precise theory for finite-size effects in this regime, etc.—is nontrivial and left for future work.

## 5.5 Heavy-Tailed

The final of the 5 main phases, the HEAVY-TAILED phase, is illustrated in Figure 14(e). This phase is formally, and operationally, characterized by an ESD that resembles the ESD of a random matrix in which the entries are drawn i.i.d. from a Heavy-Tailed distribution. This phase corresponds to modeling $\mathbf{W}$ as the sum of a small "noise" matrix $\mathbf{W}^{rand}$ and a large "strongly-correlated" matrix $\Delta^{str.corr.}$, as:

$$\mathbf{W} \simeq \mathbf{W}^{rand} + \Delta^{str.corr.}$$

where $\Delta^{str.corr.}$ represents strongly-correlated "signal" learned during the training process.[26] As usual, $\mathbf{W}^{rand}$ can be modeled as a random matrix, with entries drawn i.i.d. from a distribution in the Gaussian Universality class. Importantly, the strongly-correlated signal matrix $\Delta^{str.corr.}$ can also be modeled as a random matrix, but (as described in Section 3.2) one with entries drawn i.i.d. from a distribution in a different, Heavy-Tailed, Universality class.

In this phase, the ESD visually appears Heavy-Tailed, and it is very difficult if not impossible to get a reasonable MP fit of the layer weight matrices $\mathbf{W}$ (using standard Gaussian-based MP/RMT). Thus, the matrix $\mathbf{W}$ has zero ($\mathcal{R}_{mp}(\mathbf{W}) = 0$) or near-zero ($\mathcal{R}_{mp}(\mathbf{W}) \gtrsim 0$) MP Soft Rank; and it has intermediate Stable Rank ($1 \ll \mathcal{R}_s(\mathbf{W}) \ll \min\{N, M\}$).[27]

When modeling DNN training in terms of RMT and MP theory, the HEAVY-TAILED phase corresponds to the variant of MP theory in which elements are chosen from a non-Gaussian Universality class [38, 20, 19, 111, 7, 40, 8, 26, 22, 21]. In physics, this corresponds to modeling strongly-correlated systems with Heavy-Tailed random matrices [134, 23]. Relatedly, in the HEAVY-TAILED phase, the implicit Self-Regularization is strongest. It is, however, *very* different than the Tikhonov-like regularization seen in the BULK+SPIKES phases. Although there is a decrease in the Stable Rank (for similar reasons to why it decreases in the BULK+SPIKES phases, i.e., Frobenius mass moves out of the bulk and into the spikes), *Heavy-Tailed Self-Regularization* does *not* exhibit a "size scale" in the eigenvalues that separates the signal from the noise.[28]

**Heavy-Tailed ESDs.** Although Figure 14(e) is presented on the same linear-linear plot as the other subfigures in Figure 14, the easiest way to compare Heavy-Tailed ESDs is with a log-log histogram and/or with PL fits. Consider Figure 16(a), which displays the ESD for FC3 of pre-trained AlexNet, as a log-log histogram; and consider also Figure 16(b), which displays an overlay (in red) of a log-log histogram of the ESD of a random matrix $\mathbf{M}$. This matrix $\mathbf{M}$ has the same aspect ratio as $\mathbf{W}_{FC3}$, but the elements $M_{i,j}$ are drawn from a Heavy-Tailed Pareto distribution, Eqn. (10), with $\mu = 2.5$. We call ESDs such as $\mathbf{W}_{FC3}$ of AlexNet Heavy-Tailed because they resemble the ESD of a random matrix with entries drawn from a Heavy-Tailed distribution, as observed with a log-log histogram.[29] We can also do a PL fit to estimate $\alpha$ and then try to estimate the Universality class we are in. Our PL estimator works well for $\mu \in [1.5, 3.5]$; but, due to large finite-size effects, it is difficult to determine $\mu$ from $\alpha$ precisely. This is discussed in more detail in Section 3.2. As a rule of thumb, if $\alpha < 2$, then we can say $\alpha \approx 1 + \mu/2$, and we are in the (very) Heavy-Tailed Universality class; and if $2 < \alpha < 4$, but not too large, then $\alpha$ is well-modeled by $\alpha \approx b + a\mu$, and we are mostly likely in the (moderately, or "fat") Heavy-Tailed Universality class.

---

[26]This BULK+SPIKES phase is what we observe in nearly all pre-trained, production-quality models.

[27]In this phase, the matrix $\mathbf{W}$ *mostly* retains full hard rank ($\mathcal{R}(\mathbf{W}) > 0$); but, in some cases, some small deterioration of hard rank is observed, depending on the numerical threshold used.

[28]Indeed, this is the point of systems that exhibit Heavy-Tailed or scale-free properties.

[29]Recall the discussion around Figure 5.

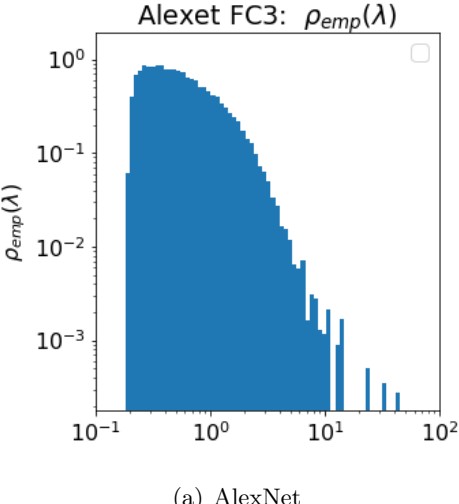
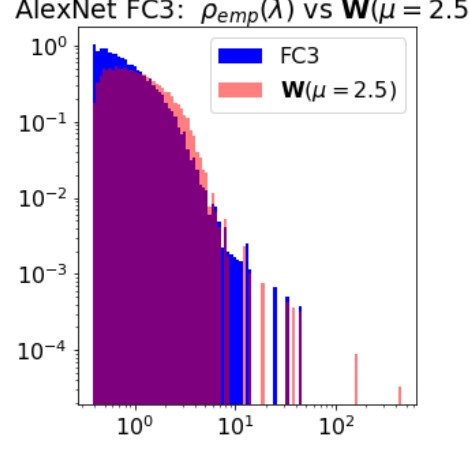

(a) AlexNet

(b) AlexNet and Heavy-Tailed ESD.

Figure 16: log-log histogram plots of the ESD for $\mathbf{W}_{FC3}$ of pre-trained AlexNet (blue) and a Heavy-Tailed random matrix $\mathbf{M}$ with same aspect ratio and $\mu = 2.5$ (red)

## 5.6 Rank-collapse

In addition to the 5 main phases, based on MP theory we also expect the existence of an additional "+1" phase, which we call the RANK-COLLAPSE Phase, and which is illustrated in Figure 14(f). For many parameter settings, the minimum singular value (i.e., $\lambda^-$ in Eqn. (9) for vanilla MP theory) is strictly positive. For certain parameter settings, the MP distribution has a spike at the origin, meaning that there is a non-negligible mass of eigenvalues equal to 0, i.e., the matrix is rank-deficient, i.e., Hard Rank is lost.[30] For vanilla Gaussian-based MP theory, this happens when $Q > 1$, and this phenomenon exists more generally for Heavy-Tailed MP theory.

# 6 Empirical Results: Detailed Analysis on Smaller Models

In this section, we validate and illustrate how to use our theory from Section 5. This involved extensive training and re-training, and thus we used the smaller MiniAlexNet model. Section 6.1 describes the basic setup; Section 6.2 presents several baseline results; Section 6.3 provides some important technical details; and Section 6.4 describes the effect of adding explicit regularization. We postpone discussing the effect of changing batch size until Section 7.

## 6.1 Experimental setup

Here, we describe the basic setup for our empirical evaluation.

**Model Deep Neural Network.** We analyzed MiniAlexNet,[31] a simpler version of AlexNet, similar to the smaller models used in [156], scaled down to prevent overtraining, and trained on CIFAR10. The basic architecture follows the same general design as older NNs such as LeNet5, VGG16, and VGG19. It is illustrated in Figure 17. It consists of two 2D Convolutional layers,

---

[30]This corresponds more to a Truncated-SVD-like regularization than a softer Tikhonov-like regularization.
[31]https://github.com/deepmind/sonnet/blob/master/sonnet/python/modules/nets/alexnet.py

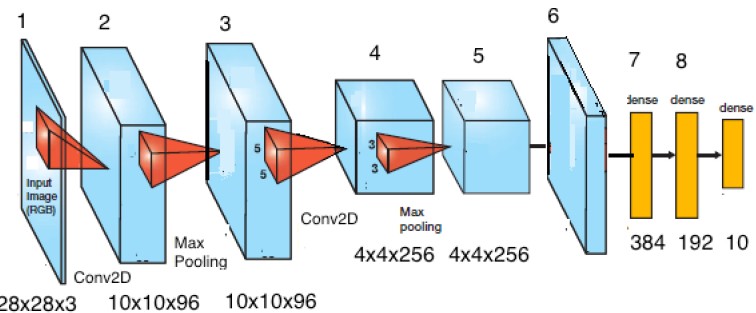

Figure 17: Pictorial illustration of MiniAlexNet.

each with Max Pooling and Batch Normalization, giving 6 initial layers; it then has two Fully Connected (FC), or Dense, layers with ReLU activations; and it then has a final FC layer added, with 10 nodes and softmax activation. For the FC layers:

$$
\begin{aligned}
\mathbf{W}_{FC1} &= (4096 \times 384) \quad \text{(Layer FC1)} \quad (Q \approx 10.67) \\
\mathbf{W}_{FC2} &= (384 \times 192) \quad\ \text{(Layer FC2)} \quad (Q = 2) \\
\mathbf{W}_{FC3} &= (192 \times 10).
\end{aligned}
$$

The $\mathbf{W}_{FC1}$ and $\mathbf{W}_{FC2}$ matrices are initialized with a Glorot normalization [54].[32] We apply Batch Normalization to the Conv2D layers, but we leave it off the FC layer; results do not change if remove all Batch Normalization. All models are trained using Keras 2.x, with TensorFlow as a backend. We use SGD with momentum, with a learning rate of 0.01, a momentum parameter of 0.9, and a baseline batch size of 32; and we train up to 100 epochs. To compare different batch sizes and other tunable knobs, we employed early stopping criteria on the total loss which causes termination at fewer than 100 epochs. We save the weight matrices at the end of every epoch, and we study the complexity of the trained model by analyzing the empirical properties of the $\mathbf{W}_{FC1}$ and $\mathbf{W}_{FC2}$ matrices.

**Experimental Runs.** It is important to distinguish between several different types of analysis. First, analysis of ESDs (and related quantities) during Backprop training during 1 training run. In this case, we consider a single training run, and we monitor empirical properties of weight matrices as they change *during* the training process. Second, analysis of the final ESDs from 1 training run. In this case, we consider a single training run, and we analyze the empirical properties of the single weight matrix that is obtained *after* the training process terminates. This is similar to analyzing pre-trained models. Third, analysis of an ensemble of final ESDs from $N_R$ training runs. In this case, we rerun the model $N_R \sim 10$–100 times, using different initial random weight matrices $\mathbf{W}_l^0$, and we form an ensemble of $N_R$ of final weight matrices $[\mathbf{W}_l^{final}]$. We do this in order to compensate for finite-size effects, to provide a better visual interpretation of our claims, and to help clarify our *scientific* claims about the learning process. Of course, as

---

[32]Here and elsewhere, most DNNs are initialized with random weight matrices, e.g., as with the Glorot Normal initialization [54] (which involves a truncation step). If we naïvely fit the MP distribution to $\mathbf{W}_{trunc}^0$, then the empirical variance will be larger than one, i.e., $\sigma_{emp}^2 > 1$. That is because the Glorot normalization is $\sqrt{2/N + M}$, whereas the MP theory is presented with normalization $\sqrt{N^{-1}}$. To apply MP theory, we must rescale our empirically-fit $\sigma_{emp}^2$ (of the bulk, if there is a spike) to match our normalization, such that the rescaled $\sigma_{emp}^2$ becomes $\sqrt{(M + N)/(2N)}\sigma_{emp}^2$. If we do this, then we find that the Glorot normal matrix has $\sigma_{emp}^2 \lesssim 1$. Even aside from the obvious scale issue of $\sqrt{(M + N)/(2N)}$, this is not perfect—since the variance may shift during SGD/Backprop training, especially if we do not employ Batch Normalization—but it suffices for our purposes.

an *engineering* matter, one wants exploit our results on a single "production" run of the training process. In that case, we expect to observe (and do observe) a noisy version of what we present.[33]

**Empirically Measured Quantities.** We compute several RMT-based quantities of interest for each layer weight matrices $\mathbf{W}_l$, for layers $l = FC1, FC2$, including the following: Matrix complexity metrics, such as the Matrix Entropy $\mathcal{S}(\mathbf{W}_l^e)$, Hard Rank $\mathcal{R}(\mathbf{W}_l^e)$, Stable Rank $\mathcal{R}_s^e(\mathbf{W}_l)$, and MP Soft Rank $\mathcal{R}_{mp}^e(\mathbf{W}_l)$; ESDs, $\rho(\lambda)$ for a single run, both during Backprop training and for the final weight matrices, and/or $\rho_{N_R}(\lambda)$ for the final states an ensemble of $N_R$ runs; and Eigenvector localization metrics, including the Generalized Vector Entropy $\mathcal{S}(\mathbf{x})$, Localization Ratio $\mathcal{L}(\mathbf{x})$, and Participation Ratio $\mathcal{P}(\mathbf{x})$, of the eigenvectors of $\mathbf{X}$, for an ensemble of runs.

**Knobs and Switches of the Learning Process.** We vary knobs and switches of the training process, including the following: number of epochs (typically $\approx 100$, well past when entropies and measured training/test accuracies saturate); Weight Norm regularization (on the fully connected layers—in Keras, this is done with an $L_2$-Weight Norm kernel regularizer, with value 0.0001); various values of Dropout; and batch size[34] (varied between 2 to 1024).

## 6.2 Baseline results

Here, we present several baseline results for our RMT-based analysis of MiniAlexNet. For our baseline, the batch size is 16; and Weight Norm regularization, Dropout, and other explicit forms of regularization are *not* employed.

**Transition in Matrix Entropy and Stable Rank.** Figure 18 shows the Matrix Entropy ($\mathcal{S}(\mathbf{W})$) and Stable Rank ($\mathcal{R}_s(\mathbf{W})$) for layers FC1 and FC2, as well as of the training and test accuracies, for MiniAlexNet, as a function of the number of epochs. This is for an ensemble of $N_R = 10$ runs. Both layers start off with an Entropy close to but slightly less than 1.0; and both retrain full rank during training. For each layer, the matrix Entropy gradually lowers; and the Stable Rank shrinks, but more prominently. These decreases parallel the increase in training and test accuracies, and both complexity metrics level off as the training/test accuracies do. The Matrix Entropy decreases relatively more for FC2, and the Stable Rank decreases relatively more for FC1; but they track the same gross changes. The large difference between training and test accuracy should not be surprising since—for these baseline results—we have turned off regularization like removing Batch Norm, Dropout layers, and any Weight Norm constraints.

**Eigenvalue Spectrum: Comparisons with RMT.** Figures 19 and 20 show, for FC1 and FC2, respectively, the layer matrix ESD, $\rho(\lambda)$, every few epochs during the training process. For layer FC1 (with $Q \approx 10.67$), the initial weight matrix $\mathbf{W}^0$ looks very much like an MP distribution (with $Q \approx 10.67$), consistent with a RANDOM-LIKE phase. Within a very few epochs, however, eigenvalue mass shifts to larger values, and the ESD looks like the BULK+SPIKES phase. Once the Spike(s) appear(s), substantial changes are hard to see in Figure 19, but minor changes do continue in the ESD. Most notably, $\lambda^{max}$ increases from roughly 3.0 to roughly 4.0 during training, indicating further Self-Regularization, even within the BULK+SPIKES phase. For layer FC2 (with $Q = 2$), the initial weight matrix also resembles an MP distribution, also consistent with a RANDOM-LIKE phase, but with a much smaller value of $Q$ than FC1 ($Q = 2$ here). Here

---

[33]We expect that well-established methods from RMT will be useful here, but we leave this for future work.

[34]Generally speaking, decreasing batch size is incompatible with Batch Normalization and can decrease performance because batch statistics are not computed accurately; we have not employed Batch Normalization, except on the Convolutional layers in MiniAlexNet.

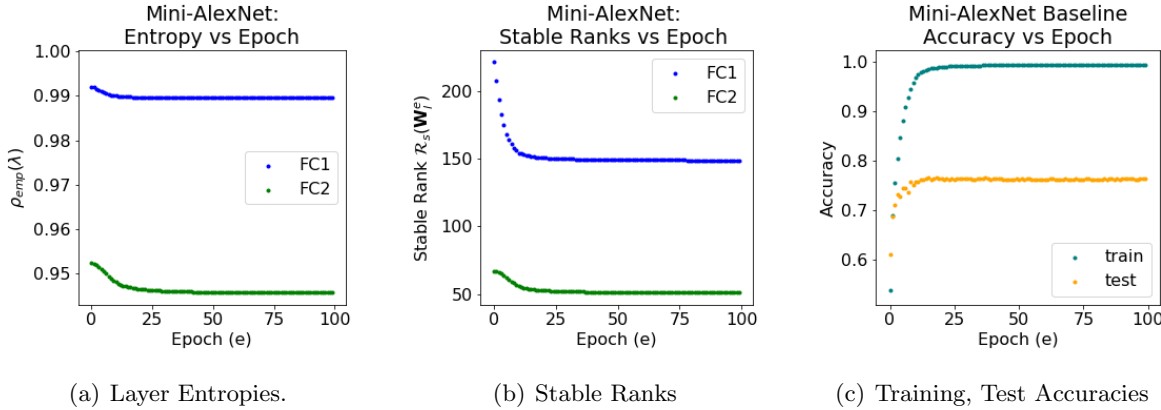

(a) Layer Entropies.     (b) Stable Ranks     (c) Training, Test Accuracies

Figure 18: Entropies, Stable Ranks, and Training and Test Accuracies per Epoch for MiniAlexNet.

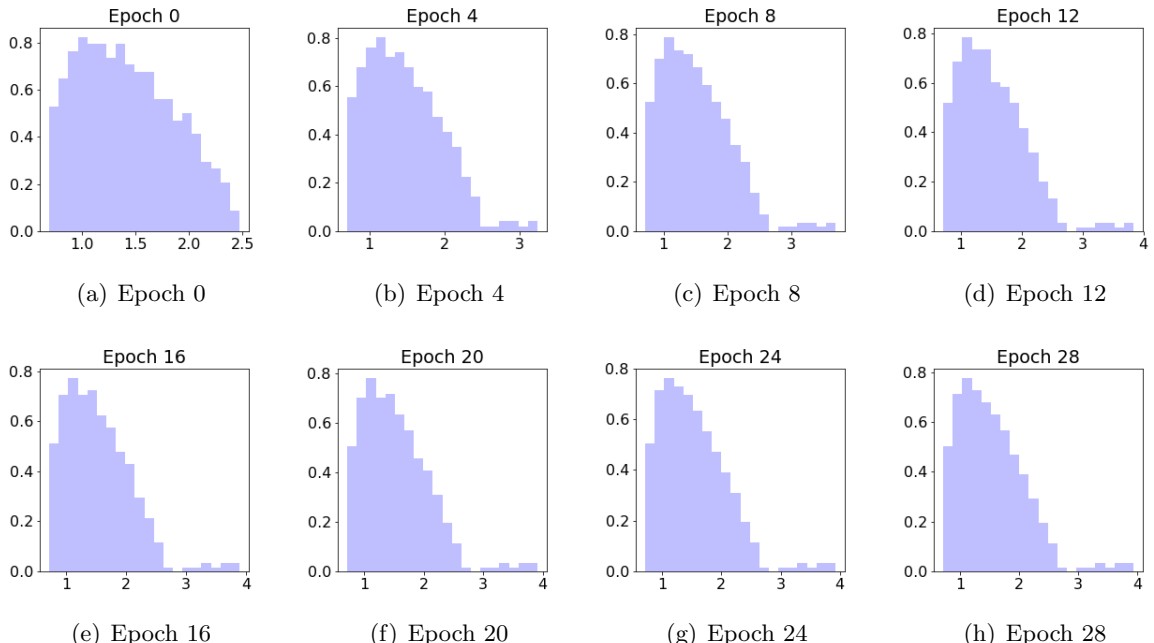

(a) Epoch 0    (b) Epoch 4    (c) Epoch 8    (d) Epoch 12

(e) Epoch 16    (f) Epoch 20    (g) Epoch 24    (h) Epoch 28

Figure 19: Baseline ESD for Layer FC1 of MiniAlexNet, during training.

too, the ESD changes during the first few epochs, after which there are not substantial changes. The most prominent change is that eigenvalue mass pulls out slightly from the bulk and $\lambda^{max}$ increases from roughly 3.0 to slightly less than 4.0.

**Eigenvector localization.**  Figure 21 plots three eigenvector localization metrics, for an ensemble $N_R = 10$ runs, for eigenvectors in the bulk and spike of layer FC1 of MiniAlexNet, after training.[35] Spike eigenvectors tend to be more localized than bulk eigenvectors. This effect is less pronounced for FC2 (not shown) since the spike is less well-separated from the bulk.

---

[35]More precisely, bulk here refers to eigenvectors associated with eigenvalues less than $\lambda^+$, defined below and illustrated in Figure 24, and spike here refers to those in the main part of the spike.

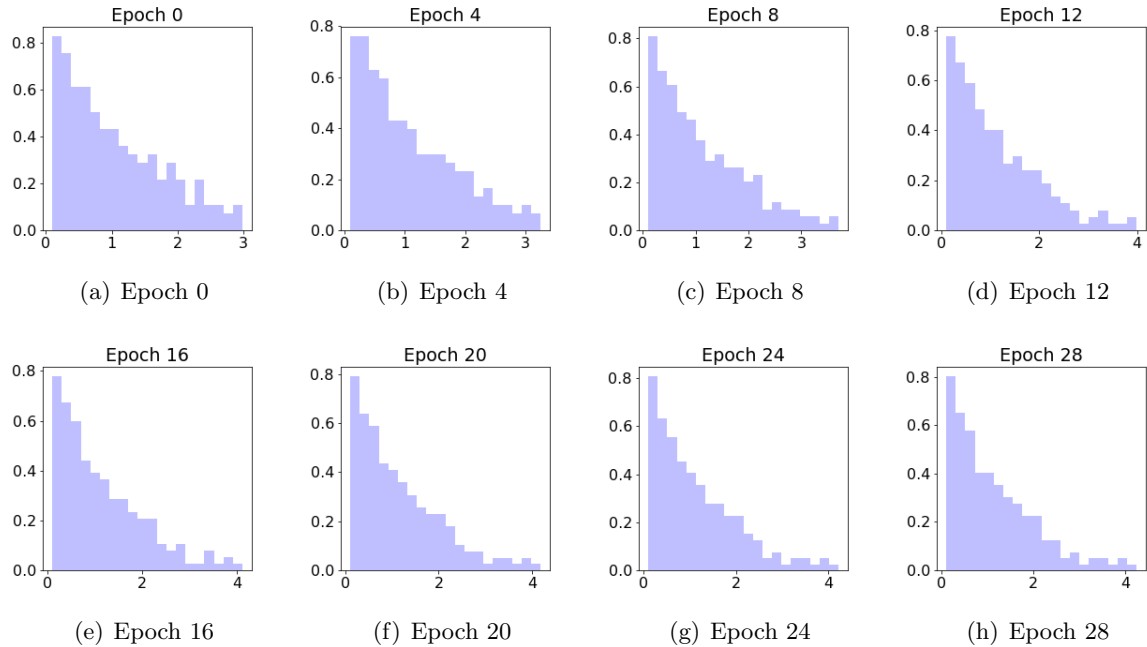

Figure 20: Baseline ESD for Layer FC2 of MiniAlexNet, during training.

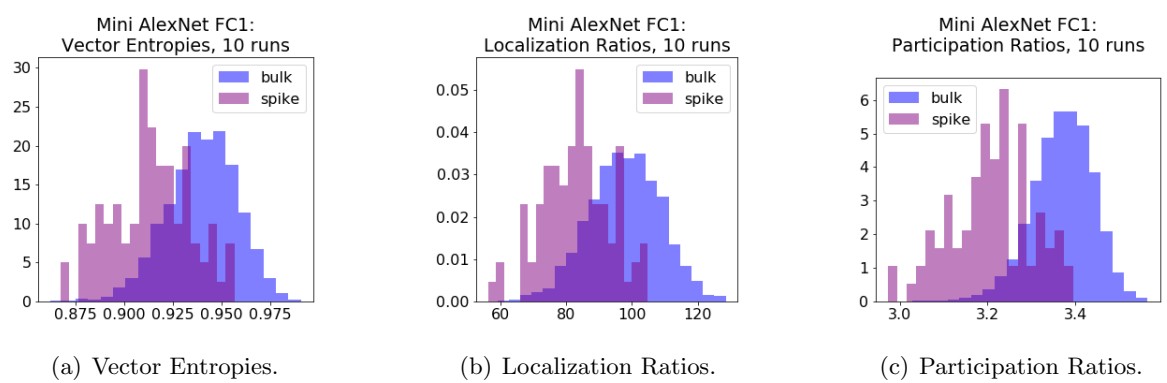

Figure 21: Eigenvector localization metrics for the FC1 layer of MiniAlexNet, for an ensemble of 10 runs. Batch size 16, and no weight regularization. Comparison of Bulk and Spike.

### 6.3 Some important implementational details

There are several technical issues with applying RMT that we discuss here.[36]

- **Single run versus an ensemble of runs.** Figures 22 shows ESDs for Layer FC1 before and after training, for a single run, as well as after training for an ensemble of runs. Figure 23 does the same for FC2. There are two distinct effects of doing an ensemble of runs: first, the histograms get smoother (which is expected); and second, there are fluctuations in $\lambda^{max}$. These fluctuations are *not* due to finite-size effects; and they can exhibit Gaussian or TW or other Heavy-Tailed properties, depending on the phase of learning. Thus, they can be used as a diagnostic, e.g., to differentiate between BULK+SPIKES versus BLEEDING-OUT.

---

[36]While we discuss these in the context of our baseline, the same issues arise in all applications of our theory.

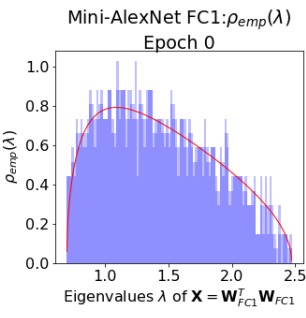
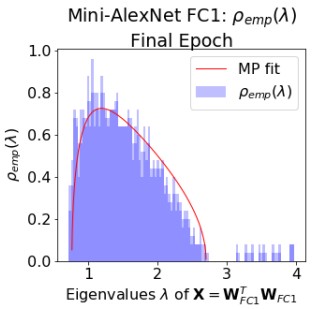
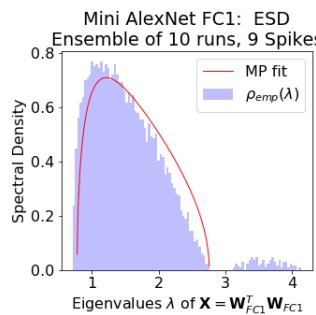

(a) Layer FC1, Epoch 0.

(b) Layer FC1, Final epoch.

(c) Layer FC1, Final epoch; Bulk + 9 Spikes

Figure 22: ESD for Layer FC1 of MiniAlexNet, with MP fit (in red): initial (0) epoch, single run (in 22(a))); final epoch, single run (in 22(b))); final epoch, ensemble of 10 runs (in 22(c))). Batch size 16, and no weight regularization. To get a good MP fit, final epochs have 9 spikes removed from the bulk (but see Figure 24).

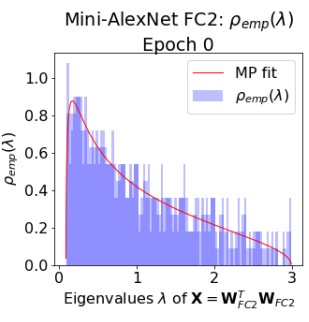
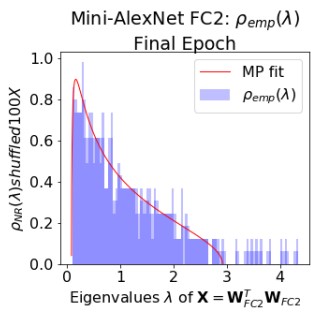
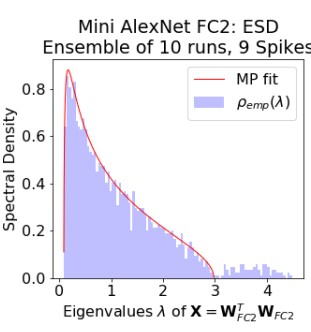

(a) Layer FC2, Epoch 0.

(b) Layer FC2, Final epoch.

(c) Layer FC2, Final epoch.

Figure 23: ESD for Layer FC2 of MiniAlexNet, with MP fit (in red): initial (0) epoch, single run (in 23(a)); final epoch, single run (in 23(b)); final epoch, ensemble of 10 runs (in 23(c)). Batch size 16, and no weight regularization. To get a good MP fit, final epochs have 9 spikes removed from the bulk.

- **Finite-size effects.** Figure 24(a) shows that we can estimate finite-size effects in RMT by shuffling the elements of a single weight matrices $\mathbf{W}_l \to \mathbf{W}_l^{shuf}$ and recomputing the eigenvalue spectrum $\rho^{shuf}(\lambda)$ of $\mathbf{X}^{shuf}$. We expect $\rho^{shuf}(\lambda)$ to fit an MP distribution well, even for small sample sizes, and we see that it does. We also expect and see a very crisp edge in $\lambda^+$. More generally, we can visually observe the quality of the fit at this sample size to gauge whether deviations are likely spurious. This is relatively-easy to do for RANDOM-LIKE, and also for BULK+SPIKES (since we can simply remove the spikes before shuffling). For BLEEDING-OUT and BULK-DECAY, it is somewhat more difficult due to the need to decide which eigenvalues to keep in the bulk. For HEAVY-TAILED, it is much more complicated since finite-size effects are larger and more pronounced.

- **Fitting the bulk edge $\lambda^+$, i.e., the bulk variance $\sigma^2_{bulk}$.** Estimating $\lambda^+$ (or, equivalently, $\sigma^2_{bulk}$) can be tricky, even when the spike is well-separated from the bulk.

  We illustrate this in Figure 24. In particular, compare Figure 24(b) and Figure 22(c). In

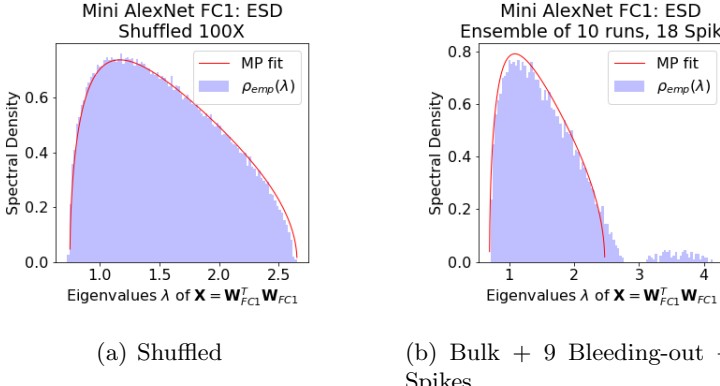

(a) Shuffled

(b) Bulk + 9 Bleeding-out + 9 Spikes

Figure 24: Illustration of fitting $\lambda^+$. ESD for Layer FC1 of MiniAlexNet, with MP fit (in red): averaged over 10 runs. Batch size 16, and no weight regularization. Compare 24(a) with Figure 22(a), which shows the ESD for a single run. Also, compare 24(b), where $\lambda^+$ is fit by removing 18 eigenvectors, with Figure 22(c), which shows "Bulk + 9 Spikes".

Figure 22(c), $\lambda^+$ is chosen to reproduce very well the bulk edge of the ESD, at the expense of having some "missing mass" in the ESD just below $\lambda^+$ (leading to a "Bulk + 9 Spikes" model). In Figure 24(b), $\lambda^+$ is chosen to reproduce very well the ESD just below $\lambda^+$, at the expense of having a slight bleeding-out region just above $\lambda^+$ (leading to a "Bulk + 18 Spikes" or a "Bulk + 9 Bleeding-out + 9 Spikes" model). If we hypothesize that a MP distribution fits the bulk very well, then the fit in Figure 24(b) is more appropriate, but Figure 22(b) shows this can be challenging to identify in a single run.

We recommend choosing the bulk maximum $\lambda^+$ and (from Eqn. (9)) selecting $\sigma^2_{bulk}$ as $\sigma^2_{bulk} = \lambda^+ \left(1 + 1/\sqrt{Q}\right)^{-2}$. In fitting $\sigma^2_{bulk}$, we expect to lose some variance due to the eigenvalue mass that "bleeds out" from the bulk (e.g., due to BLEEDING-OUT or BULK-DECAY), relative to a situation where the MP distribution provides a good fit for the entire ESD (as in RANDOM-LIKE). Rather than fitting $\sigma^2_{mp}$ directly on the ESD of $\mathbf{W}_l$, without removing the outliers (which may thus lead to poor estimates since $\lambda_{max}$ is particularly large), we can always define a baseline variance for any weight matrix $\mathbf{W}$ by shuffling it elementwise $\mathbf{W} \to \mathbf{W}^{shuf}$, and then finding the MP $\sigma^2_{shuf}$ from the ESD of $\mathbf{W}^{shuf}$. In doing so, the Frobenius norm is preserved $\|\mathbf{W}_l^{shuf}\|_F = \|\mathbf{W}_l\|_F$, thus providing a way to (slightly over-) estimate the unperturbed variance of $\mathbf{W}_l$ for comparison.[37] Since at least one eigenvalue bleeds out, $\sigma^2_{bulk} < \sigma^2_{shuf}$, i.e., the empirical bulk variance $\sigma^2_{bulk}$ will always be (slightly) less that than shuffled bulk variance $\sigma^2_{shuf}$.

The best way to automate these choices, e.g., with a kernel density estimator, remains open.

## 6.4 Effect of explicit regularization

We consider here how explicit regularization affects properties of learned DNN models, in light of baseline results of Section 6.2. We focus on $L_2$ Weight Norm and Dropout regularization.

---

[37]We suggest shuffling $\mathbf{W}_l$ at least 100 times then fitting the ESD to obtain an $\sigma^2$. We can then estimate $\sigma^2_{bulk}$ as $\sigma^2$ minus a contribution for each of the $K$ bleeding-out eigenvalues, giving, as a rule of thumb, $\sigma^2_{bulk} \sim \sigma^2 - \frac{1}{M}(\lambda_1 + \lambda_2 \cdots \lambda_K)$, $\lambda_k \in$ Bleeding-out.

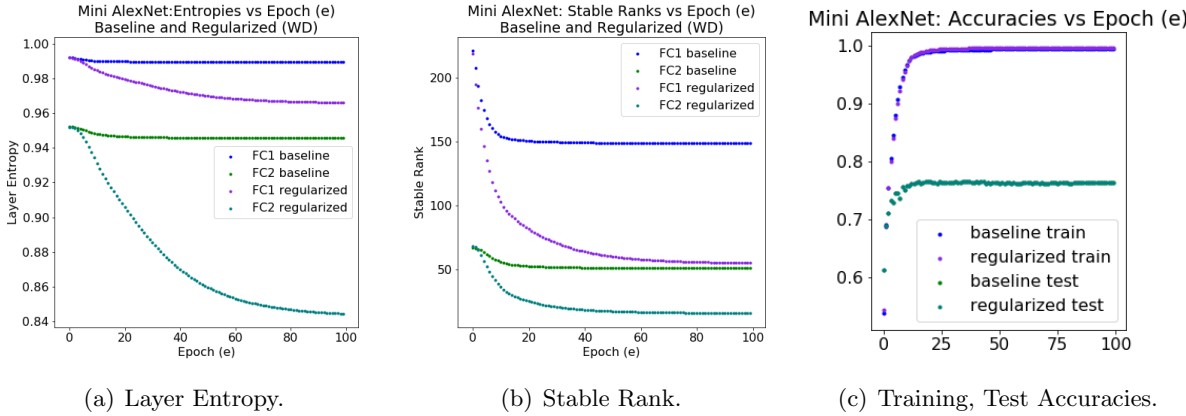

(a) Layer Entropy.  (b) Stable Rank.  (c) Training, Test Accuracies.

Figure 25: Entropy, Stable Rank, and Training and Test Accuracies of last two layers for MiniAlexNet, as a function of the number of epochs of training, without and with explicit $L_2$ norm weight regularization.

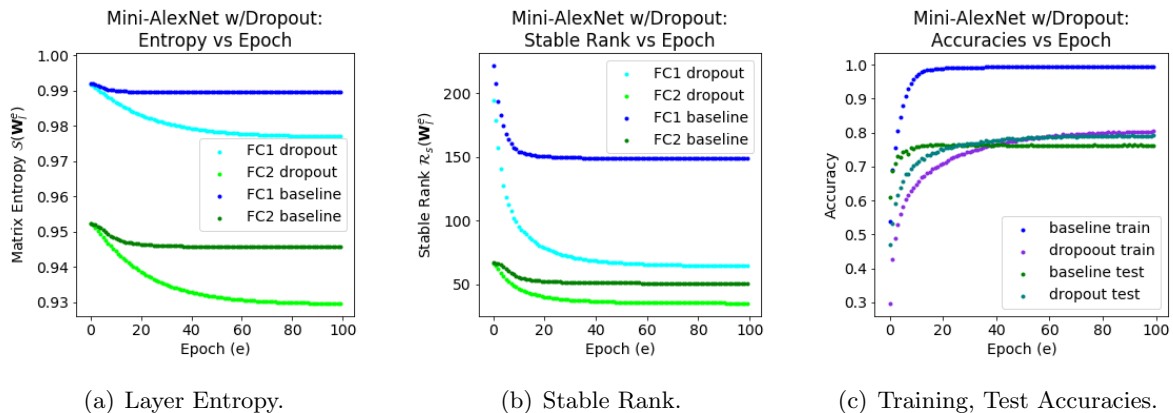

(a) Layer Entropy.  (b) Stable Rank.  (c) Training, Test Accuracies.

Figure 26: Entropy, Stable Rank, and Training and Test Accuracies of last two layers for MiniAlexNet, as a function of the number of epochs of training, without and with explicit Dropout.

**Transition in Layer Entropy and Stable Rank.** See Figure 25 for plots for FC1 and FC2 when $L_2$ norm weight regularization is included; and see Figure 26 for plots when Dropout regularization is included. In both cases, baseline results are provided, and compare with Figure 18. In each case, we observe a greater decrease in the complexity metrics with explicit regularization than without, consistent with expectations; and we see that explicit regularization affects these metrics dramatically. Here too, the Layer Entropy decreases relatively more for FC2, and the Stable Rank decreases relatively more for FC1.

**Eigenvalue Spectrum: Comparisons with RMT.** See Figure 27 for the ESD for layers FC1 and FC2 of MiniAlexNet, with explicit Dropout, including MP fits to a bulk when 9 or 10 spikes are removed. Compare with Figure 22 (for FC1) and Figure 23 (for FC2). Note, in particular, the differences in the scale of the X axis. Figure 27 shows that when explicit Dropout regularization is added, the eigenvalues in the spike are pulled to much larger values (consistent with a much more implicitly-regularized model). A subtle but important consequence of this regularization[38]

[38]This is seen in Figure 24 in the baseline, but it is seen more clearly here, especially for FC2 in Figure 27.

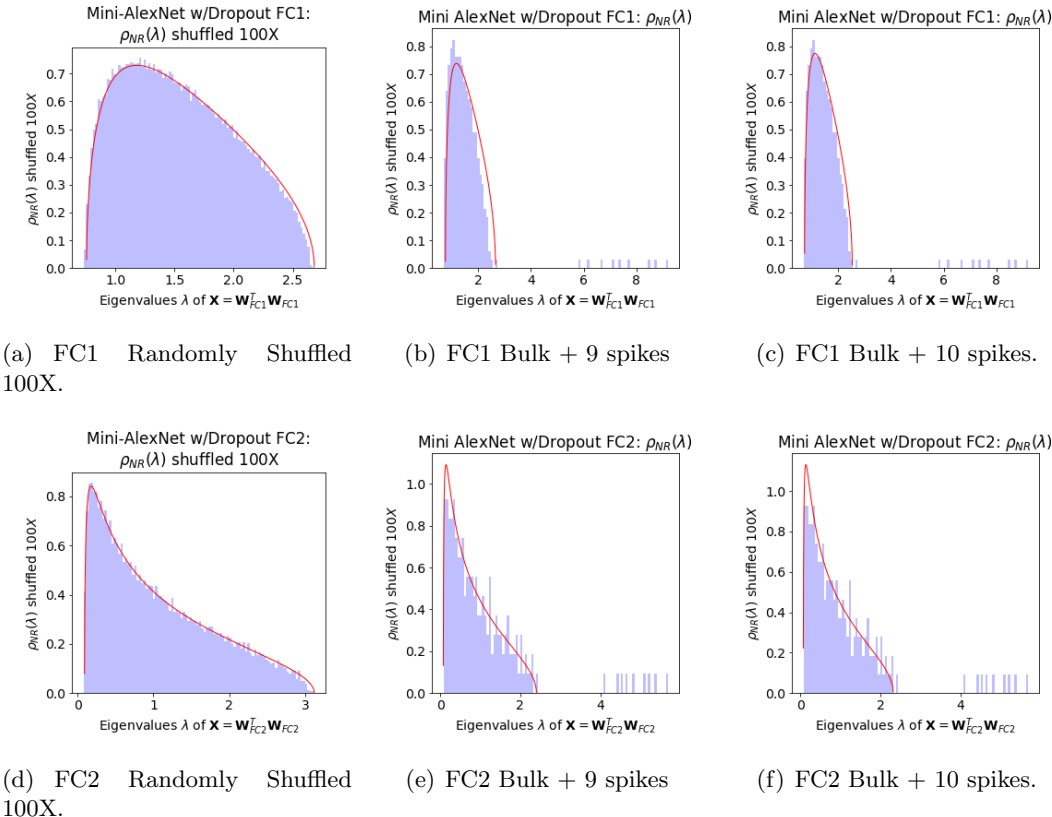

(a) FC1 Randomly Shuffled 100X.

(b) FC1 Bulk + 9 spikes

(c) FC1 Bulk + 10 spikes.

(d) FC2 Randomly Shuffled 100X.

(e) FC2 Bulk + 9 spikes

(f) FC2 Bulk + 10 spikes.

Figure 27: ESD for layers FC1 and FC2 of MiniAlexNet, with explicit Dropout. (Compare with Figure 22 (for FC1) and Figure 23 (for FC2).)

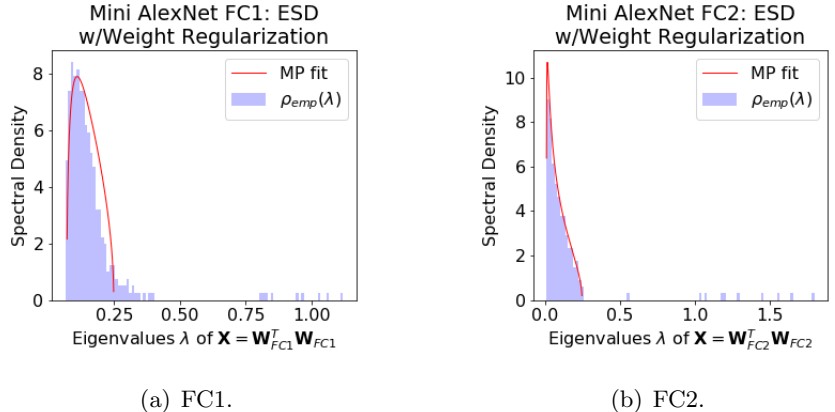

(a) FC1.

(b) FC2.

Figure 28: ESD for layers FC1 and FC2 of MiniAlexNet, with explicit $L_2$ norm weight regularization. (Compare with Figure 22 (for FC1) and Figure 23 (for FC2).)

is the following: this leads to a smaller bulk MP variance parameter $\sigma_{mp}^2$, and thus smaller values for $\lambda^+$, when there is a more prominent spike. See Figure 28 for similar results for the ESD for layers FC1 and FC2 of MiniAlexNet, with explicit $L_2$ norm weight regularization.

**Eigenvalue localization.** We observe that eigenvector localization tends to be more prominent when the explicit regularization is stronger, presumably since explicit ($L_2$ Weight Norm or Dropout) regularization can make spikes more well-separated from the bulk.

# 7 Explaining the Generalization Gap by Exhibiting the Phases

In this section, we demonstrate that we can exhibit all five of the main phases of learning by changing a single knob of the learning process.[39] We consider the batch size (used in the construction of mini-batches during SGD training) since it is not traditionally considered a regularization parameter and due to its its implications for the generalization gap phenomenon.

The *Generalization Gap* refers to the peculiar phenomena that DNNs generalize significantly less well when trained with larger mini-batches (on the order of $10^3 - 10^4$) [80, 64, 72, 57]. Practically, this is of interest since smaller batch sizes makes training large DNNs on modern GPUs much less efficient. Theoretically, this is of interest since it contradicts simplistic stochastic optimization theory for convex problems. The latter suggests that larger batches should allow better gradient estimates with smaller variance and should therefore improve the SGD optimization process, thereby increasing, not decreasing, the generalization performance. For these reasons, there is interest in the question: what is the mechanism responsible for the drop in generalization in models trained with SGD methods in the large-batch regime?

To address this question, we consider here using different batch sizes in the DNN training algorithm. We trained the MiniAlexNet model, just as in Section 6 for the Baseline model, except with batch sizes ranging from moderately large to very small ($b \in \{500, 250, 100, 50, 32, 16, 8, 4, 2\}$).

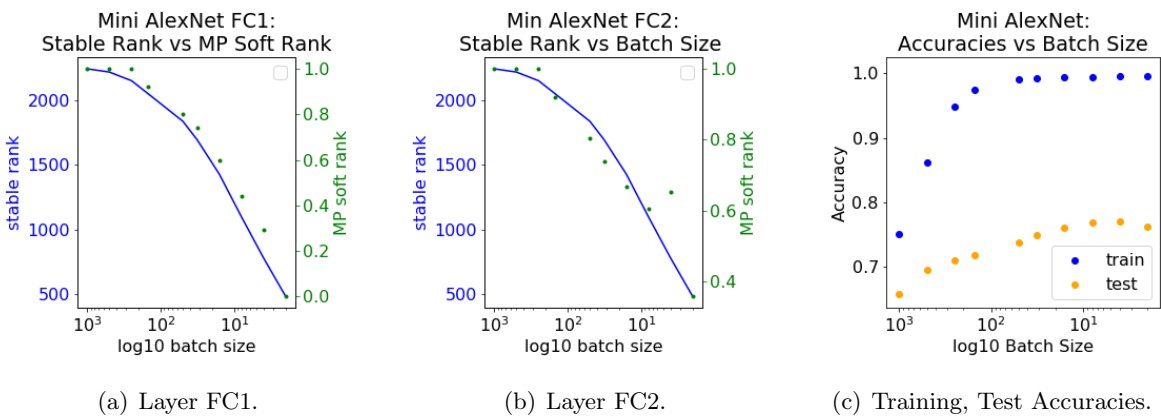

(a) Layer FC1.      (b) Layer FC2.      (c) Training, Test Accuracies.

Figure 29: Varying Batch Size. Stable Rank and MP Softrank for FC1 (29(a)) and FC2 (29(b)); and Training and Test Accuracies (29(c)) versus Batch Size for MiniAlexNet.

**Stable Rank, MP Soft Rank, and Training/Test Performance.** Figure 29 shows the Stable Rank and MP Softrank for FC1 (29(a)) and FC2 (29(b)) as well as the Training and Test Accuracies (29(c)) as a function of Batch Size for MiniAlexNet. Observe that the MP Soft Rank ($\mathcal{R}_{mp}$) and the Stable Rank ($\mathcal{R}_s$) both track each other, and both systematically *decrease* with decreasing batch size, as the test accuracy *increases*. In addition, both the training and test accuracy decrease for larger values of $b$: training accuracy is roughly flat until batch size $b \approx 100$,

---

[39]We can also exhibit the "+1" phase, but in this section we are interested in changing only the batch size.

and then it begins to decrease; and test accuracy actually increases for extremely small $b$, and then it gradually decreases as $b$ increases.

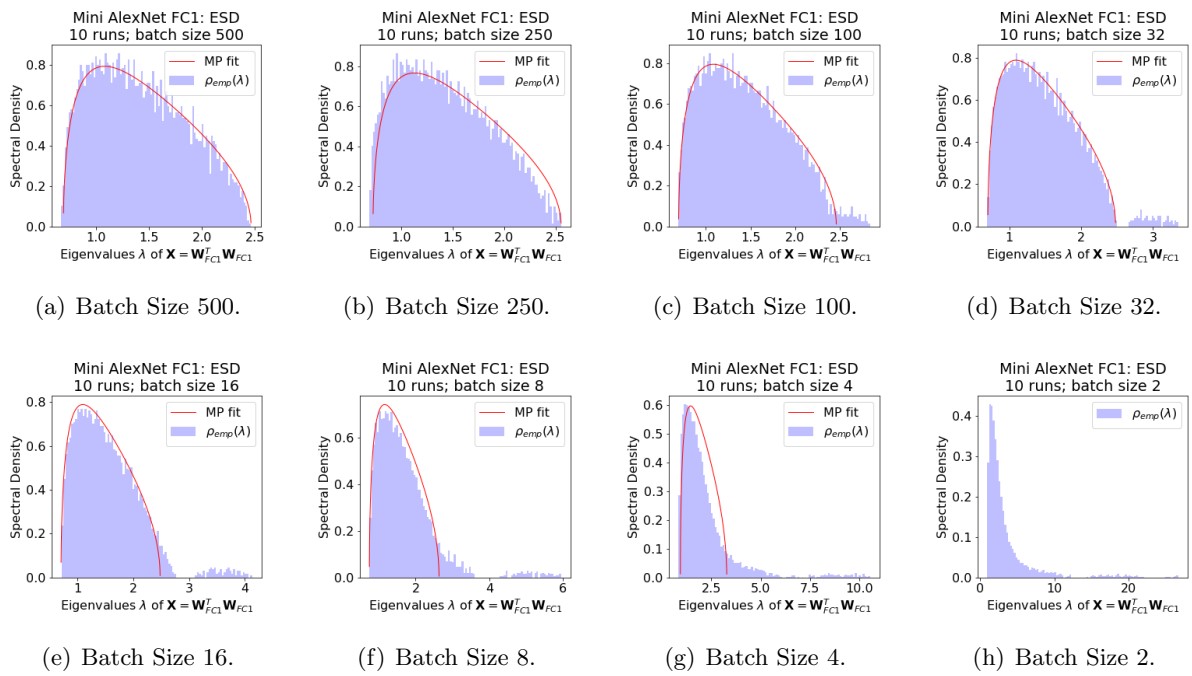

(a) Batch Size 500.     (b) Batch Size 250.     (c) Batch Size 100.     (d) Batch Size 32.

(e) Batch Size 16.     (f) Batch Size 8.     (g) Batch Size 4.     (h) Batch Size 2.

Figure 30: Varying Batch Size. ESD for Layer FC1 of MiniAlexNet, with MP fit (in red), for an ensemble of 10 runs, for Batch Size ranging from 500 down to 2. (Compare with Figure 22 as a reference.) Smaller batch size leads to more implicitly self-regularized models. For FC1, we exhibit all 5 of the main phases of training by varying only the batch size.

**ESDs: Comparisons with RMT.** Figures 30 and 31 show the final ensemble ESD for each value of $b$ for Layer FC1 and FC2, respectively, of MiniAlexNet. For both layers, we see systematic changes in the ESD as batch size $b$ decreases. Consider, first, FC1.

- At batch size $b = 250$ (and larger), the ESD resembles a pure MP distribution with no outliers/spikes; it is RANDOM-LIKE.

- As $b$ decreases, there starts to appear an outlier region. For $b = 100$, the outlier region resembles BLEEDING-OUT.

- Then, for $b = 32$, these eigenvectors become well-separated from the bulk, and the ESD resembles BULK+SPIKES.

- As batch size continues to decrease, the spikes grow larger and spread out more (observe the increasing scale of the X-axis), and the ESD exhibits BULK-DECAY.

- Finally, at the smallest size, $b = 2$, extra mass from the main part of the ESD plot almost touches the spike, and the curvature of the ESD changes, consistent with HEAVY-TAILED.

While the shape of the ESD is different for FC2 (since the aspect ratio of the matrix is less), very similar properties are observed. In addition, as $b$ decreases, some of the extreme eigenvectors associated with eigenvalues that are not in the bulk tend to be more localized.

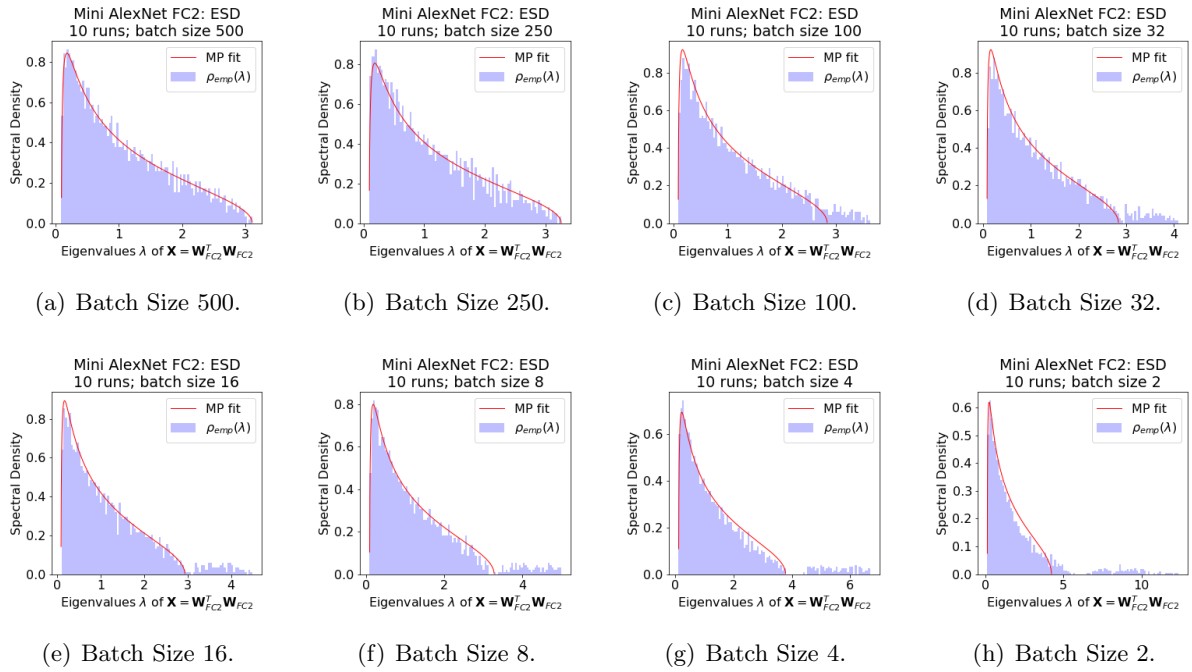

Figure 31: Varying Batch Size. ESD for Layer FC2 of MiniAlexNet, with MP fit (in red), for an ensemble of 10 runs, for Batch Size ranging from 500 down to 2. (Compare with Figure 23 as a reference.) Smaller batch size leads to more implicitly self-regularized models. For FC2, we exhibit 4 of the 5 of the main phases of training by varying only the batch size.

**Implications for the generalization gap.** Our results here (both that training/test accuracies decrease for larger batch sizes and that smaller batch sizes lead to more well-regularized models) demonstrate that the generalization gap phenomenon arises since, for smaller values of the batch size $b$, the DNN training process itself implicitly leads to stronger Self-Regularization. (Depending on the layer and the batch size, this Self-Regularization is either the more traditional Tikhonov-like regularization or the Heavy-Tailed Self-Regularization corresponding to strongly-correlated models.) That is, training with smaller batch sizes implicitly leads to more well-regularized models, and it is this regularization that leads to improved results. The obvious mechanism is that, by training with smaller batches, the DNN training process is able to "squeeze out" more and more finer-scale correlations from the data, leading to more strongly-correlated models. Large batches, involving averages over many more data points, simply fail to see this very fine-scale structure, and thus they are less able to construct strongly-correlated models characteristic of the HEAVY-TAILED phase. Our results also suggest that, if one hopes to compensate for this by decreasing the learning rate, then one would have to decrease the learning rate by an extraordinary amount.

# 8 Discussion and Conclusion

There is a large body of related work, much of which either informed our approach or should be informed by our results. This includes: work on large-batch learning and the generalization gap [148, 72, 64, 57, 67, 131, 66, 146, 94, 151, 152]; work on Energy Landscape approaches to NN training [70, 149, 44, 123, 30, 29, 27, 65, 47, 12, 104, 150, 50, 83, 82, 95]; work on using weight matrices or properties of weight matrices [15, 102, 103, 4, 14, 153, 101, 3, 86, 98]; work on

different Heavy-Tailed Universality classes [46, 32, 18, 25, 20, 5, 111, 7, 38, 17, 90, 8, 99]; other work on RMT approaches [133, 120, 114, 112, 87, 87, 137, 85, 126]; other work on statistical physics approaches [132, 52, 117, 125, 137, 119, 113]; work on fitting to noisy versus reliable signal [136, 156, 75, 122, 6]; and several other related lines of work [63, 96, 116, 34, 1, 106, 107, 97, 84]. We conclude by discussing several aspects of our results in this broader context.

## 8.1  Some immediate implications

**Failures of VC theory.**  In light of our results, we have a much better understanding of why VC theory does not apply to NNs. VC theory assumes, at its core, that a learning algorithm could sample a very large, potentially infinite, space of hypothesis functions; and it then seeks a uniform bound on this process to get a handle on the generalization error. It thus provides a very lose, data-independent bound. Our results suggest a very different reason why VC theory would fail than is sometimes assumed: naïvely, the VC hypothesis space of a DNN would include all functions described by all possible values of the layer weight matrices (and biases). Our results suggest, in contrast, that the actual space is in some sense "smaller" or more restricted than this, in that the FC layers (at least) cover only one Universality class—the class of Heavy (or Fat) Tailed matrices, with PL exponent $\mu \in [2, 4]$. During the course of training, the space becomes smaller—through Self-Regularization—since even if the initial matrices are random, the class of possible final matrices is very strongly correlated. The process of Self-Regularization and Heavy-Tailed Self-Regularization collapses the space of available functions that can be learned. Indeed, this also suggests why transfer learning is so effective—the initial weigh matrices are much closer to their final versions, and the space of functions need not shrink so much. The obvious conjecture is that what we have observed is characteristic of general NN/DNN learning systems. Since there is nothing like this in VC theory, our results suggest revisiting more generally the recent suggestions of [93].

**Information bottleneck.**  Recent empirical work on modern DNNs has shown two phases of training: an initial "fast" phase and a second "slower" phase. To explain this, Tishby et al. [141, 129] have suggested using the Information Bottleneck Theory for DNNs. See also [140, 128, 124, 154]. While this theory may be controversial, the central concept embodies the old thinking that DNNs implicitly lose some capacity (or information/entropy) during training. This is also what we observe. Two important differences with our approach are the following: we provide *a posteriori* guarantees; and we provide an unsupervised theory. An *a posteriori* unsupervised theory provides a mechanism to minimize the risk of "label leakage," clearly a practical problem. The obvious hypothesis is that the initial fast phase corresponds to the initial drop in entropy that we observe (which often corresponds to a Spike pulling out of the Bulk), and that the second slower phase corresponds to "squeezing out" more and more correlations from the data (which, in particular, would be easier with smaller batches than larger batches, and which would gradually lead to a very strongly-correlated model that can then be modeled by Heavy-Tailed RMT).

**Energy landscapes and rugged convexity.**  Our observations about the onset of Heavy-Tailed or scale-free behavior in realistic models suggest that (relatively) simple (i.e., Gaussian) Spin-Glass models, used by many researchers, may lead to very misleading results for realistic systems. Results derived from such models are very specific to the Gaussian Universality class; and other Spin-Glass models can show very different behaviors. In particular, if we select the elements of the Spin-Glass Hamiltonian from a Heavy-Tailed Levy distribution, then the local minima *do not concentrate* near the global minimum [31, 32]. See also [149, 49, 48, 28, 138]. Based on this, as well as the results we have presented, we expect that well-trained DNNs will exhibit a

*ruggedly convex global energy landscape*, as opposed to a landscape with a large number of very different degenerate local minima. This would clearly provide a way to understand phenomena exhibited by DNN learning that are counterintuitive from the perspective of traditional ML [93].

**Connections with glass theory.**  It has been suggested that the slow training phase arises because the DNN optimization landscape has properties that resemble a glassy system (in the statistical physics sense), meaning that the dynamics of the SGD is characterized by slow Heavy-Tailed or PL behavior. See [13, 72, 64, 10]—and recall that, while this connection is sometimes not explicitly noted, glasses are *defined* in terms of their slow dynamics. Using the glass analogy, however, it can also shown that very large batch sizes can, in fact, be used—if one adjusts the learning rate (potentially by an extraordinary amount). For example, it is argued that, when training with larger batch sizes, one needs to change the learning rate adaptively in order to take effectively more times steps to reach a obtain good generalization performance. Our results are consistent with the suggestion that DNNs operate near something like a finite size form of a spin-glass phase transition, again consistent with previous work [93]. This is likewise similar in spirit to how certain spin glass models are Bayes optimal in that their optimal state lies on the Nishimori Line [105]. Indeed, these ideas have been a great motivation in looking for our empirical results and formulating our theory.

**Self-Organization in Natural (and Engineered) Phenomena.**  Typical implementations of Tikhonov regularization require setting a specific regularization parameter or regularization size scale, whereas *Self-Regularization* just arises as part of the DNN training process. A different mechanism for this has been described by Sornette, who suggests it can arise more generally in natural *Self-Organizing* systems, without needing to tune specific exogenous control parameters [134]. Such Self-Organization can manifest itself as BULK+SPIKES [92], as true (infinite order) Power Laws, or as a finite-sized HEAVY-TAILED (or Fat-Tailed) phenomena [134]. This corresponds to the three Heavy-Tailed Universality classes we described. To the best of our knowledge, ours is the first observation and suggestion that a Heavy-Tailed ESD could be a signature/diagnostic for such Self-Organization. That we are able to induce both BULK+SPIKES and HEAVY-TAILED Self-Organization by adjusting a single internal control parameter (the batch size) suggests similarities between Self-Organized Criticality (SOC) [11] (a very general phenomena also thought to be "a fundamental property of neural systems" more generally [62, 36]) and modern DNN training.

## 8.2  Theoretical niceties, or Why RMT makes good sense here

There are subtle issues that make RMT particularly appropriate for analyzing weight matrices.

**Taking the right limit.**  The matrix $\mathbf{X}$ is an empirical correlation matrix of the weight layer matrix $\mathbf{W}_l$, akin to an estimator of the true covariance of the weights. It is known, however, that this estimator is not good, unless the aspect ratio is very large (i.e., unless $Q = N/M \gg 1$, in which case $\mathbf{X}_l$ is *very* tall and thin). The limit $Q \to \infty$ (e.g., $N \to \infty$ for fixed $M$) is the case usually considered in mathematical statistics and traditional VC theory. For DNNs, however, $M \sim N$, and so $Q = \mathcal{O}(1)$; and so a more appropriate limit to consider is $(M \to \infty, N \to \infty)$ such that $Q$ is a fixed constant [93]. This is the regime of MP theory, and this is why deviations from the limiting MP distribution provides the most significant insights here.

**Relation to the SMTOG.**  In recent work [93], Martin and Mahoney examined DNNs using the Statistical Mechanics Theory of Generalization (SMTOG) [127, 147, 60, 43]. As with RMT,

the STMOG also applies in the limit $(M \to \infty, N \to \infty)$ such that $Q = 1$ or $Q = \mathcal{O}(1)$, i.e., in the so-called Thermodynamic Limit. Of course, RMT has a long history in theoretical physics, and, in particular, the statistical mechanics of the energy landscape of strongly-correlated disordered systems such as polymers. For this reason, we believe RMT will be very useful to study broader questions about the energy landscape of DNNs. Martin and Mahoney also suggested that overtrained DNNs—such as those trained on random labelings—may effectively be in a finite size analogue of the (mean field) spin glass phase of a neural network, as suggested by the SMTOG [93]. We should note that, in this phase, self-averaging may (or may not) break down.

**The importance of Self-Averaging.** Early RMT made use of replica-style analysis from statistical physics [127, 147, 60, 43], and this assumes that the statistical ensemble of interest is *Self-Averaging*. This property implies that the theoretical ESD $\rho(\lambda)$ is independent of the specific realization of the matrix $\mathbf{W}$, provided $\mathbf{W}$ is a *typical* sample from the true ensemble. In this case, RMT makes statements about the empirical ESD $\rho_N(\lambda)$ of a large random matrix like $\mathbf{X}$, which itself is drawn from this ensemble. To apply RMT, we would like to be able inspect a single realization of $\mathbf{W}$, from one training run or even one epoch of our DNN. If our DNNs are indeed self-averaging, then we may confidently interpret the ESDs of the layer weight matrices of a single training run.

As discussed by Martin and Mahoney, this may *not* be the case in certain situations, such as *severe overtraining* [93]. From the SMTOG perspective, NN overfitting,[40] which results in NN overtraining,[41] is an example of non-self-averaging. When a NN generalizes well, it can presumably be trained, using the same architecture and parameters, on any large random subset of the training data, and it will still perform well on any test/holdout example. In this sense, the trained NN is a typical random draw from the implicit model class. In contrast, an overtrained model is when this random draw from this implicit model class is *atypical*, in the sense that it describes well the training data, but it describes poorly test data. A model can enter the spin glass phase when there is not enough training data and/or the model is too complicated [127, 147, 60, 43]. The spin glass phase is (frequently) non-self-averaging, and this is why overtraining was traditionally explained using spin glass models from statistical mechanics.[42] For this reason, it is not obvious that RMT can be applied to DNNs that are overtrained; we leave this important subtly for future work.

## 8.3 Other practical implications

Our practical theory opens the door to address very practical questions, including the following.

- What are design principles for good models? Our approach might help to incorporate domain knowledge into DNN structure as well as provide finer metrics (beyond simply depth, width, etc.) to evaluate network quality.

- What are ways in which adversarial training/learning or training/learning in new environments affects the weight matrices and thus the loss surface? Our approach might help characterize robust versus non-robust and interpretable versus non-interpretable models.

- When should training be discontinued? Our approach might help to identify empirical properties of the trained models, e.g., of the weight matrices—*without* explicitly looking at labels—that will help to determine when to *stop* training.

---

[40] Overfitting is due to a small load parameter, e.g., due to insufficient reasonable-quality data for the model class.

[41] Overtraining occurs when, once it has been trained with the data, the NN does not generalize well.

[42] Thus, overfitting leads to non-self-averaging, which results in overtraining (but it is not necessarily the case that overtraining implies overfitting).

Finally, one might wonder whether our RMT-based theory is applicable to other types of layers such as convolutional layers and/or other types of data such as natural language data. Initial results suggest yes, but the situation is more complex than the relatively simple picture we have described here. These and related directions are promising avenues to explore.

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
