# OpenReview forum: "Traditional and Heavy Tailed Self Regularization in Neural Network Models"
_ICLR.cc/2019/Conference_

### Official Review · AnonReviewer2 · 2018-10-15
**A valuable contribution; yet, not well polished**

**Rating:** 6
**Confidence:** 5

**Review:**

The paper attempts to examine the reasons behind the strong generalisation performance of DNNs trained via SGD. The authors propose an analysis which offers a fresh view to this problem. This view has been articulated very well, and is based on sound mathematical arguments.

On the other hand, since there is no formal theorem to support the introduced assumptions, the authors have attempted to provide empirical evidence through experiments with standard DNN architectures and benchmark datasets. However, this is where the weakness of this paper lies: The provided empirical evidence, while nicely executed, is not enough to convince the critical reader. We need experiments with more diverse datasets and experimental setups.

Although I accept the claim of the authors concerning the lack of space, they could also trim the Introduction so as to free up some space, as well as provide an indefinite number of extra supporting evidence in the form of Supplementary Material/Appendices.

---

### Official Review · AnonReviewer3 · 2018-11-02
**Interesting idea but not made rigorous/precise**

**Rating:** 4
**Confidence:** 1

**Review:**

The paper analyzes the empirical spectral density of DNN layher matrices and compares them with the traditionally-regularized statistical models, and develop a theory to identify 5+1 phases of training based on it. The results with different batch sizes illustrate the Generalization Gap pheneomena, and explains it as being causes by implicit self-regularization.

However, the paper seems a little bit handwavy to me, without any serious theoretical justification. For example, why are \mu=2 and 4 chosen as the threshold between weakly/moderately/very heavy-tailed? In addition, the paper is build upon o the 5+1 model as in Figure 2 and the graphical comparison between the empirical ESD and the expected ESD of the five models in Table 1, and they lack any mathematical/rigorous definition---see table 2. The simulations are performs over a particular data set and a particular setting, and I wonder if the observations would be different for a different data set and a different setting.

As a result, it may give some important intuition, but the content is not sufficiently rigorous to my knowledge.

---

> ### Public Comment · (anonymous) · 2018-11-08
> **Response to AnonReviewer3**
>
> I seriously question the good-faith of this reviewer and even s/he read the paper in detail. AnonReviewer3 is declining to accept empirical evidence. If you are going to make an "educated guess" without basic understanding, don't review highly technical papers.
>
> 1.   \mu ranges are basically lack of reviewers RMT knowledge. Paper rigorously cites the findings where how this range is applied. These are established solidly in RMT.
> 2.  Reviewer complaint on Figure 2 and Table 1 saying "they lack any mathematical/rigorous definition". This is absolutely not true. Spectral theory is very well established in mathematical theory along with RMT this paper uses.
> 3. "if the observations would be different for a different data set and a different setting. " This is totally absurd. There is no way that any single machine learning paper can test all datasets and different settings. Paper used large range of different architectures and presented the empirical evidence as any published machine learning paper.
>
> I strongly recommend Chairs to not give any further reviews to this reviewer.  Reviewing without good-faith is a disservice to the community.

---

### Official Review · AnonReviewer4 · 2018-11-08
**Although the experiments seem interesting,  theoretical results are not clearly presented.**

**Rating:** 4
**Confidence:** 4

**Review:**

This manuscript studies the implicit regularization of neural networks from the perspective of random matrix theory. The authors provide both empirical and theoretical results that aim to show that the empirical spectral density of weights of DNN captures the implicit regularization phenomenon. However, the results are far from rigorous theory and it is not clear how recent results in MP theory yields the statements made in the paper.


Detailed comments:

1. The empirical studies seem interesting. It seems that two kinds of results are shown. The first one is that ESD fits perfectly for small models, and the second one is that deep models fit heavy-tailed random matrices class. It would be interesting to see more details about how these models are trained, as training greatly affects the value of the weights.

2. Theoretical results are not clearly stated. In section 2, the authors introduce the basics of MP theory. However, it is not clear how to derive the theory in this paper based on the MP theory. It seems that the main theory is the "5+1 phases of training". The definition of these 6 phases is not even explicitly given in this paper. Moreover, the theory of all these phases seems to depend on equation (3) , but there are no lemmas or propositions that gives a rigourious theoretical guarantee.

---

### Public Comment · ~Gershon_Mathew_Wolfe1 · 2018-10-18
**Great paper that goes into detail regarding the statistical mechanics of machine learning and explains the energy landscape**

I believe the only people who will advance ML are those with an understanding of statistical mechanics.  Your ideas of how to treat the energy landscape are spot on!  You are doing some great work.  My intuition tells me that self organization plays a large roll in how to treat the weight matrices.

---

### Public Comment · ~Mehmet_Suezen1 · 2018-10-19
**Spectral properties of deep learning architectures are key to understand generalisation problems**

Authors have shown a clearly original contribution by directly mapping a classification of Random Matrix Theory (RMT) to resulting weight matrices of deep architectures empirically. The resulting taxonomy would help in understanding under what circumstances trained networks generalised better. This approach is novel because it avoids causality objections while it classify networks without any response metric.  Moreover, the intimate connection between statistical mechanics and deep learning architectures are often omitted by practitioners up to now, but it may be a mainstream approach in the near future. I strongly recommend this paper for publication.

---

### Author Response · Authors · 2018-11-21
**Please read our full response/comment to reviewer comments, or read "Response to AnonReviewer3".**


Thanks to the two positive reviewers who were willing to write their names.

Regarding AnonReviewer3, we will simply point to the rebuke from anonymous "Response to AnonReviewer3".  We could not have said it better ourselves.

Actually, some of these comments also are relevant for AnonReviewer4.

AnonReviewer4 thinks "The empirical studies seem interesting." but would like to see "more details about how these models are trained".  One of the points of our methods it that we can apply them to already-trained models, as we did.  In particular, if you train a model, e.g., in a way that is not reproducible by others, then we can still apply our theory to that model.  (We also tested the theory on models we trained with mini-AlexNet, and more details on that are provided in the 59 page supplementary material.)  AnonReviewer4 also says that "training greatly affects the value of the weights".  This is not true.  One of our main empirical observations is that nearly every state-of-the-art pre-trained network exhibits these properties.  (Again, see the 59 page supplementary material for more.)  This empirical observation itself should merit publication of this paper.

Our theory is a phenomenological predictive theory.  That means it does not include lemmas.  Also our theory does not depend on Eqn. (3).  That equation is just a heuristic guide to understand our theory, for readers familiar with models of that form.  We are not interested in gratuitous rigor.  We are interested in a predictive theory, which is what we have used RMT to construct.  Our theory predicts 5 phases of learning, and we are able to exhibit all 5 phases of learning by changing a single knob, the batch size.  That theoretical prediction and empirical validation itself should merit publication of this paper.

AnonReviewer2 has said "The provided empirical evidence, while nicely executed, is not enough to convince the critical reader. We need experiments with more diverse datasets and experimental setups."  That's hard to do in the page limitation.  For that reason, we have uploaded a revised version of the paper, identical to what we submitted, but with a 59 page supplementary material, which can also serve as a technical report version of this page-limited submission.

Deep learning is by its very nature an empirical science.  It seems to violate a number of basic notions, most notably that a non-convex optimization problem can be applied successfully in such a wide number of problem domains and generalize so well.  It seems likely that it is not possible to understand this, i.e. why deep learning even works, relying purely on very loose, very general theoretical techniques.  We suggest that the reviewers and senior reviewers eschew false rigor, and instead focus on the empirical contributions and the theory that makes predictions that we validate.

---

### Meta-Review · Area_Chair1 · 2018-12-11
**Concerns on the justifications and practical values of the proposed theorems**

**Confidence:** 5
**Recommendation:** Reject

**Metareview:**

While it appears that the authors have done significant amount of work to investigate this topic, there are concerns that the theorems are not rigorously/precisely presented, and it is unclear how they can guide the design and training of neural network models in practice. The response and revision of the authors do not provide sufficient materials to address these concerns.